# Over a ten-year record of aerosol optical properties at SMEAR II

Krista Luoma[1], Aki Virkkula[1,2], Pasi Aalto[1], Tuukka Petäjä[1] and Markku Kulmala[1]

[1]Institute for Atmospheric and Earth System Research, University of Helsinki, Helsinki, 00014, Finland
[2]Finnish Meteorological Institute, Helsinki, 00560, Finland

5 *Correspondence to*: Krista Luoma (krista.q.luoma@helsinki.fi), Aki Virkkula (aki.virkkula@fmi.fi)

**Abstract.** Aerosol optical properties (AOPs) describe the ability of aerosols to scatter and absorb radiation at different wavelengths. Since aerosol particles interact with the sun's radiation, they impact the climate. Our study focuses on the long-term trends and seasonal variations of different AOPs measured at a rural boreal forest site in Northern Europe. To explain the observed variations in the AOPs, we also analyzed changes in the aerosol size distribution. AOPs of particles smaller than 10

10 μm (PM10) and 1 μm (PM1) have been measured at SMEAR II, in Southern Finland, since 2006 and 2010, respectively. For PM10 particles, the median values of the scattering and absorption coefficients, single-scattering albedo, and backscatter fraction at $\lambda$ = 550 nm were 9.8 Mm$^{-1}$, 1.3 Mm$^{-1}$, 0.88 and 0.14. The median values of scattering and absorption Ångström exponents at the wavelength ranges 450–700 nm and 370–950 nm were 1.88 and 0.99, respectively. We found statistically significant trends for the PM10 scattering and absorption coefficients, single-scattering albedo, and backscatter fraction, and

15 the slopes of these trends were -0.32 Mm$^{-1}$, -0.086 Mm$^{-1}$, 2.2·10$^{-3}$, and 1.3·10$^{-3}$ per year. The tendency for the extensive AOPs to decrease correlated well with the decrease in aerosol number and volume concentrations. The tendency for the backscattering fraction and single-scattering albedo to increase indicates that the aerosol size distribution consist of less larger particles and that aerosols absorb less light than at the beginning of the measurements. The trends of the single-scattering albedo and backscattering fraction influenced the aerosol radiative forcing efficiency, indicating that the aerosol particles are scattering

the radiation more effectively back into space.

## 1 Introduction

Aerosols affect the radiative balance of the atmosphere both directly by aerosol–radiation interactions (ARI), i.e., by scattering and absorbing solar radiation and by absorbing and emitting terrestrial infrared radiation, and indirectly by aerosol–cloud interactions (ACI), i.e., by influencing the properties and processes of clouds (Charlson et al., 1992; Lohmann and Feichter,

2005; Ramanathan et al., 2001; Stocker, 2013). The uncertainty of the estimated radiative forcing of climate by ACI is larger than that by ARI but also the latter is substantial (Stocker, 2013). Both ARI and ACI have been shown to be responsible of dimming, the reduction of solar radiation received at the surface of the Earth (Wild, 2009, 2012; Stocker et al., 2013). Dimming and brightening have been shown to be often reconcilable with the trends in anthropogenic emissions of aerosols and their precursors and atmospheric aerosol loadings (Wild, 2012).

Aerosol optical properties (AOPs) describe the ability of aerosol particles to absorb and scatter radiation at different wavelengths. Knowing how aerosol particles interact with radiation is essential in determining the direct effect that aerosols have on the climate. The direct effect of aerosol can either be warming or cooling, depending on the AOPs and the properties of the surface below the aerosol layer (Haywood and Shine, 1995). Determining the global radiative forcing (RF) related to

the direct effect of aerosol particles has vast uncertainties (Stocker, 2013), which are due to the wide spatial and temporal variations of the number concentration, chemical composition and size distribution of aerosol particles, so it is challenging to consider them in climate models.

Making long-term observations of aerosol concentrations and properties at several regionally representative sites is necessary

to understand and quantify the global influence of aerosols (e.g. Laj et al., 2009; WMO/GAW, 2012; Weatherhead et al., 2018; Andrews et al., 2019). There are several networks of stations where such measurements are conducted, both global and regional (Pandolfi et al., 2018; WMO/GAW, 2012). The goal of the Global Atmosphere Watch (GAW) is to ensure long-term measurements of atmospheric variables in order to detect trends and reasons for those trends (WMO/GAW, 2012). This can be considered as the goal of all long-term aerosol measurements. Trends in AOPs can also be used as indicators of emission

control measures (Pandolfi et al., 2018). Recently Collaud Coen et al. (2013) presented trends of in situ AOPs at 24 GAW and IMPROVE stations, Sherman et al. (2015) presented trends of AOPs at four North American surface monitoring sites, Lihavainen et al. (2015) presented trends of AOPs at the Pallas GAW station, and Pandolfi et al. (2018) presented trends of scattering coefficients at 28 ACTRIS observatories located mainly in Europe.

Here we present the results of long-term measurements of AOPs at SMEAR II (Station for Measuring Ecosystem–Atmosphere Relations; Hari and Kulmala, 2005). The location represents the typical conditions of a boreal forests (Hari et al., 2013), which are a source of new aerosol particles formed by gas-to-particle conversion (Kulmala et al., 2004; Kulmala et al., 2013). Boreal forests (also known as Taiga) cover approximately 30 % of the world's forests and 8 % of the Earth's surface, so they greatly affect the global radiation budget.

AOPs at SMEAR II have previously been discussed by Virkkula et al. (2011), Zieger et al. (2015), and Pandolfi et al. (2018). Virkkula et al. (2011) presented the scattering and absorption data from a 3-year period (2006–2009), Zieger et al. (2015) presented the hygroscopic properties of AOPs measured during a campaign in May–Augusts in 2013, and Pandolfi et al. (2018) included SMEAR II in the paper on aerosol scattering at 28 ACTRIS stations. At SMEAR II, the study by Pandolfi et al. (2018)

involved nephelometer data, from 2006 to 2015, but did not include absorption data.

Long time series (2006–2017) of both scattering and absorption coefficients together at SMEAR II have not been presented before. The aim of this study is to present the characteristics and the temporal variation, especially trends, of AOPs at SMEAR II in this period. We also present the optical properties of particles smaller than 1 µm in diameter (PM1) that were presented

neither by Virkkula et al. (2011) nor Pandolfi et al. (2018). To be consistent with the GAW recommendations (WMO/GAW, 2016), we present the results for dry aerosol particles (RH < 40 %), if not stated otherwise.

## 2 Measurements and methods

### 2.1 The boreal research station SMEAR II

The measurements were conducted at SMEAR II (Station for Measuring Ecosystem–Atmosphere Relations; Hari and Kulmala, 2005). SMEAR II is located in Hyytiälä, Southern Finland (61° 51' N, 24° 17' E, 181 m above sea level.), in middle of a forest that consists mostly of Scots pine (*Pinus sylvestris* L.) trees (Hari et al., 2013). SMEAR II is classified as a rural measurement station and there are no large pollution sources nearby the station. The nearest larger cities, Tampere (220 000 inhabitants) and Jyväskylä (140 000 inhabitants), are located about 60 and 100 km from the measurement station. Otherwise, the area is sparsely
populated.

### 2.2 Instrumentation

#### 2.2.1 Measurements of AOPs

The data were measured between 21 June 2006 and 31 December 2017. Measurements of AOPs at SMEAR II started in 2006
for aerosol particles smaller than 10 µm in diameter (PM10). The PM10 measurements are sensitive to coarse particles that are typically primary and originated from natural sources, such as soil dust and sea salt. To obtain additional information about submicron particles, parallel measurements of AOPs for PM1 were launched in June 2010. Motivation to measure also PM1 particles is that secondary aerosols (both natural and anthropogenic), and anthropogenic primary aerosols are typically submicron particles. Having measurements for different cut-offs makes the measurements also more comparable between
different stations, since stations might use different cut-off sizes. This is also in line with the GAW recommendation that the aerosol supplied to the nephelometer should be size-segregated to determine the total (< 10 µm diameter) and submicron aerosol light scattering coefficient (WMO/GAW, 2003).

AOPs are often divided into two different categories: extensive and intensive. Extensive AOPs, such as scattering and
absorption coefficients and aerosol optical depth depend on the amount of the particles whereas the intensive AOPs depend on the nature of the aerosol, such as size, shape and chemical composition. Intensive AOPs describe for instance the fraction of aerosol light extinction due to scattering (single-scattering albedo), the wavelength dependence of scattering and absorption (Ångström exponents), and the angular dependence of scattering (hemispheric backscatter fraction) (Ogren, 1995; Sheridan and Ogren, 1999). Intensive properties are calculated from the scattering, backscattering and absorption measurements at

different wavelengths. The backscatter fraction (*b*) and the Ångström exponent of scattering ($\alpha_{sca}$) depend on particle size so by measuring the AOPs at different wavelengths, we can also obtain indirect information on the size distribution.

The extensive AOPs, which are scattering, backscattering and absorption coefficients ($\sigma_{sca}$, $\sigma_{bsca}$ and $\sigma_{abs}$), were measured at several wavelengths using an integrating nephelometer (TSI model 3563) and an aethalometer (Magee Scientific model AE-31), since 2013 also with a Multi-Angle Absorption Photometer (MAAP, Thermo Scientific model 5012). The integrating nephelometer measures scattering and backscattering at blue, green and red wavelengths (450, 550 and 700 nm) and the aethalometer measures absorption at seven wavelengths ranging from the ultraviolet to the near-infrared (370, 470, 520, 590, 660, 880 and 950 nm). Here, absorption data from the AE-31 and scattering data from the TSI3563 were used since they have the longest time series, and an important part of our discussion is the analysis of trends. We used the MAAP data in determining a multiple scattering correction factor for the Aethalometer to get more accurate absorption measurements (see Sect. 2.3.2).

Scattering and absorption measurements were recorded at a 5-minute resolution before June 2010 and after that with a 10-minu resolution. From June 2006 to June 2010, the measurements were conducted for the PM10 particles only and since June 2010 also for the PM1 particles. The sample air was taken through a PM10 inlet (Digitel, Low volume PM10 inlet) and led alternatingly either directly to the instruments or via an impactor that removes particles larger than 1 µm in diameter. The path of the sample alternated every ten minutes.

The aerosol hygroscopic growth is often significant when relative humidity (RH) increases above ~40 ± 5% and therefore the World Meteorological Organization and Global Atmosphere Watch (WMO and GAW) recommends aerosol monitoring stations to keep sample air RH lower than that (WMO/GAW, 2016). Until March 2010, the integrating nephelometer and the aethalometer measured sample air that was not dried with any external driers. The sample air was only dried passively by letting it warm from the outdoor temperature to the room temperature (about 22 °C). During winter, RH remained below 40 %, since the difference between the outdoor and the room temperatrue was high. In summer, however, the temperature difference was a lot lower. Therefore, sometimes in summer, RH of the sample exceeded the 40 % limit. If the RH was above 40 %, the data were flagged as invalid and omitted from the data analysis, if not stated otherwise. When we discuss about dry aerosols, we mean that the measurements were condcucted for sample air that had RH < 40 %.

### 2.2.2 Measurements of aerosol size distribution

To study the causalities between the AOPs and the aerosol size distribution, we included the measurements of aerosol size distribution in our study. The size distributions were measured with a Twin Differential Mobility Particle Sizer (TDMPS) in the size range 3–1000 nm (Aalto et al., 2001) and a TSI Aerodynamic Particle Sizer (APS, Model 3321) in the size range 0.53–10 µm. In the overlapping range of the TDMPS and the APS, we used the number concentrations from the TDMPS up to 700

nm. The TDMPS and APS located in the same building as the integrating nephelometer and the aethalometer. The TDMPS and APS had their own individual measurement lines. In the TDMPS measurement line, there was an inlet removing particles larger than 1 µm. There was no active drying system in the TDMPS sample line to prevent particle losses. However, the sheath flows, used in the TDMPS system, were dried (RH < 40 %) so the particles were sampled in dry conditions. In the APS measurement line there was a pre-impactor that removed particles larger than 10 µm. The APS had its own dryer that heated the sample air to 40 °C. This temperature might have evaporate some semivolatile compounds, for instance ammonium nitrate but this is mainly an issue of urban sites (e.g. Bergin et al. 1997), whereas at the forest site in Hyytiälä low-volatile organic compounds are common (Ehn et al., 2014). Nevertheless, semivolatile aerosol particles are typically secondary particles smaller than 1 µm in diameter so evaporation of them does not have a large effect on the APS measurements.

## 2.3 Data processing

All the optical data were quality assured manually and averaged for one hour. All the optical data were converted from ambient conditions to the standard temperature and pressure (STP) conditions (1013 hPa, 0 °C). We excluded the data from further analysis if the internal RH in any of the optical instruments exceeded 40 %, if not stated otherwise.

### 2.3.1 Corrections for the integrating nephelometer data

Both $\sigma_{sca}$ and $\sigma_{bsca}$ measured with the nephelometer were corrected for the truncation error according to Anderson and Ogren (1998). The truncation correction uses the Ångström exponent (see Sect. 2.4.1) calculated from the uncorrected data.

Sherman et al. (2015) presented a well documented analysis for determining the uncertainty of the different AOPs. They determined a total fractional uncertainty of 9.2 % and 8.9 % (8.0 % and 8.1 %) for PM10 (PM1) $\sigma_{sca}$ and $\sigma_{bsca}$.

Generally, in this study, the results are presented for dry aerosol (RH < 40 %), and therefore the hygroscopic growth had no notable effect on scattering. However, we had to take the effect of hygroscopic growth on scattering into account in two special cases: 1) to test if excluding the moist data (RH > 40 %) had a notable effect on the average AOPs and their trends; and 2) to calculate the aerosol radiative forcing efficiency (see Sect. 2.4.2) for ambient RH conditions. In these special cases, we used a scattering enhancement factor $f$(RH) to estimate the effect of RH on scattering. $f$(RH) describes the increase of $\sigma_{sca}$ with increasing RH

$$f(\text{RH}) = \frac{\sigma_{sca}(\text{RH})}{\sigma_{sca}(\text{RH= dry})}. \tag{1}$$

$f$(RH) is the ratio of $\sigma_{sca}$ measured at high RH and at dry conditions. The $f$(RH) can be described by the empirical relationship

$$f(\text{RH}) = q \left(1 - \frac{\text{RH}}{100\,\%}\right)^{-\gamma}, \tag{2}$$

with a parametrization presented by Andrews et al. (2006). They determined mean values for $q$ and $\gamma$ that were $0.84 \pm 0.10$ and $0.37 \pm 0.15$ for $\sigma_{\text{sca}}$ and $0.96 \pm 0.16$ and $0.12 \pm 0.15$ for $\sigma_{\text{bsca}}$. Andrews et al. (2006) presented the parameterization at 550 nm so we corrected only the $\sigma_{\text{sca}}$ and $\sigma_{\text{bsca}}$ at green wavelength. Andrews et al. (2006) used a four-year-long dataset measured

at the Southern Great Plains near Lamont, in Oklahoma, US. We decided to use this parametrization, since they provided the parametrization for both $\sigma_{\text{sca}}$ and $\sigma_{\text{bsca}}$. Also, the measurements were conducted for continental aerosol, which is closer to aerosols in a boreal forest than the mixtures of pollution, dust, sea salt, and volcanic aerosol, for which Carrico et al. (2003) presented the parametrization for both $\sigma_{\text{sca}}$ and $\sigma_{\text{bsca}}$. Zieger et al. (2015) determined scattering enhancement at SMEAR II in summer and obtained somewhat different values: $q = 1.01 \pm 0.05$ and $\gamma = 0.25 \pm 0.07$. This would have suited our needs very

well, but the $q$ and $\gamma$ were determined only for $\sigma_{\text{sca}}$ and not for $\sigma_{\text{bsca}}$, which is why we did not use this parametrization here.

In testing whether excluding the moist data has an effect on the AOPs, we performed the analysis also to a data set, where we included the periods of high humidity (RH > 40 %), but estimated the moist data (RH > 40 %) to dry conditions by using the $f(\text{RH})$. We used the $f(\text{RH})$ also in calculating the aerosol radiative forcing efficiency in ambient conditions, for which the dry

$\sigma_{\text{sca}}$ and $\sigma_{\text{bsca}}$ were converted to ambient RH.

### 2.3.2 Corrections for the Aethalometer data

The reported flow by the aethalometer was corrected by comparing the flow with the weekly flow measurements conducted at the station. The correction was applied by using a moving average of these measurements (see Sect. S3.1). An average spot

size diameter of $8.3 \pm 0.2$ mm was measured from the old aethalometer filters by using a loupe measuring scale magnifier (Eschenbach) with 0.1 mm accuracy and it was used instead of the spot size reported by the aethalometer.

We corrected the Aethalometer data by using the correction algorithm described by Collaud Coen et al. (2010)

$$\sigma_{\text{abs},i} = \frac{\sigma_{\text{ATN},i} - a_{s,i}\bar{\sigma}_{\text{sca},s,i}}{C_{\text{ref}}\,L_{s,i}}, \tag{3}$$

where

$$L_{s,i} = \left(\frac{1}{l(1-\bar{\omega}_{0,s,i})+1} - 1\right) \cdot \frac{\text{ATN}_i}{50\,\%} + 1, \tag{4}$$

and

$$a_{s,i} = \bar{\zeta}_{\text{sca},s,i}^{d-1} \cdot c \cdot \lambda^{-\bar{a}_{sca,s,i}\cdot(d-1)}. \tag{5}$$

In Eqs. (3)–(5), the subscript $i$ indicates the number of the measurement and the subscript $s$ indicates the average properties of

30 the aerosol particles that are embedded in the filter spot. The parameters with an overbar are the mean values from the start of

the filter spot to the $i$th measurement. In Eq. (3), the $\sigma_{\text{ATN}}$ is the attenuation coefficient reported by the Aethalometer, $a$ is the scattering correction parameter, $C_{\text{ref}}$ is the multiple scattering correction factor, and $L$ is the loading correction function. In Eq. (4), the $\omega_{\text{o}}$ is the single-scattering albedo (see Sect. 2.4.1) and the ATN is the light attenuation through the filter spot in percentages. In Eq. (5) the $\zeta_{\text{sca}}$ is the proportionality constant of the wavelength power law dependence of $\sigma_{\text{sca}}$ and $\alpha_{\text{sca}}$ is the

Ångström exponent of the $\sigma_{\text{sca}}$ (see Sect. 2.4.1). For $l$, $d$, and $c$ we used values 0.74, 0.564 and 0.329·10$^{-3}$ respectively. For scattering correction, we used measured $\sigma_{\text{sca}}$ values that were interpolated and extrapolated to the AE-31 wavelengths. Note that most of the symbols used for the variables are different from Collaud Coen et al. (2010). The reason is that in the present work the symbols are used for other variables below.

The $C_{\text{ref}}$ was determined by comparing the Aethalometer data, that was corrected only for the filter loading artefact, against the reference absorption coefficient ($\sigma_{\text{abs,ref}}$) measured by the MAAP

$$C_{\text{ref}} = \frac{\sigma_{\text{ATN}}}{L \cdot \sigma_{\text{abs,ref}}}. \tag{6}$$

The resulted median value for $C_{\text{ref}}$ was 3.19, with a standard deviation of 0.67.

The uncertainty of the $\sigma_{\text{ATN}}$ was determined according to Backman et al. (2017)

$$\frac{\delta\sigma_{\text{ATN}}}{\sigma_{\text{ATN}}} = \sqrt{f_{\text{A}}^2 + f_{\text{Q}}^2 + \left(\frac{\delta\sigma_{\text{ATN,zero}}\Delta t_{\text{zero}}}{\sigma_{\text{ATN}}\Delta t_{\text{avg}}}\right)^2}, \tag{7}$$

where the $f_{\text{A}}$ and $f_{\text{Q}}$ are the fractional uncertainties of the Aethalometer spot size and flow, which we determined to be 4.9 % and 1.5 % respectively; $\delta\sigma_{\text{ATN,zero}}$ is the standard deviation of the zero measurements; $\Delta t_{\text{zero}}$ is the averaging time of the zero measurements; and $\Delta t_{\text{avg}}$ is the averaging time of the measurements. For the uncertainty of $\sigma_{\text{abs}}$ we took into account the

fractional uncertainty of the $C_{\text{ref}}$, which was $f_{\text{C}} = 21$ %

$$\frac{\delta\sigma_{\text{abs}}}{\sigma_{\text{abs}}} = \sqrt{\left(\frac{\delta\sigma_{\text{ATN}}}{\sigma_{\text{ATN}}}\right)^2 + f_C^2}. \tag{8}$$

At 520 nm, the uncertainty of $\sigma_{\text{abs}}$ ranges from 22 % to 24 % if the value of $\sigma_{\text{ATN}}$ varies from 14.2 Mm$^{-1}$ to 1.3 Mm$^{-1}$, which are the 10$^{\text{th}}$ and 90$^{\text{th}}$ percentiles of $\sigma_{\text{ATN}}$. In this estimation of uncertainty, we did not take the uncertainty of scattering correction into account.

In calculating the single-scattering albedo (see Sect. 2.4.1) and in iterating the complex refractive index (see Sect. 2.4.3), the absorption data had to be interpolated to the same wavelength with the scattering measurements. The absorption data were then interpolated to the blue, green, and red wavelengths (450, 550, and 700 nm), using the Ångström exponent (α) described in Eqs. (11) and (12).

**2.4 Data analysis**

Here we describe the intensive optical properties, the aerosol radiative forcing efficiency, and parameters derived from the size distribution measurements.

**2.4.1 Intensive optical properties**

The extensive AOPs, $\sigma_{sca}$, $\sigma_{bsa}$, and $\sigma_{abs}$, were used to calculate intensive properties presented in detail below.

The single-scattering albedo ($\omega_0$) describes how much of the total light extinction (sum of $\sigma_{sca}$ and $\sigma_{abs}$) caused by the aerosol particles is due to scattering:

$$\omega_0 = \frac{\sigma_{sca}}{\sigma_{sca} + \sigma_{abs}}. \tag{9}$$

The $\omega_0$ can be linked to the source and chemical composition of the aerosol particles. High values of $\omega_0$ mean that the aerosol particles are mostly scattering. Particles that have a lower $\omega_0$, have a relatively higher mass fraction of absorbing material, such as soot, which is emitted in combustion processes.

15    The backscatter fraction ($b$) describes how much aerosol particles scatter radiation in the backward hemisphere compared with the total scattering

$$b = \frac{\sigma_{bsca}}{\sigma_{sca}}. \tag{10}$$

The angular dependency of particle scattering depends mostly on the particle size. The $b$ is smaller for a size distribution that consists of larger particles, since large particles scatter light heavily in the forward direction and thus $b$ can be used as an
20    indicator of the shape of the particle size distribution. The $b$ is an important variable for modeling the direct effect of aerosol particles on the climate, since it is used to describe how much sunlight is scattered upwards back into space.

The Ångström exponent ($\alpha$) is used to describe the wavelength ($\lambda$) dependency of a certain optical property ($\sigma$) (Ångström, 1929)

$$\alpha = -\frac{\ln\frac{\sigma_1}{\sigma_2}}{\ln\frac{\lambda_1}{\lambda_2}}. \tag{11}$$

After calculating $\alpha$, the optical property can be extrapolated or interpolated into different wavelengths

$$\sigma_1 = \sigma_2 \left(\frac{\lambda_1}{\lambda_2}\right)^{-\alpha}. \tag{12}$$

In this study, $\alpha$ values were calculated for $\sigma_{sca}$ and $\sigma_{abs}$ to obtain $\alpha_{sca}$ and $\alpha_{abs}$.

Scattering by aerosol particles dependend on the relation between the sizes of the particles and the wavelength of the radiation, therefore also $\alpha_{sca}$ is used as an indicator of the particle size distribution. The $\alpha_{sca}$ is larger for the smaller particles, since they have a stronger wavelength dependency. If the value of $\alpha_{sca}$ is larger than 2, the volume distribution is typically dominated by particles smaller than 0.5 µm, and the parameter $\alpha_{sca}$ is smaller than 1, larger particles (physical diameter $D_p > 0.5$ µm) predominate in the distribution (Schuster et al., 2006). In comparison to $b$, $\alpha_{sca}$ is more sensitive to the coarse mode particles (e.g. Collaud Coen et al., 2007). However, for multimodal size distributions this relationship is not quite unambiguous, as discussed by, for example, Schuster et al. (2006) and Virkkula et al. (2011).

The $\alpha_{abs}$ depends also on the chemical composition, coating, and size of the particles, even though the chemical composition is generally considered as a more important factor. The $\alpha_{abs}$ is usually used to identify black carbon (BC) and brown carbon (BrC) particles. The BC particles absorb radiation effectively at all wavelengths; the BrC particles, which consist of organic carbon compounds, absorb light strongly at shorter wavelengths, but not at longer wavelengths. If the particles consist purely of BC, the absorption would have a wavelength dependency of approximately $\lambda^{-1}$ and $\alpha_{abs}$ would be equal to unity. However, if the particles also consist of material that absorbs light only at ultraviolet wavelengths, $\alpha_{abs}$ would be larger than one. In ageing processes, the BC particles may become coated by some purely scattering material, such as sulfuric acid or ammonium sulfate, or by slightly absorbing organic material (Schnaiter et al., 2005; Zhang et al., 2008). The coating greatly affects the absorption wavelength dependency, and thus the division into BC and BrC by considering only the $\alpha_{abs}$ is not that simple. If the sizes of the BC particles and the thickness and complex refractive index ($m$) of the coating are not known, it is challenging to use $\alpha_{abs}$ to describe the chemical composition of the particles (Gyawali et al., 2009; Lack and Cappa, 2010). In spite the fact that the $\alpha_{abs}$ depends also on the coating, the absorption wavelength dependency is often used to describe the source of the BC (Sandradewi et al., 2008; Zotter et al., 2017). The source apportionment assumes that there are BC emissions only from traffic and wood burning and that the BC from these sources has a specific wavelength dependency.

The estimated uncertainties for the intensive AOPs are presented in Sect. S4 in the supplementary material. The uncertainties were calculated according to Sherman et al. (2015).

### 2.4.2 Aerosol radiative forcing efficiency

To investigate how the AOPs at SMEAR II would affect the climate, the aerosol radiative forcing efficiency ($\Delta F \delta^{-1}$ or RFE) was calculated. The RFE is a simplified formula that describes how large a difference the aerosol particles would make to the radiative forcing ($\Delta F$ or RF) per unit of aerosol optical depth ($\delta$) (Sheridan and Ogren, 1999)

$$\frac{\Delta F}{\delta} = -DS_o T_{at}^2 \omega_0 \beta (1 - A_c) \left[ (1 - R_s)^2 - \left( \frac{2R_s}{\beta} \right) \left( \frac{1}{\omega_0} - 1 \right) \right]. \tag{13}$$

RFE does not take into account that the properties and amount of aerosol particles vary vertically in the atmospheric column. In Eq. (13), $D$ is the fractional day length, $S_0$ the solar constant, $T_{at}$ the atmospheric transmission, $A_C$ the fractional cloud amount, and $R_S$ the surface reflectance for which the following global average values were used respectively: $D = 0.5$, $S_0 =$ 1370 W m$^{-2}$, $T_{at} = 0.76$, $A_C = 0.6$ and $R_S = 0.15$. The global averages do not represent the conditions at SMEAR II and the RFE calculated by using these constants is probably not realistic for the regional area. However, using the global averages makes it possible to compare the reported aerosol properties between different stations around the world. The values were chosen according to Haywood and Shine (1995), who used these values independent of wavelength in calculating the RF. Sheridan and Ogren (1999) used these same constants later in calculating the RFE at 550 nm and in this study we determine the RFE also at 550 nm. The factor $\beta$ is the upscatter fraction and is calculated by using the $b$ (Delene and Ogren, 2002)

$$\beta = 0.0817 + 1.8495b - 2.9682b^2. \tag{14}$$

It must be noted that Eq. (14) does not take into account the variation in the sun's zenith angle.

As stated by Sherman et al. (2015), the purpose of determining the RFE is to provide means for comparing the intrinsic aerosol forcing efficiency of aerosols measured at different sites. We calculated the RFE by using the same constant values to have results comparable with other studies in very different types of environments (e.g. Sheridan and Ogren, 1999; Andrews et al., 2011; Sherman et al., 2015; Shen et al., 2018) and to study how the RFE changes with varying $\omega_0$ and $b$. We refer the RFE that was calculated by using the above-mentioned constant values as RFE$_{H\&S}$. It must be noted that $\omega_0$ and $b$ used in Eq. (13) are defined for dried sample air and therefore RFE$_{H\&S}$ does not represent ambient conditions. In the ambient air, RH is larger and the AOPs are different due to hygroscopic growth.

In addition to RFE$_{H\&S}$, we calculated a seasonal RFE by allowing the $D$ to vary and by using more realistic seasonal values for $A_C$, and $R_S$. The seasonal variations of these parameters are presented in Fig. S1. We refer the seasonal RFE as RFE$_S$. The effect of ambient RH on $\omega_0$ and $b$, and hence to RFE, was also studied. The seasonal RFE calculated for ambient RH is referred as RFE$_{S,moist}$.

Seasonal $A_C$ was derived by using ceilometer data. The ceilometer was deployed at the Halli airport (about 25 km from SMEAR II) by Finnish Meteorological Institute (FMI) in 2010. The data were averaged for each month to get a seasonal variation. The lowest mean of $A_C$ occurred in July (~0.25) and the highest mean occurred in January (~0.76).

For the seasonal $R_S$, reflectivity determined by Kuusinen et al. (2012) was used. They determined the $R_S$ in a boreal forest for different amounts of canopy snow cover. According to the FMI, the average season of snow cover in Hyytiälä is from 16 November to 20 April (FMI: http://ilmatieteenlaitos.fi/lumitilastot, in Finnish only, last accessed: 13 March 2019) and for that time period we used $R_S = 0.314 \pm 0.14$ that Kuusinen et al. (2012) determined as the average albedo for a snow covered canopy.

For snow-free forest we used $R_S = 0.126$, which is an average of the mean monthly albedos Kuusinen et al. (2012) determined for snow-free months.

In calculating the $\omega_0$ and $b$ in ambient conditions, we used the equations and parametrization presented in Sect. 2.3.1 to convert the $\sigma_{sca}$ and $\sigma_{bsca}$ to ambient RH; $\sigma_{abs}$ was assumed to be constant with increasing RH, as Nessler et al. (2005) showed that the change in the $\sigma_{abs}$ with increasing RH is very small compared to scattering. The seasonally averaged RH was determined from RH measurements conducted at the height of 16 m. The lowest mean RH occurred in May (~62 %) and the highest in November (~95 %).

More information about the seasonal $D$, $A_C$, $R_S$, and RH can be found in the supplementary material Sect. S2.

### 2.4.3 Properties calculated from particle size distribution

Size distributions were used to calculate differently weighted mean diameters. In this study, we used the geometric mean diameter (GMD) and the volume mean diameter (VMD). The GMD is the mean diameter that is weighted by the number concentration ($N$)

$$\text{GMD} = \exp\left(\frac{\sum N_i \ln D_{p,i}}{\sum N_i}\right), \tag{15}$$

and the VMD is weighted by the particle volume ($V$)

$$\text{VMD} = \frac{\sum D_{p,i} V_i}{V_{tot}} = \frac{\sum N_i D_{p,i}^4}{\sum N_i D_{p,i}^3}. \tag{16}$$

Since the particle number concentrations are the highest for the nucleation and Aitken modes, the GMD describes the distribution changes in the smallest sizes. The VMD, in contrast, is affected by the changes in the accumulation and coarse mode, since they contribute the most to the volume size distribution.

The measurements of the AOPs and size distribution can be combined to determine the complex refractive index ($m = n + ik$) that describes how much the particles scatter and absorb light. The $m$ can be used to model $\sigma_{sca}$, $\sigma_{bsca}$ and $\sigma_{abs}$ from the size distribution. The $m$ consists of the real part ($n$), which accounts for the scattering, while the absorption is described by the imaginary part ($k$). Like $\omega_0$, $m$ provides information on the chemical composition of the aerosol particles.

In this study, $m$ was iterated from the $\sigma_{sca}$, $\sigma_{abs}$ and size distribution measurements in a manner similar to that described by Virkkula et al. (2011). In the first step of the interpolation $\sigma_{sca,Mie}$ and $\sigma_{abs,Mie}$, which are the modeled scattering and absorption coefficients, were determined for the measured size distribution by using the Mie-theory with initial $m = 1.544 + 0.019i$. The $\sigma_{sca,Mie}$ and $\sigma_{abs,Mie}$ were then compared with the measured $\sigma_{sca}$ and $\sigma_{abs}$. If the calculated and measured values did not agree, the

real part of $m$ was first varied stepwise by 0.001 until the measured and modeled $\sigma_{sca}$ agreed. Next, the imaginary part of $m$ was varied in the same way until the measured and modeled $\sigma_{abs}$ agreed. This iteration was continued until the measured and calculated values agreed within 1 %. The new imaginary part of $m$ also affected $\sigma_{sca}$ so the real part had to be reiterated. The MATLAB codes developed by (Mätzler, 2002) were used to model the Mie scattering and absorption.

### 2.4.4 Long-term trend analysis

Over the whole measurement period, 81 % of the nephelometer data and 70 % of the aethalometer data were considered valid. All of the AOPs had some gaps in the data (see Fig. 4). More detailed monthly data coverages of $\sigma_{sca}$ and $\sigma_{abs}$ are presented in Table S1. Most of the gaps in the time series of the AOPs during the summers of 2006 to 2010 were due to too high RH. If the
10  moist data was included, the overall data coverage would increase to 89 % and 77 % for scattering and absorption data, respectively. After the installation of the Nafion-dryers in March 2010, the humidity caused no further problems. The gap in 2010 was due to maintenance and installation of the dryers and the switching inlet system. Some additional $\sigma_{bsca}$ data were missing, due to malfunction of the backscatter shutter of the integrating nephelometer. Dirty optics, malfunctions and maintenance caused the gaps in the $\sigma_{abs}$ data in 2012 and 2015.

All the months that had at least 14 days of valid data were included in the long-term trend analysis. The trends and their significance were determined using the seasonal Kendall test described by Gilbert (1987). This test determines if there is a similar trend for each season (month) separately. All of the trends were calculated for the monthly medians, and at least 14 days of valid data in a given month were required for this month to be taken into account in the trend analysis.

### 3 Results and discussion

Below, we present the descriptive statistics of the AOPs, their seasonal variations and long-term trends at SMEAR II. The figures of the AOPs in this section are presented in the green wavelength (550 nm for the scattering and intensive properties and 520 nm for the absorption measurements). In the figures of $\alpha_{sca}$ and $\alpha_{abs}$, wavelength ranges of 450–700 nm and 370–950
25  nm were used. The results are presented for dry aerosols (RH < 40 %), if not stated otherwise.

### 3.1 Characterization of boreal aerosol particles

The descriptive statistics of the AOPs of both the PM10 and PM1 particles are shown in Tables 1 and 2. The statistics are calculated from hourly data. Tables 1 and 2 show that for most of the variables the mean and the median values were quite

different, which means that the data were not normally distributed. Therefore, we use the medians, which are not as sensitive to extreme values as the mean, to describe the characteristics of the AOPs.

5 The median values of PM10 $\sigma_{sca}$ and $\sigma_{abs}$ at SMEAR II at green wavelength were 9.8 Mm$^{-1}$ and 1.4 Mm$^{-1}$. In comparison to similar studies conducted at other Finnish measurement stations at Pallas in northern Finland (Lihavainen et al., 2015) and at Puijo tower in Kuopio, eastern Finland (Leskinen et al., 2012), results at SMEAR II showed the highest $\sigma_{sca}$ and $\sigma_{abs}$ for PM10 particles. At SMEAR II, the median value of $\sigma_{sca}$ was about two times higher and the value of $\sigma_{abs}$ more than three times higher than at Pallas, where the median $\sigma_{sca} = 4.4$ Mm$^{-1}$ and $\sigma_{abs} = 0.4$ Mm$^{-1}$ were measured at green wavelength. The Pallas station is

10 remote, located 170 km north of the Arctic Circle, far from anthropogenic sources, which explains the low concentrations. At SMEAR II, parameters $\sigma_{sca}$ and $\sigma_{abs}$ were about 1.4 and 1.1 times higher, than that measured at the Puijo tower, where the median values of $\sigma_{sca} = 7.2$ Mm$^{-1}$ and $\sigma_{abs} = 1.0$ Mm$^{-1}$ were measured at green and red wavelengths, respectively. Puijo tower is a semi-urban measurement station located only 2 km away from the Kuopio city center. At the Puijo tower, the measurements were conducted only on particles smaller than 2.5 µm, which explains part of the differences, at least for $\sigma_{sca}$.

Even though the $\sigma_{sca}$ measured at SMEAR II is high compared to other measurements conducted in Finland, the air measured at SMEAR II is still clean when compared to European sites. Due to the remote location, Pandolfi et al. (2018) observed low $\sigma_{sca}$ at SMEAR II compared to other European sites. Lower median $\sigma_{sca}$ were observed only in the arctic region, at another Nordic rural station in Birkenes, Norway, and at several high mountain sites. The highest median $\sigma_{sca}$ (> 40 Mm$^{-1}$) Pandolfi et

20 al. (2018) observed in urban and regional sites in central and Eastern Europe.

From Table 1 we see that the PM10 AOPs differ somewhat from the results of Virkkula et al. (2011) and Pandolfi et al. (2018) that can be explained by the trends and by differences in the data processing. For example the median value of $\sigma_{sca}$ (~10 Mm$^{-1}$) at $\lambda = 550$ nm in this study was lower than that presented by Virkkula et al. (2011) (~12 Mm$^{-1}$) and by Pandolfi et al. (2018)

25 (~11 Mm$^{-1}$), which is probably due to the tendency of $\sigma_{sca}$ to decrease (see Sect. 3.2). Another reason is that in the data processing Virkkula et al. (2011) used the earlier WMO/GAW recommendation (WMO/GAW, 2003) and used data measured at RH < 50% and did not do any RH corrections.

We also determined a strong tendency for $\sigma_{abs}$ to decrease as well and the median value of $\sigma_{abs}$ (~1.3 Mm$^{-1}$, interpolated to 550

30 nm) was somewhat lower than the median (~1.5 Mm$^{-1}$, at 550 nm) in the study by Virkkula et al. (2011). However, the observed $\sigma_{abs}$ between these two studies are not fully comparable due to the differences in the aethalometer data-processing. Virkkula et al. (2011) reported no flow or spot size corrections and they used the algorithm of Arnott et al. (2005) and $C_{ref} = 3.688$ at $\lambda = 520$ nm. Naturally, the different methods used in the absorption data processing also affected the optical properties that are dependent on the $\sigma_{abs}$, such as $\omega_0$ and $k$. In the correction algorithm by Arnott et al. (2005), the $C_{ref}$ is wavelength depended,

which increases the $\alpha_{abs}$. Virkkula et al. (2011) reported a median value of $\alpha_{abs} = 1.4$ that is notably higher than the median value of $\alpha_{abs} = 1.0$ determined by our study. The difference in parameter $\alpha_{abs}$ is attributed to the correction algorithm since also in the present work and median value of $\alpha_{abs} = 1.34$ for the wavelength range 370–950 when the Arnott et al. (2005) algorithm is used. (see Table S2).

The differences between the AOPs of the PM1 and PM10 particles are explained by the differences in concentrations, size distributions and chemical compositions. If only the PM10 data overlapping with the PM1 measurements were taken into account, the median values of PM10 $\sigma_{sca}$, $\sigma_{abs}$, $\omega_0$, $b$, $\alpha_{sca}$, $\alpha_{abs}$, $n$, and $k$ would have been 9.6 Mm$^{-1}$, 1.3 Mm$^{-1}$, 0.89, 0.14, 1.92, 0.97, 1.525 and 0.014 ($\sigma_{sca}$, $\omega_0$, $b$, $\alpha_{sca}$, $n$ and $k$ at 550 nm, $\sigma_{sca}$ at 520 nm). For PM1 the median values were 7.1 Mm$^{-1}$, 1.2 Mm$^{-1}$, 0.87, 0.15, 2.41, 1.03, 1.487 and 0.021, respectively. The extensive variables ($\sigma_{sca}$, $\sigma_{bsca}$ and $\sigma_{abs}$) were smaller for the PM1 measurements, since there was less particle volume interacting with the radiation. The $\alpha_{sca}$ and $b$ are related to the sizes of the particles, so they were naturally different between the PM1 and PM10 particles. For the smaller PM1 particles, the $\alpha_{sca}$ and $b$ were larger than for the PM10 particles. However, $b$ does not have as large a difference between the PM1 and PM10 particles as $\alpha_{sca}$.

On average submicron particles caused about 75 % of the total scattering of the PM10 particles. This was apparently a lower fraction than in the previous analysis of SMEAR II scattering data. Virkkula et al. (2011) stated that the average contributions of submicron particles to the total $\sigma_{sca}$ was in the range of 88–92 %, clearly more than in the present work. However, in that study the scattering size distribution and the contributions of the various size ranges were calculated from particle number size distributions with a Mie model and the physical diameters ($D_p$) were used whereas here the PM1 corresponds to particles smaller than the aerodynamic diameter $D_a$ of 1 µm. With particle density of 1.7 g cm$^{-3}$ this corresponds to the physical diameter $D_p = (1/1.7)^{\frac{1}{2}}$ 1 µm $\approx$ 0.77 µm. The contribution of particles smaller than 0.77 µm is approximately 85 % if it is estimated from Fig. 11 of Virkkula et al. (2011), still more than the ~75 % contribution of submicron scattering shown here. This may have resulted from the cutoff diameter of the PM1 impactor is not exactly sharp and also that the particles entering the impactor may have still been somewhat moist and thus larger than their dry size and were therefore removed from the sample stream. Further analysis of the difference is omitted here.

The PM1 particles absorbed about 90 % of the total PM10 particle absorption. So for the $\sigma_{abs}$ there were no large difference in the $\sigma_{abs}$ of the PM1 and PM10 particles. The coarse mode particles are typically primary and they have a quite high $\omega_0$ so their absorption is minor compared with the PM1 particles. The soot particles, which account for most of the particulate absorption, are typically submicron particles. Due to the relative differences in the scattering and absorption, the median $\omega_0$ and $n$ were lower for the PM1 particles than PM10 particles.

### 3.1.1 Seasonality of AOPs

The seasonal variation in the PM10 AOPs was clearly visible in the 11.5-year record shown in Fig. 1. The seasonal variations in $\sigma_{sca}$ and $\sigma_{bsca}$ (Figs. 1a and b) were not yet as clear in Virkkula et al. (2011) as it is now. For the $\sigma_{sca}$ and $\sigma_{bsca}$, two local maxima occurred during late winter (February) and late summer (July). The local minima occurred during spring (April) and late autumn (October). The $\sigma_{abs}$ showed the highest values during winter (February) and the lowest values during summer (June). Part of this variation is explained by boundary layer dynamics. In summer, the boundary layer is higher and well mixed, therefore diluting the aerosol concentration and in winter the situation is the opposite and the pollution accumulates in the shallow boundary layer. Also, the sources of aerosol particles vary seasonally, which affects the seasonal concentrations.

For the extensive properties, the highest values occurred at the same time in winter (February) when the $\omega_0$ was also low, which indicates that there were larger amounts of particles from anthropogenic sources than in summer. Hyvärinen et al. (2011) observed increased equivalent black carbon (eBC, meaning optically measured BC) concentration at SMEAR II in winter, when the long-range transport brings pollution from the central and Eastern Europe. However, Hienola et al. (2013) estimated that about 70 % of the measured eBC at SMEAR II is emitted from local or regional sources or transported from Finnish cities, so also the local and regional emissions have a significant role in the elevated eBC concentration. Since February is one of the coldest months in Finland, domestic wood burning in the local and regional area increases the particle concentration (Karvosenoja et al., 2011). Pollution can also be transported from nearby cities (the largest and closest are Tampere and Jyväskylä). Hyvärinen et al. (2011) observed no remarkable changes in the Hyytiälä eBC concentration coming from the Tampere region. However, the largest concentrations they observed came from the direction of Orivesi, a small town (population about 9 000) 20 km from the measurement station.

The $\alpha_{abs}$ is typically associated with the source of the BC and it is often used to quantify whether the BC is traffic or wood burning related (Sandradewi et al., 2008; Zotter et al., 2017) so that high $\alpha_{abs}$ is a sign of wood burning. In the source apportionment, $\alpha_{abs}$ close to one indicates that the BC is from traffic-related sources. Since we observed relatively higher $\alpha_{abs}$ in winter, the results are in line with the assumption of domestic wood burning that takes place during winter. However, in summer, $\alpha_{abs}$ was often < 1, which would yield an unphysical fraction (over a 100 %) of traffic related BC. Values below one could have been caused by large BC particles ($D_p > 100$ nm) that have a purely scattering coating (Lack and Cappa, 2010). It must be noted that the $\alpha_{abs}$ depends also on the correction algorithm. For example, if the $\sigma_{abs}$ was corrected with the algorithm proposed by Arnott et al. (2005), the median value of $\alpha_{abs}$ would have been $1.34 \pm 0.51$ (see Table S2). Using the $\alpha_{abs}$, which was determined by using the correction by Arnott et al. (2005), the results for the source apportionment would be different and they would show higher fraction of BC from wood burning.

The $\alpha_{abs}$ is also used to describe the chemical properties of the particles. Higher $\alpha_{abs}$ indicates that light is absorbed not only by BC, but also by some light-absorbing organic carbon compounds, i.e. brown carbon (BrC). In using only $\alpha_{abs}$, it is difficult to determine if the particles consist of BrC, since BC particles with coating can also reach $\alpha_{abs}$ values up to 1.6 (Lack and Cappa, 2010). In Fig. 1g we can see that the value of 1.6 is not really reached at SMEAR II if the correction algorithm by Collaud

Coen et al. (2010) was used. Since $\alpha_{abs}$ is dependent on the size of the BC core, the thickness of the coating and the $m$ of the coating, further investigation of its complex nature is omitted here.

In summer, the $\omega_0$ had its highest values since the parameter $\sigma_{sca}$ was high and the parameter $\sigma_{abs}$ was low. High $\sigma_{sca}$ but low

$\sigma_{abs}$ suggests that the anthropogenic influence was not strong in summer and that there was higher contribution of particles from natural sources when the vegetation was active and growing. The scattering maximum in summer was probably caused by secondary organic particles (Tunved et al., 2006).

The seasonal variation in $\alpha_{sca}$ and $b$ depends on the seasonal variation in the size distribution of the particles. Both $\alpha_{sca}$ and $b$

were maximal in summer and minimal in winter, suggesting that in summer, the particle population consisted of smaller particles than in winter. A trajectory analysis by Virkkula et al. (2011) showed that the highest $\alpha_{sca}$ were originated within a ~200 km radius around the station, which means that the smallest particles were rather freshly emitted. This supports the hypothesis that in summer a high fraction of the aerosols are secondary organic particles.

The impact of smaller particles in summer, indicated by the high $\alpha_{sca}$ and $b$, is seen also in Fig. 2a, which presents the seasonal variation of the $\sigma_{sca}$ PM1/PM10 ratio. The ratio describes the fraction of fine particles (PM1) on the PM10 $\sigma_{sca}$ and it shows that, in addition to summer, the fine particles have a high impact also in winter, which was not seen in $\alpha_{sca}$ and $b$ variation. Closer investigation on the seasonally averaged size distributions, which are presented in Fig. 3 (and S7), reveal that the seasonal variations of $\alpha_{sca}$ and $b$ are more depended on the shifts in the accumulation mode than in the coarse mode. Figure 3

shows that in winter, the $VMD_{tot}$ had its minimum due to a lack of coarse particles. This is in contrast with the observation of smaller $\alpha_{sca}$ and $b$, but it supports the maximum we see in the $\sigma_{sca}$ PM1/PM10 ratio. The seasonal variations of $\alpha_{sca}$ and $b$ were then explained by the seasonal variation of volumetric mean diameter calculated for particles smaller than 1 µm ($VMD_{fine}$). $VMD_{fine}$ is a good indicator for the shifting accumulation mode and it is not affected by the coarse mode. In winter, when the $\alpha_{sca}$ and $b$ were small, the accumulation mode was shifted towards larger sizes and the median of $VMD_{fine}$ was about 350 nm.

In summer, when the $\alpha_{sca}$ and $b$ had their maxima, the situation was the opposite and $VMD_{fine}$ was smaller, about 250 nm.

The seasonal variation in the size distribution did not affect the $\sigma_{abs}$ PM1/PM10 ratio and the median of the ratio did not vary seasonally, which is shown in Fig. 2b. However, the deviation of the $\sigma_{abs}$ PM1/PM10 ratio had a clear seasonal variation. In summer, the variation was considerably higher than in winter. In the correction algorithm, which was used for the absorption

data (Eq. (1)), part of the $\sigma_{sca}$ is subtracted from $\sigma_{abs}$ as an apparent absorption (Muller et al., 2011). The subtraction of $\sigma_{sca}$ causes relatively high uncertainty when the $\sigma_{abs}$ is low and $\sigma_{sca}$ is high, like it is in summer. This uncertainty is emphasized for PM10 measurements, since the $\sigma_{sca}$ is relatively higher than $\sigma_{abs}$, if compared to PM1 measurements. The uncertainty in the measurements also explains why many values were above one in the PM1/PM10 $\sigma_{abs}$ ratio.

## 3.2 Long-term trends of the AOPs

The about 11.5-year-long time series of the PM10 AOPs were used to determine long-term trends. For a comparison, we also conducted the trend analysis for the PM1 data, which had shorter, about 7.5-year-long, time series. It must be noted that trends for shorter time series are more sensitive to the year-to-year variability and must be interpreted with caution. The slopes of the

trends and the trend statistics are presented in Table 3. The table also presents the trends as percentages, which were calculated by dividing the slope by the overall median value of the variable. The trends are also plotted in Fig. 4, where the monthly medians of the PM10 AOPs at SMEAR II used in this analysis are presented. The monthly medians are included in Fig. 4 only if the month had at least 14 days of valid data.

In all the extensive properties, the trends were negative. The slopes of the trends for PM10 $\sigma_{sca}$, $\sigma_{bsca}$ and $\sigma_{abs}$ were -0.32, -0.038, and -0.086 Mm$^{-1}$ yr$^{-1}$, respectively. The decrease in the extensive properties were due to decrease in the total particle number concentration ($N_{tot}$) and total volume of the particles ($V_{tot}$), which time series are presented in Figs. 5a and 5b. The trend statistics for $N_{tot}$ and $V_{tot}$ are presented in Table 3. The $N_{tot}$ and $V_{tot}$ were determined by using the combined TDMPS and APS data for particles smaller than 10 µm in diameter. The relative decrease in $V_{tot}$ (-4 % yr$^{-1}$) was rather similar to that of $\sigma_{sca}$

(-3 % yr$^{-1}$). Also, Pandolfi et al. (2018) showed a statistically significant trend for $\sigma_{sca}$ (-0.588 Mm$^{-1}$ yr$^{-1}$) measured at SMEAR II. They reported negative trends at other European sites as well and they determined that the average decrease was about -35 % for a ten-year period, which is a bit larger reduction than that observed at SMEAR II (-30 % for a ten-year period). The results are in line with the decrease in particle number concentrations observed in European countries (Asmi et al., 2013). Also the remotely measured decreasing trend for aerosol optical depth ($\delta$) supports the decreasing trends in Europe (Li et al., 2014).

Decreasing trends for $\sigma_{sca}$ are not only observed in Europe; Collaud Coen et al. (2013) and Sherman et al. (2015) reported negative trends for $\sigma_{sca}$ in North America as well.

The observed relative decrease in $\sigma_{abs}$ (-6 % yr$^{-1}$) was about twice as large as that of $\sigma_{sca}$ (-3 % yr$^{-1}$). The differences in the trends indicates that during the measurement period, the amount of absorbing material, such as BC and BrC, decreased

relatively faster than the amount of scattering material (e.g. sulfate). It is also possible that the decrease in non-absorbing compounds decreased the $\sigma_{abs}$ since a non-absorbing coating around an absorbing particle can act as a lens, which increases absorption. The study by Collaud Coen et al. (2013), which included also $\sigma_{abs}$ data, observed negative trends for both $\sigma_{sca}$ and

$\sigma_{abs}$ at the Bondville measurement station in Illinois, USA, but there the trends of both $\sigma_{sca}$ and $\sigma_{abs}$ were about -3 % yr⁻¹. Sherman et al. (2015) did not observe this decreasing trend in $\sigma_{abs}$ later.

Since the aerosol particles were absorbing less light than in the beginning of the measurements, there was a tendency for the $\omega_0$ to increase. As shown by the increase in $\omega_0$ and the decrease in the extensive properties, the air measured at SMEAR II was less polluted than before. Higher $\omega_0$ indicates that the measurements were less affected by particles produced by traffic emissions or incomplete combustion. Li et al. (2014) reported mostly positive trends for $\omega_0$, which were determined by remote measurements conducted in Europe. The decreasing trend for $k$ supports the tendency for $\omega_0$ to increase, since the negative trend for the imaginary part of $m$ means that particulate matter absorbs less light. The $\alpha_{abs}$ and $n$, which are also related to the chemical composition of the particles, showed no significant trends.

The trend of $b$ and $\alpha_{sca}$ describe how the size distribution of the aerosol particles has changed. For the PM10 $b$ and $\alpha_{sca}$ the trends were positive, but for the $\alpha_{sca}$ however, the $p$ value was 0.07, so there was only a weak evidence for the positive trend in $\alpha_{sca}$. Increasing $b$ indicates that the size distribution moved towards smaller particles. This hypothesis was investigated by conducting the trend analysis for the volumetric and geometric mean diameters of particles smaller than 10 µm (VMD$_{tot}$ and GMD$_{tot}$), which were calculated from the size distribution. The results are presented in Table 3. We did not observe significant trend for the GMD$_{tot}$, which is sensitive to the smallest particles of the size distribution. However, a statistically significant trend was observed for VMD$_{tot}$, which time series are shown in Fig. 5c. The VMD$_{tot}$ trend statistics are presented in Table 3. The trend for VMD$_{tot}$ depends on the accumulation and coarse mode particles and therefore correlates better with the extensive AOPs than the GMD (Virkkula et al., 2011). Decreasing VMD$_{tot}$ indicates a shift in the size distribution towards smaller diameters supporting the increasing $b$ and $\alpha_{sca}$.

In addition to SMEAR II, Pandolfi et al. (2018) observed significant increasing trends for $b$ at several European stations. SMEAR II they observed significant increasing trend also for $\alpha_{sca}$, which was determined by using the wavelength range of 550–700 nm. At other sites, they observed mostly decreasing trends. Pandolfi et al. (2018) suspected that the variation was caused by differing trends of the coarse and accumulation mode particle concentrations. Also Li et al. (2014) observed negative trends for the $\alpha_{sca}$ across the Europe and they suggested the trends were caused by a decrease in fine particle emissions.

As a comparison, we also conducted the trend analysis for the PM1 measurements, even though there was only 7.5-years-long time series available. The trends observed for the PM1 particles were similar to those of PM10: decreasing trends for the extensive properties and increasing trends for the $\omega_0$ and $b$. For the PM1, the trends for both, $b$ and $\alpha_{sca}$, were positive and statistically significant. This observation suggests that especially the concentration of larger particles in the accumulation mode was decreasing, since a decrease in coarse particle concentration only could not cause the decreasing trend of PM1 $\alpha_{sca}$ nor $b$. A closer look in the size distribution, which is presented in Sect. S6, pointed out that relatively greatest decrease occurred for

accumulation mode particles that were 500–800 nm in diameter. On average, the volume size distribution of accumulation mode particles peaks around 300 nm (see Fig. 3) so the greatest decrease occurred at the larger sizes of the accumulation mode.

The decrease in the larger side of the accumulation mode might be caused by a decrease in long-range transported pollution. Aged pollution particles might be grown by other substances, such as $SO_2$ in the atmosphere so their sizes are larger than freshly emitted or formed particles. $SO_2$, emissions have decreased in Europe (Tørseth et al., 2012), which supports this assumption. A trajectory analysis by Virkkula et al. (2011) showed that $\alpha_{sca}$ was clearly higher in air masses from continental Europe than from the North Atlantic but also that the highest $\alpha_{sca}$ values were measured in air masses sources from within southern Finland, which would suggest that larger particles are not from nearby the station.

The installation of the Nafion-dryers in 2010 could have caused an artificial decrease in the AOPs since the dryers increase the deposition of the particles and may decrease the sizes of hygroscopic particles. However, the similar trend between the PM1 and PM10 AOPs does not support this suspicion, since during the PM1 measurements, there were no large changes in the measurement line, so the observed trends were probably not caused by any technical issues in the measurement line.

In addition to the general trends, we also investigated how the trends of $\sigma_{sca}$ and $\sigma_{abs}$ varied between the seasons. In this analysis, the periods of RH > 40 % were included in order to avoid the data gaps in summer and autumn before 2010 and to have time series with equal lengths for each season. The moist $\sigma_{sca}$ was estimated to dry conditions according to Eqs. (1) and (2). The trends were determined separately for spring (March, April, May), summer (June, July, August), autumn (September, October, November), and winter (December, January, February). The trend calculations were conducted by using the monthly medians (see timeseries in Fig. S3) and the results are presented in Table 4.

Table 4 shows that $\sigma_{sca}$ and $\sigma_{abs}$ had a decreasing trend for each season, but in the autumn the trends were not significant. Both $\sigma_{sca}$ and $\sigma_{abs}$ experience the fastest absolute decrease in winter when the energy consumption is the highest and pollution sources are more pronounced; on the opposite, the trends were the least negative in summer when there is less pollution. In spring, the absolute trends were less negative than compared to winter.

### 3.3 Seasonality and long-term trend of radiative forcing efficiency

To study the climate impact of the aerosol particles, the aerosol radiative forcing efficiency (RFE) the average values, trends, and seasonal variations were investigated. We determined three different kind of RFE: 1) $RFE_{H\&S}$ was calculated to dry aerosol particles by using global average values suggested by Haywood and Shine (1995), 2) $RFE_S$ was calculated also to dry aerosols but here we used more realistic environmental parameters ($D$, $R_S$, and $A_C$) at SMEAR II and here also the seasonality of the parameters was taken into account, and 3) $RFE_{S,moist}$ also used the more realistic and seasonally varying environmental

parameters but here we took into account effect of ambient RH on $\omega_0$ and $b$. The statistics of the $RFE_{H\&S}$, $RFE_S$ and $RFE_{S,moist}$ are presented in Table 1 and their time series and seasonal variations are presented in Fig. 6.

In general, the aerosols measured at SMEAR II tended to have a cooling effect on the climate ($RFE < 0$) as seen in Table 1.
By using the global average values suggested by Haywood and Shine (1995), the mean value of $RFE_{H\&S}$ was -22 W m$^{-2}$. This was about 12 % less negative than the mean value of $RFE_{H\&S}$ (about -25 W m$^{-2}$) determined by Sherman et al. (2015) for different North American stations. The difference is explained by higher mean value of $\omega_0$ (about 0.91) observed by Sherman et al. (2015). The mean value of $b$ (about 0.14) was similar if compared to average values observed at SMEAR II. A mean value of $RFE_{H\&S}$ -25 W m$^{-2}$ was determined also at SORPES station in Nanjing, China (Shen et al., 2018). Shen et al. (2018) observed a lower mean value of $b$ (0.12 at 520 nm), which would increase the $RFE_{H\&S}$. However, they also observed a notably higher mean value of $\omega_0$ (0.93 at 520 nm) at SORPES and that overcame the effect of lower parameter $b$.

If the seasonal variation of $D$, $A_C$, and $R_S$ were taken into account, the RFE became more negative at SMEAR II. The median value of $RFE_S$, which takes the seasonality of the environmental parameters into account, was -26 W m$^{-2}$, lower than the median value of $RFE_{H\&S}$, which was -23 W m$^{-2}$. This was mainly due to the higher $D$, and lower $R_S$ and $A_C$ in summer. If the ambient RH was taken into account, the median value for the $RFE_{S,moist}$ (-24 W m$^{-2}$) increased a bit compared to $RFE_S$. Taking the ambient RH (that was RH > 40 % every month) into account increases the $\omega_0$ due to increase in the scattering. At the same time the $b$ decreases since the particles grow in size and scatter relatively less light backwards (Birmili et al., 2009). These two changes have opposite effects on the RFE: increasing $\omega_0$ decreases the RFE, and decreasing $b$ increases the RFE. Here the decreasing $b$ overcomes the effect of $\omega_0$ and therefore the median $RFE_{S,moist}$ is higher than that of $RFE_S$.

The long-term trends of $\omega_0$ and $b$ tended to increase, which makes the $RFE_{H\&S}$ to decrease (i.e., become more negative). The decreasing $RFE_{H\&S}$ means that the properties of dry aerosol particles have changed so that they cool the climate more efficiently. The trends for the $RFE_{H\&S}$, $RFE_S$ and $RFE_{S,moist}$ are presented in Table 3 as well. Since we used seasonal averages in calculating the $RFE_S$ and $RFE_{S,moist}$, their trends were also depended only on the changes of the $\omega_0$ and $b$ and thus their trends are also decreasing and similar in magnitude as the trend for $RFE_{H\&S}$. However, in reality the trend of $RFE_S$ and $RFE_{S,moist}$, which take into account the realistic environmental parameters, do not depend only on the $\omega_0$ and $b$. For example, a decrease in the snow cover due to global warming would decrease the $R_S$ and make the decrease of $RFE_S$ and $RFE_{S,moist}$ steeper. Here, we omitted further analysis on the effect that the changes of $A_C$, $R_S$, $T_{at}$ and RH have on the $RFE_S$ and $RFE_{S,moist}$.

The seasonal variation in the $RFE_{H\&S}$ followed the seasonal cycles of the $\omega_0$ and $b$. The $RFE_{H\&S}$ was minimal in summer and maximal in winter. Since $b$ was lowest (forward-scattering particles) and the $\omega_0$ is also low (absorbing particles) in winter, the particles clearly did not have as strong a cooling effect as in summer when particles were smaller and highly scattering. If the seasonal changes of $D$, $A_C$, and $R_S$, were taken into account, the seasonal variability of $RFE_S$ amplified remarkably as seen in

Fig. 7b. In winter, the higher $R_S$ causes the aerosol particles to be less cooling or even warming, but since the $D$ is low and the $A_C$ high (see Fig. S1), the aerosol particles to have less effect (RFE closer to zero) than in summer. We chose to use the $R_S$ determined for a boreal forest according to the surroundings of SMEAR II. However, the area around the station consists also of fields and lakes, which in winter would act as smooth snow fields. Even for snow containing impurities the $R_S$ is notably

higher ($> 0.7$) than $R_s$ for snow covered boreal forest (Warren and Wiscombe, 1980). Using $R_S = 0.7$ for winter time data, would increase the $\text{RFE}_S$ and amplify its seasonal variation even more.

Taking the effect of ambient RH into account decreases the seasonality in $\text{RFE}_S$ a bit. The seasonality of RH is presented in Fig. S1d and on average the RH is higher in winter than in summer. Figure 6b shows that that compared to $\text{RFE}_S$, the $\text{RFE}_{S,\text{moist}}$

is less negative in summer when the effect of RH on $b$ overcomes the effect on $\omega_0$. Also Fierz-Schmidhauser et al. (2010) observed this kind of behavior at the Jungfraujoch station. In winter, the situation was actually the opposite and the $\text{RFE}_{S,\text{moist}}$ was more negative than $\text{RFE}_S$. However the difference was small, partly due to the low $D$ and high $A_C$. In general, the effect of the RH on the seasonal variation of RFE was smaller than the effect of taking the seasonal variation of $D$, $A_C$, and $R_S$ into account.

The RFE (or $\Delta F \delta^{-1}$) describes only the efficiency of the aerosol particles in cooling or warming the climate per unit of aerosol optical depth ($\delta$). Even if the RFE was very negative, the influence of aerosol particles on the climate would be small if the $\delta$ was small. Equation (13) assumes that the properties of aerosol particles are uniform in the atmospheric column that is rarely the case in reality. In ambient air, we should also take into account the variability in RH as a function of height, since at the

top of the boundary layer we typically have RH values close to 100 %. Here, we determined the RFE by using the RH measured near the ground (16 m). The simplified RFE does not give an absolute value for the aerosol forcing; however, it can still indicate how the changes in AOPs affect the climate.

**3.4 Effect of excluding the moist data**

Only about 62 % of AOP data measured before 2010, was marked as valid. A big fraction of the not-valid data was invalidated due to too high RH. If we took the moist data into account and estimated it to dry conditions, the data coverage from June 2006 to December 2010 increases to 87 %. To test, if excluding large amount of data had a significant difference to the results, we used Eqs. (1) and (2) to estimate the moist (RH > 40 %) $\sigma_{\text{sca}}$ and $\sigma_{\text{bsca}}$ to dry conditions and included this data in calculating the median values and in the trend analysis. Also, moist (RH > 40 %) $\sigma_{\text{abs}}$ measurements were included here.

If the moist periods of $\sigma_{\text{sca}}$ and $\sigma_{\text{abs}}$ measurements were included in the analysis and the moist scattering data were estimated to dry conditions by using the parametrization suggested by Andrews et al. (2006), we would get median values of $\sigma_{\text{sca}} = 10.4$ $\text{Mm}^{-1}$, $\sigma_{\text{bsca}} = 1.5 \ \text{Mm}^{-1}$, $\sigma_{\text{abs}} = 1.5 \ \text{Mm}^{-1}$, $\omega_0 = 0.88$, $b = 0.14$, $\alpha_{\text{abs}} = 0.97$, and $\text{RFE}_{\text{H\&S}} = -23 \ \text{W m}^{-2}$ for PM10 ($\sigma_{\text{sca},} \sigma_{\text{bsca},} \omega_0, b,$

and RFE$_{\text{H&S}}$ at 550 nm, $\sigma_{\text{abs}}$ at 520 nm, $\alpha_{\text{abs}}$ at 370 nm/950). Taking the moist samples into account, the $\sigma_{\text{sca}}$ and $\sigma_{\text{abs}}$ increased in summer and autumn (see Fig. S8). We could not determine the $\alpha_{\text{sca}}$, since we only converted the $\sigma_{\text{sca}}$ at 550 nm to dry conditions. If we used the parameters observed by Zieger et al. (2015), the median $\sigma_{\text{sca}} = 10.3$ Mm$^{-1}$, which is very close to the value obtained by using the parameters suggested by Andrews et al. (2006). For the extensive properties, including also the

moist data increased their median values about 7 % and for the intensive properties there were no notable effect. Omitting the moist data periods from the data set does not seem have a large effect on the median AOPs in this data set.

Including the originally omitted data in the trend analysis, we observed statistically significant ($p$-value $< 0.05$) trends for the PM10 $\sigma_{\text{sca}}$, $\sigma_{\text{bsca}}$, $\sigma_{\text{abs}}$, $\omega_0$, and RFE with the slopes of -4 % yr$^{-1}$, -5 % yr$^{-1}$, -5 % yr$^{-1}$, 0.2 % yr$^{-1}$, and 0.6 % yr$^{-1}$, respectively.

Still, there were decreasing trends for extensive properties and positive trends for $\omega_0$. For $b$, there was no significant trend anymore due to. The difference between the $\sigma_{\text{sca}}$ and $\sigma_{\text{abs}}$ trends decreased from 3 % yr$^{-1}$ to 1 % yr$^{-1}$ if compared against the trends that were determined only for the dry conditions. Including the moist data and acquiring longer data sets in the trend analysis suggests that the relative difference between the trends of $\sigma_{\text{sca}}$ and $\sigma_{\text{abs}}$ might not be that large.

**4 Summary and conclusions**

In this study, we presented 11.5-year-long time series of AOPs measured at SMEAR II, a station in southern Finland. Compared to regional and rural European sites, the $\sigma_{\text{sca}}$ at the boreal SMEAR II station was low. However, the average $\sigma_{\text{sca}}$ and $\sigma_{\text{abs}}$ were higher than those observed at other Finnish measurement stations that were the arctic station in Pallas and the semi-urban station in Kuopio, Eastern Finland. Because of the more southern location, the SMEAR II was probably more affected by regional emissions and long-transport pollution from Europe than the other Finnish measurement sites, which would explain

the higher concentrations.

The highest $\sigma_{\text{sca}}$ and $\sigma_{\text{abs}}$ were measured in winter when the boundary layer is lower and the pollution is not diluted as efficiently as in summer. Transported pollution from the regional area and from Europe, also increases the concentration in winter, when the energy consumption is higher. In winter, the $\omega_0$ was low (i.e. absorption was relatively high compared to scattering), which

also indicates that there was a higher fraction of particles from anthropogenic combustion sources. The $\sigma_{\text{sca}}$ had high values also in summer but the $\sigma_{\text{abs}}$ had its minimum and therefore the $\omega_0$ reached it maximum in summer. This observation indicates that the particle concentration was high in summer due to active vegetation.

Closer investigation on the size distribution revealed that the seasonal variations of $b$ and $\alpha_{\text{sca}}$ were caused by shifting

accumulation mode and not by changes in the coarse mode particle concentration. In summer, $b$ and $\alpha_{\text{sca}}$ had their maxima (i.e. there was a higher fraction of smaller accumulation mode particles); and in winter, they had their minima (i.e. there was a higher fraction of large accumulation mode particles).

The extensive AOPs, as well as the aerosol number and volume concentrations, tended to decrease. Our observations were in line with the other studies conducted in Europe and North America, which also observed decreasing trends for the extensive AOPs (Collaud Coen et al., 2013; Pandolfi et al., 2018; Sherman et al., 2015), number concentration (Asmi et al., 2013) and aerosol optical depth (Li et al., 2014). This uniform decreasing trend in the amount of aerosol particles suggests that the anthropogenic emissions have been decreasing in Europe and North America. The observed tendency for $b$ and $\alpha_{sca}$ to increase together with the decreasing extensive properties indicated that the particle size distribution consisted of less larger particles. A more detailed investigation revealed that the number of larger accumulation mode particles (500–800 nm in diameter) decreased relatively the fastest, which would indicate a decrease in transported anthropogenic pollution

Since the aerosol particles were scattering light more efficiently to backward hemisphere and because they absorbed less light than in the beginning of the measurements, their RFE tended to decrease (i.e. became more negative), which means that the ability of aerosols to cool the climate per unit $\delta$ increased. However, since the extensive properties and particle number concentration were decreasing, which means that the $\delta$ decreased as well, the total aerosol forcing was probably also decreasing. Here, we only studied the effect of AOPs on the RFE. Studying and taking also the long-term trends of the environmental parameters into account would give more realistic trend for the RFE.

*Data availability* All the data presented in this study is open access. The optical properties and the size distribution data from SMEAR II has been uploaded on the EBAS database (EBAS: http://ebas.nilu.no/, last access: 18 March 2019) run by the Norwegian Institute for Air Research (NILU). Meteorological parameters measured at SMEAR II, such as the RH used here, can be accessed by the Smart-SMEAR online tool (Junninen et al., 2009). Also the Finnish Meteorological Institute provides open access data and we used their online data tool (FMI: https://ilmatieteenlaitos.fi/havaintojen-lataus, last access: 18 March 2019) to access the ceilometer data measured at Halli airport.

*Author contribution* Krista Luoma did the data analysis and wrote the manuscript together with Aki Virkkula. Pasi Aalto and Aki Virkkula have set up the long term measurements of AOPs at SMEAR II. All authors reviewed and commented on the manuscript.

*Acknowledgements* The research leading to these results has received funding from the European Union Seventh Framework Programme under grant agreement No 262254. This research has also received funding from the European Union's Horizon 2020 research and innovation programme under grant agreement No 654109 via project ACTRIS-2 and grant agreement No 689443 via project iCUPE. The work was also funded by the Academy of Finland (project. No 307331).

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

**Table 1: Descriptive statistics of the AOPs for the PM10 particles. The average values were calculated from all valid data.**

| PM10 | $\lambda$ (nm) | mean ± SD | 1 % | 10 % | 25 % | 50 % | 75 % | 90 % | 99 % |
|---|---|---|---|---|---|---|---|---|---|
| $\sigma_{sca}$ (Mm$^{-1}$) | 450 | 21.8 ± 23.3 | 1.8 | 4.5 | 7.6 | 14.2 | 26.8 | 48.5 | 114.1 |
| | 550 | 15.2 ± 16.7 | 1.3 | 3.4 | 5.5 | 9.8 | 18.3 | 33.4 | 82.5 |
| | 700 | 9.5 ± 10.5 | 0.8 | 2.3 | 3.7 | 6.3 | 11.3 | 20.3 | 52.3 |
| $\sigma_{bsca}$ (Mm$^{-1}$) | 450 | 2.5 ± 2.9 | 0.2 | 0.6 | 1.0 | 1.8 | 3.2 | 5.3 | 11.1 |
| | 550 | 2.0 ± 1.8 | 0.2 | 0.5 | 0.8 | 1.4 | 2.5 | 4.2 | 8.8 |
| | 700 | 1.6 ± 1.5 | 0.2 | 0.4 | 0.7 | 1.2 | 2.0 | 3.4 | 7.4 |
| $\sigma_{abs}$ (Mm$^{-1}$) | 370 | 3.0 ± 3.6 | 0.2 | 0.6 | 1.0 | 1.9 | 3.6 | 6.6 | 18.1 |
| | 470 | 2.5 ± 2.9 | 0.2 | 0.5 | 0.8 | 1.6 | 3.0 | 5.4 | 14.3 |
| | 520 | 2.2 ± 2.4 | 0.1 | 0.4 | 0.7 | 1.4 | 2.6 | 4.7 | 12.3 |
| | 590 | 1.9 ± 2.2 | 0.1 | 0.4 | 0.7 | 1.3 | 2.4 | 4.2 | 10.8 |
| | 660 | 1.8 ± 2.0 | 0.1 | 0.3 | 0.6 | 1.2 | 2.2 | 3.8 | 9.9 |
| | 880 | 1.3 ± 1.4 | 0.1 | 0.3 | 0.5 | 0.9 | 1.6 | 2.9 | 7.2 |
| | 950 | 1.2 ± 1.3 | 0.1 | 0.3 | 0.4 | 0.8 | 1.5 | 2.6 | 6.5 |
| $\omega_0$ | 450 | 0.88 ± 0.07 | 0.64 | 0.80 | 0.85 | 0.89 | 0.93 | 0.95 | 0.98 |
| | 550 | 0.87 ± 0.07 | 0.62 | 0.78 | 0.84 | 0.88 | 0.92 | 0.94 | 0.98 |
| | 700 | 0.84 ± 0.08 | 0.55 | 0.74 | 0.80 | 0.85 | 0.90 | 0.93 | 0.97 |
| $b$ | 450 | 0.13 ± 0.03 | 0.08 | 0.10 | 0.11 | 0.12 | 0.14 | 0.16 | 0.21 |
| | 550 | 0.14 ± 0.03 | 0.09 | 0.11 | 0.13 | 0.14 | 0.16 | 0.17 | 0.21 |
| | 700 | 0.19 ± 0.07 | 0.07 | 0.13 | 0.15 | 0.18 | 0.21 | 0.25 | 0.44 |
| $\alpha_{sca}$ | 450/550 | 1.73 ± 0.52 | 0.23 | 1.03 | 1.49 | 1.82 | 2.09 | 2.29 | 2.58 |
| | 450/700 | 1.80 ± 0.55 | 0.32 | 1.00 | 1.53 | 1.88 | 2.17 | 2.39 | 2.80 |
| | 550/700 | 1.85 ± 0.64 | 0.23 | 0.95 | 1.53 | 1.95 | 2.26 | 2.50 | 3.15 |
| $\alpha_{abs}$ | 370/520 | 0.95 ± 0. 48 | -0.29 | 0.51 | 0.76 | 0.98 | 1.16 | 1.32 | 1.97 |
| | 370/950 | 0.95 ± 0.36 | -0.16 | 0.55 | 0.80 | 0.99 | 1.13 | 1.24 | 1.69 |
| | 470/660 | 0.95 ± 0.49 | -0.52 | 0.52 | 0.80 | 1.01 | 1.15 | 1.29 | 2.07 |
| | 470/950 | 0.99 ± 0.41 | -0.32 | 0.58 | 0.86 | 1.06 | 1.18 | 1.28 | 1.83 |
| | 660/950 | 1.02 ± 0.57 | -0.77 | 0.57 | 0.90 | 1.11 | 1.23 | 1.34 | 2.17 |
| $n$ | 450 | 1.541 ± 0.065 | 1.330 | 1.478 | 1.512 | 1.542 | 1.572 | 1.607 | 1.697 |
| | 550 | 1.518 ± 0.067 | 1.289 | 1.452 | 1.490 | 1.522 | 1.550 | 1.581 | 1.674 |
| | 700 | 1.491 ± 0.091 | 1.247 | 1.379 | 1.454 | 1.501 | 1.536 | 1.574 | 1.740 |
| $k$ | 450 | 0.021 ± 0.020 | 0.002 | 0.006 | 0.009 | 0.016 | 0.026 | 0.039 | 0.097 |
| | 550 | 0.020 ± 0.018 | 0.002 | 0.006 | 0.010 | 0.016 | 0.025 | 0.038 | 0.089 |
| | 700 | 0.022 ± 0.019 | 0.003 | 0.007 | 0.011 | 0.018 | 0.027 | 0.041 | 0.092 |
| RFE$_{H\&S}$ (W m$^{-2}$) | 550 | -22 ± 6 | -32 | -28 | -26 | -23 | -19 | -16 | -3 |
| RFE$_S$ (W m$^{-2}$) | 550 | -35 ± 32 | -97 | -82 | -67 | -26 | -5 | 0 | 12 |
| RFE$_{S,moist}$ (W m$^{-2}$) | 550 | -33 ± 28 | -88 | -74 | -62 | -24 | -5 | -2 | 4 |

**Table 2: Descriptive statistics of the AOPs for the PM1 particles. The average values were calculated from all valid data; therefore if compared with the PM10 average values, there is a four-year shorter dataset.**

| PM1 | $\lambda$ (nm) | mean ± SD | 1 % | 10 % | 25 % | 50 % | 75 % | 90 % | 99 % |
|---|---|---|---|---|---|---|---|---|---|
| $\sigma_{sca}$ (Mm⁻¹) | 450 | 17.7 ± 19.2 | 1.2 | 3.1 | 5.6 | 11.3 | 22.3 | 40.4 | 96.1 |
| | 550 | 11.4 ± 13.0 | 0.8 | 2.1 | 3.6 | 7.1 | 14.1 | 26.1 | 64.8 |
| | 700 | 6.3 ± 7.5 | 0.4 | 1.2 | 2.0 | 3.8 | 7.6 | 14.4 | 37.4 |
| $\sigma_{bsca}$ (Mm⁻¹) | 450 | 2.1 ± 2.0 | 0.2 | 0.4 | 0.8 | 1.4 | 2.7 | 4.5 | 9.7 |
| | 550 | 1.6 ± 1.5 | 0.1 | 0.3 | 0.6 | 1.1 | 2.0 | 3.4 | 7.5 |
| | 700 | 1.2 ± 1.2 | 0.1 | 0.3 | 0.5 | 0.8 | 1.5 | 2.6 | 5.9 |
| $\sigma_{abs}$ (Mm⁻¹) | 370 | 2.4 ± 2.9 | 0.1 | 0.4 | 0.8 | 1.6 | 2.9 | 5.2 | 15.0 |
| | 470 | 2.0 ± 2.3 | 0.1 | 0.4 | 0.7 | 1.3 | 2.4 | 4.3 | 11.7 |
| | 520 | 1.7 ± 1.9 | 0.1 | 0.3 | 0.6 | 1.2 | 2.1 | 3.7 | 10.0 |
| | 590 | 1.6 ± 1.7 | 0.1 | 0.3 | 0.5 | 1.0 | 1.9 | 3.3 | 8.8 |
| | 660 | 1.4 ± 1.6 | 0.1 | 0.3 | 0.5 | 01.0 | 1.8 | 3.1 | 8.0 |
| | 880 | 1.1 ± 1.1 | 0.1 | 0.2 | 0.4 | 0.7 | 1.3 | 2.3 | 5.8 |
| | 950 | 0.9 ± 1.0 | 0.0 | 0.2 | 0.3 | 0.6 | 1.2 | 2.0 | 5.1 |
| $\omega_0$ | 450 | 0.88 ± 0.08 | 0.62 | 0.78 | 0.84 | 0.89 | 0.93 | 0.96 | 0.98 |
| | 550 | 0.85 ± 0.08 | 0.59 | 0.75 | 0.81 | 0.87 | 0.91 | 0.94 | 0.98 |
| | 700 | 0.80 ± 0.10 | 0.48 | 0.67 | 0.75 | 0.81 | 0.87 | 0.91 | 0.96 |
| $b$ | 450 | 0.13 ± 0.03 | 0.07 | 0.10 | 0.11 | 0.13 | 0.14 | 0.17 | 0.23 |
| | 550 | 0.15 ± 0.03 | 0.09 | 0.12 | 0.13 | 0.15 | 0.17 | 0.19 | 0.24 |
| | 700 | 0.23 ± 0.13 | -0.06 | 0.14 | 0.17 | 0.21 | 0.26 | 0.34 | 0.78 |
| $\alpha_{sca}$ | 450/700 | 2.22 ± 0.44 | 0.88 | 1.70 | 1.99 | 2.28 | 2.51 | 2.66 | 2.95 |
| | 450/550 | 2.36 ± 0.55 | 0.74 | 1.76 | 2.09 | 2.41 | 2.66 | 2.87 | 3.70 |
| | 550/700 | 2.48 ± 0.81 | 0.25 | 1.73 | 2.16 | 2.52 | 2.82 | 3.13 | 4.69 |
| $\alpha_{abs}$ | 370/520 | 0.96 ± 0.61 | -0.67 | 0.47 | 0.74 | 0.99 | 1.20 | 1.39 | 2.32 |
| | 370/950 | 0.97 ± 0.44 | -0.36 | 0.52 | 0.80 | 1.03 | 1.19 | 1.33 | 1.96 |
| | 470/660 | 0.94 ± 0.66 | -0.94 | 0.46 | 0.76 | 1.00 | 1.17 | 1.33 | 2.35 |
| | 470/950 | 1.03 ± 0.51 | -0.51 | 0.56 | 0.87 | 1.11 | 1.25 | 1.39 | 2.24 |
| | 660/950 | 1.13 ± 0.72 | -1.10 | 0.60 | 0.97 | 1.20 | 1.35 | 1.54 | 2.96 |
| $n$ | 450 | 1.509 ± 0.057 | 1.348 | 1.441 | 1.478 | 1.513 | 1.542 | 1.568 | 1.634 |
| | 550 | 1.484 ± 0.054 | 1.338 | 1.422 | 1.456 | 1.487 | 1.516 | 1.540 | 1.598 |
| | 700 | 1.471 ± 0.074 | 1.294 | 1.393 | 1.435 | 1.472 | 1.505 | 1.537 | 1.677 |
| $k$ | 450 | 0.025 ± 0.020 | 0.003 | 0.008 | 0.013 | 0.020 | 0.031 | 0.045 | 0.099 |
| | 550 | 0.025 ± 0.018 | 0.004 | 0.009 | 0.014 | 0.021 | 0.031 | 0.044 | 0.093 |
| | 700 | 0.028 ± 0.019 | 0.005 | 0.011 | 0.017 | 0.024 | 0.035 | 0.049 | 0.098 |

**Table 3: Slopes of the trends (in absolute values and in estimated percentages per year) and their statistical significance. The lower and upper limits in the 95 % confidence interval for different optical properties are also shown. The trend in the percentage was determined by comparing the slope of the trend with the overall median of the data.**

| | $\lambda$ (nm) | PM10 Trend (yr$^{-1}$) | | Lower (yr$^{-1}$) | Upper (yr$^{-1}$) | p-value | PM1 Trend (yr$^{-1}$) | | Lower (yr$^{-1}$) | Upper (yr$^{-1}$) | p-value |
|---|---|---|---|---|---|---|---|---|---|---|---|
| $\sigma_{sca}$ (Mm$^{-1}$) | 550 | -0.32 | -3 % | -0.52 | -0.17 | < 0.01 | -0.30 | -4 % | -0.55 | -0.12 | < 0.01 |
| $\sigma_{bsca}$ (Mm$^{-1}$) | 550 | -0.038 | -3 % | -0.070 | -0.021 | < 0.01 | -0.051 | -5 % | -0.087 | -0.013 | < 0.01 |
| $\sigma_{abs}$ (Mm$^{-1}$) | 520 | -0.086 | -6 % | -0.133 | -0.044 | < 0.01 | -0.141 | -12 % | -0.166 | -0.098 | < 0.01 |
| $\omega_0$ | 550 | 2.2e-3 | 0.3 % | 0.7e-3 | 3.6e-3 | < 0.01 | 5.5e-3 | 0.6 % | 1.5e-3 | 10e-3 | < 0.01 |
| $b$ | 550 | 1.3e-3 | 0.9 % | 0.9e-3 | 1.7e-3 | < 0.01 | 1.5e-3 | 1 % | 0.7e-3 | 2.6e-3 | < 0.01 |
| $\alpha_{sca}$ | 450/700 | 0.012 | 0.7 % | -0.001 | 0.024 | 0.07 | 0.014 | 0.6 % | 0.004 | 0.024 | < 0.01 |
| $\alpha_{abs}$ | 370/950 | -1.5e-4 | 0 % | -3.0e-3 | 2.9e-5 | 0.95 | -3.5e-3 | -0.3 % | -7.9e-3 | 13e-3 | 0.34 |
| $n$ | 550 | -2.0e-3 | -0 % | -3.8e-3 | 0.6e-3 | 0.11 | -5.7e-3 | -0.4 % | -7.5e-3 | -2.9e-3 | < 0.01 |
| $k$ | 550 | -6.6e-4 | -4 % | -9.1e-4 | -3.8e-4 | < 0.01 | -1.3e-3 | -6 % | -2.0e-3 | -0.7e-3 | < 0.01 |
| RFE$_{H\&S}$ (W m$^{-2}$) | 550 | -0.30 | -1 % | -0.43 | -0.20 | < 0.01 | | | | | |
| RFE$_S$ (W m$^{-2}$) | 550 | -0.43 | -2 % | -0.64 | -0.25 | < 0.01 | | | | | |
| RFE$_{S,moist}$ (W m$^{-2}$) | 550 | -0.37 | -2 % | -0.50 | -23 | < 0.01 | | | | | |
| $N_{tot}$ (cm$^{-3}$) | | -40 | -3 % | -52 | -28 | < 0.01 | | | | | |
| $V_{tot}$ (µg cm$^{-3}$) | | -0.093 | -4 % | -0.120 | -0.064 | < 0.01 | | | | | |
| GMD$_{tot}$ (nm) | | -0.092 | -0 % | -0.531 | 0.342 | 0.63 | | | | | |
| VMD$_{tot}$ (nm) | | -12 | -1 % | -17 | -7 | < 0.01 | | | | | |

**Table 4: Slopes of the seasonal trends and their statistical significance for $\sigma_{sca}$ and $\sigma_{abs}$. The trend in the percentage was determined by comparing the slope of the trend with the seasonal median of the data.**

| | $\sigma_{sca}$ (Mm$^{-1}$) | | | | | $\sigma_{abs}$ (Mm$^{-1}$) | | | | |
|---|---|---|---|---|---|---|---|---|---|---|
| | Trend (yr$^{-1}$) | | Lower (yr$^{-1}$) | Upper (yr$^{-1}$) | *p*-value | Trend (yr$^{-1}$) | | Lower (yr$^{-1}$) | Upper (yr$^{-1}$) | *p*-value |
| **Spring** | -0.44 | -5 % | -0.84 | -0.04 | < 0.05 | -0.12 | -9 % | -0.20 | -0.05 | < 0.01 |
| **Summer** | -0.38 | -3 % | -0.79 | -0.14 | < 0.01 | -0.06 | -5 % | -0.11 | -0.03 | < 0.01 |
| **Autumn** | -0.12 | -1 % | -0.49 | 0.17 | 0.48 | -0.04 | -3 % | -0.10 | 0.03 | 0.14 |
| **Winter** | -0.85 | -7 % | -1.60 | -0.20 | < 0.01 | -0.17 | -8 % | -0.31 | -0.03 | < 0.05 |

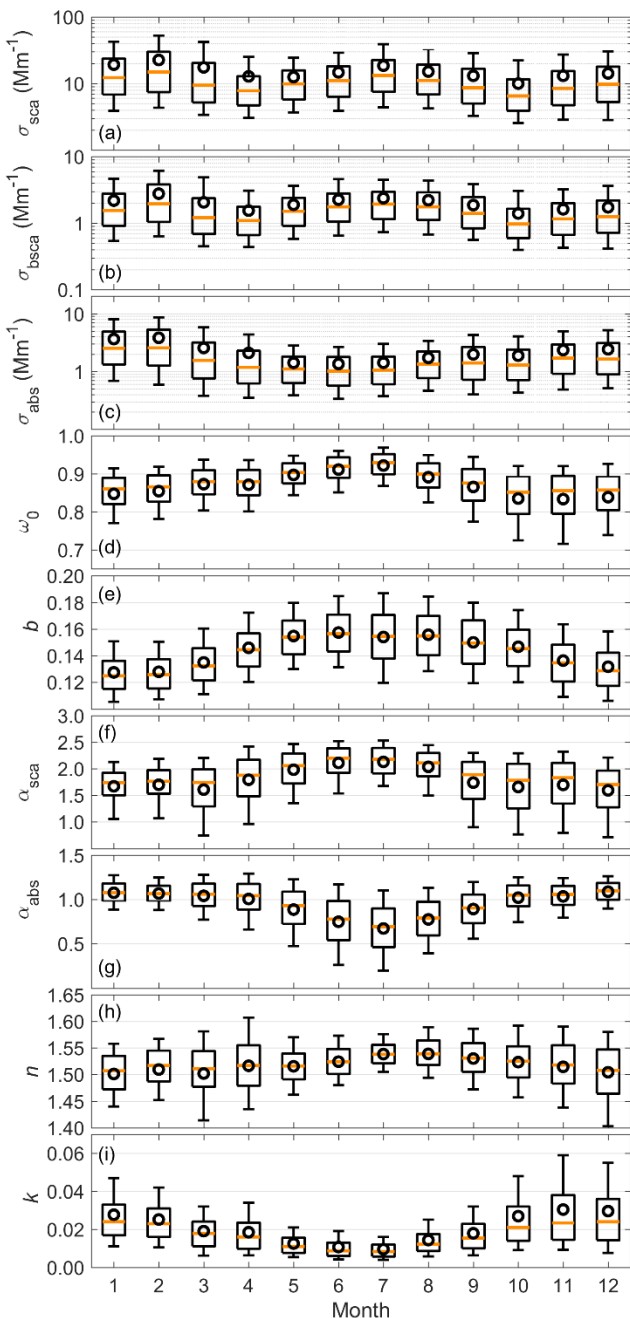

**Figure 1: Seasonal variation in the aerosol optical properties for PM10 particles. The boxes represent the 25th and 75th percentiles and the whiskers the 10th and 90th percentiles of the data. The orange line is the median and the mean is presented with a black circle.**

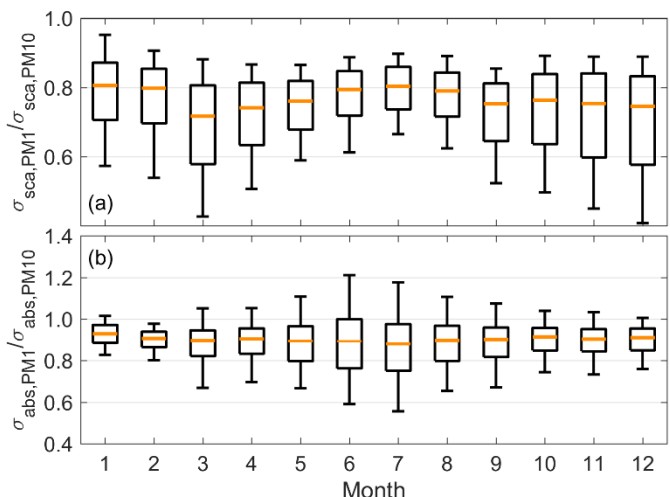

Figure 2: Seasonal variation in the PM1/PM10 ratio for a) $\sigma_{sca}$ and b) $\sigma_{abs}$. The explanation for the boxplots are the same as in Fig. 1.

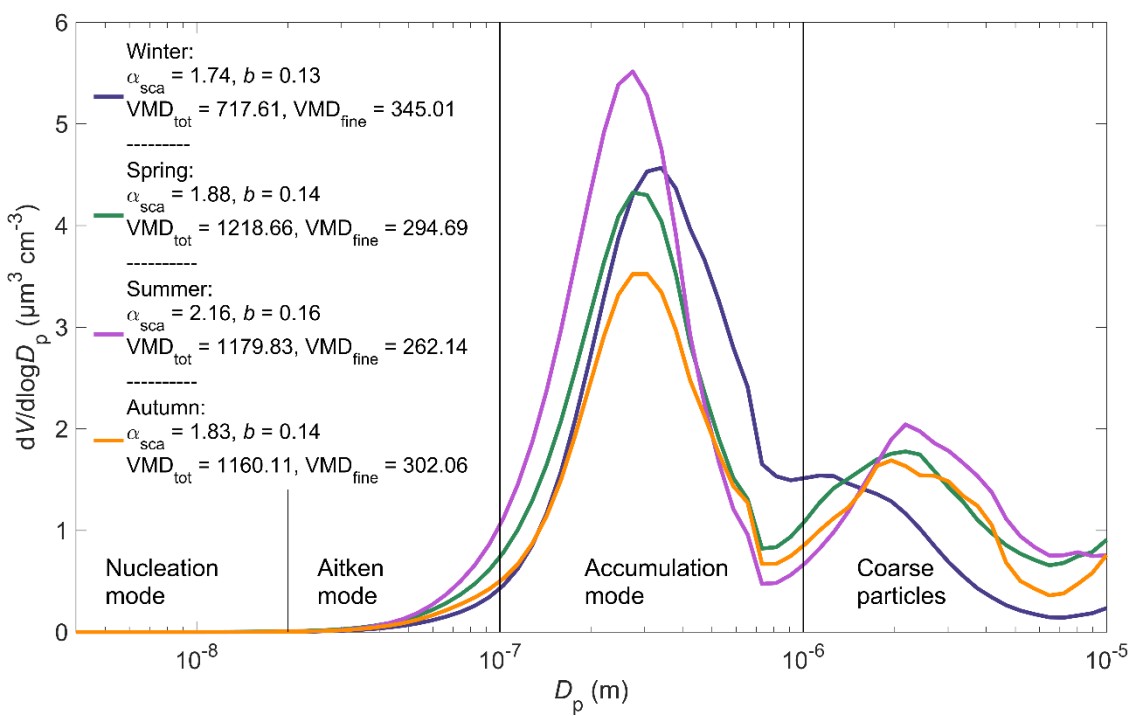

5 Figure 3: Mean size distribution for winter (December–February), spring (March–May), summer (June–August), and autumn (September–November). Also, the average $\alpha_{sca}$, $b$, $VMD_{tot}$, and $VMD_{fine}$ for the seasons are presented in the figure. $VMD_{tot}$ and $VMD_{fine}$ are reported in nm.

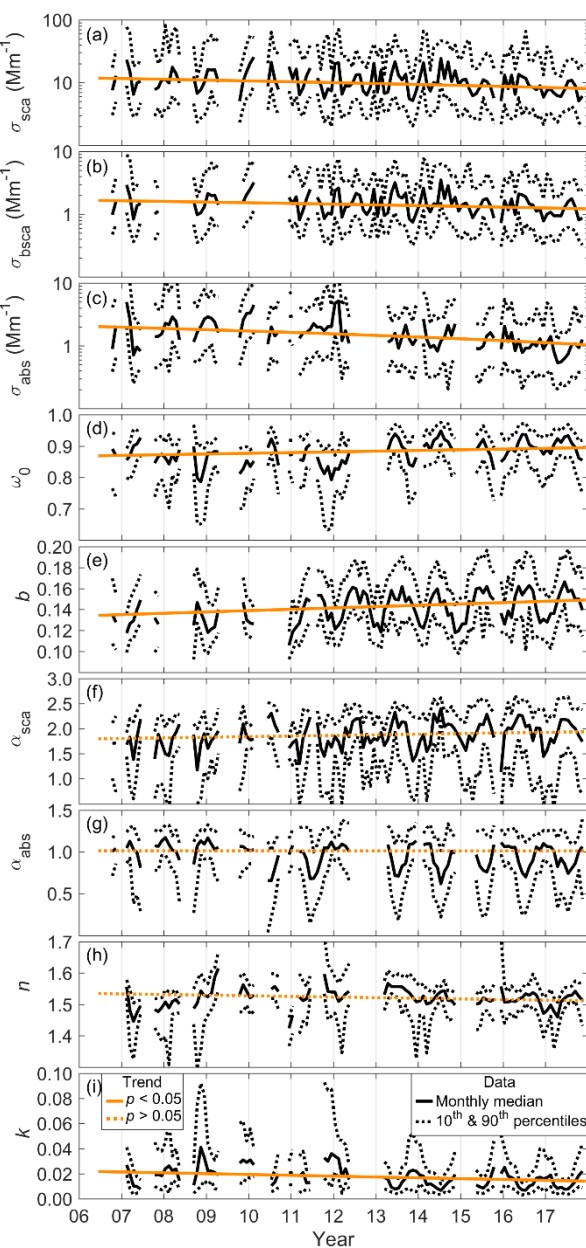

**Figure 4: Time series of the PM10 AOPs. The uniform black line presents the monthly median and the dotted black lines present the monthly 10th and 90th percentiles. The trends (see Table 3) of the AOPs are shown with orange lines. If the trend was statistically significant, the line is uniform and if the *p* value of the trend was > 0.05, the line is dashed.**

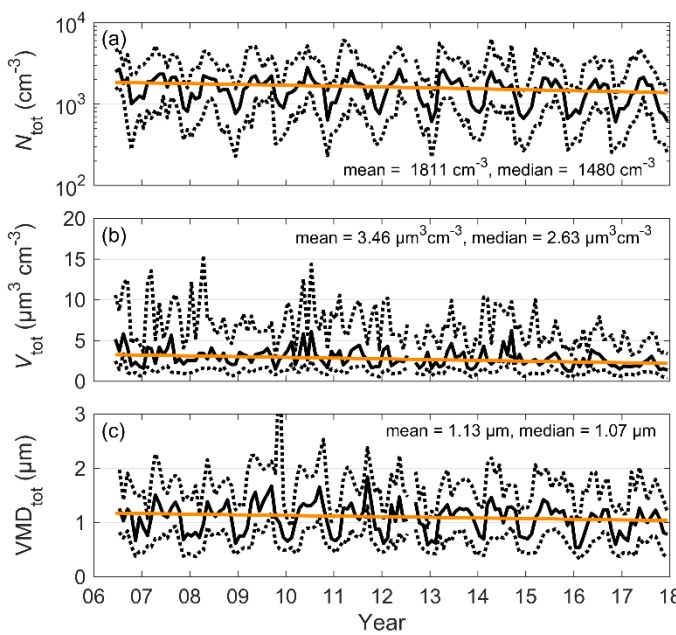

**Figure 5: Time series and trends of the total particle ($D_p$ < 10 µm) a) number concentration ($N_{tot}$), b) volume ($V_{tot}$) and c) VMD$_{tot}$. The mean and median values of the variables are also marked in the subfigures and the statistics of their slopes are presented in Table 3. The explanations for the different lines are the same as in Fig. 5.**

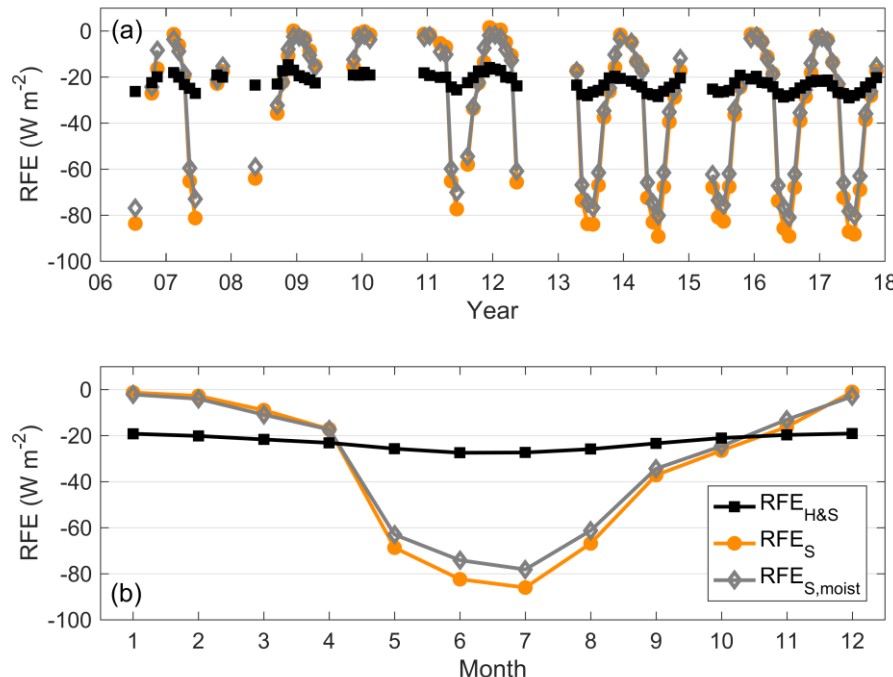

**Figure 6: Variations in the different radiative forcing efficiencies at SMEAR II in 2006–2018. a) Time series of the RFE$_{H\&S}$, RFE$_S$, and RFE$_{S,moist}$. The monthly medians are presented if the month had at least 14 days of valid data. b) Seasonal variation of the RFE$_{H\&S}$, RFE$_S$, and RFE$_{S,moist}$ as overall monthly medians. RFE was calculated for PM10 particles.**