# Peer review of "A ten-year record of aerosol optical properties at SMEAR II – Supplementary material"

_Atmospheric Chemistry and Physics, 2018_

## Referee Comment (RC1) · Anonymous Referee #1 · 4 Dec 2018

GENERAL COMMENT

The here presented manuscript describes the variability of several aerosol optical properties (AOPs) measured in southern Finland for more than a decade. The multi-year variation of AOPs is presented together with a detailed analysis of AOPs variability on a shorter timescale. Due to its time coverage, the dataset presented here is of great relevance and might help to understand how aerosol changed in a North European back-ground site during the last ten years. The scheme and structure of the manuscript are linear and follows a logical order. However, the amplitude of the dataset generates a certain overloading of the manuscript, meaning that the results are not always properly discussed within a climatologic perspective but simply described. As a consequence, is difficult to identify the overall scientific message of the work. I truly

believe that the paper covers the topic of interest of ACP, but I would recommend the authors to improve the discussion and interpretation of their results in order to better transmits their message to the readers. Hopefully, the major and specific comments reported below will help the authors to improve their work.

MAJOR COMMENTS

I have the strong feeling that the manuscript is overloaded with figures, especially multi-panels figures. First of all, due to the similarity between PM1 and PM10 (Figure 1, 2, 4), the discussion and presentation of results become particularly redundant in Section 3.2, 3.3, 3.4. The subsequent effect is that the discussion often focusses on the differences between the two aerosol fractions rather than on the reasons leading to the multi-year trends or seasonality. I thus suggest the authors show and describe PM10. This will lighten the paper and give more space for the climatologic interpretation of the results. Moreover, it appears that a considerable number of figures is poorly described or is not essential to the understanding of the results. I thus suggest the authors to reconsider the absolute relevance of certain graphs and to remove them from the manuscript or move them to the supplementary. More details can be found in the specific comments.

The dataset allows the investigation of multi-year variability and trends of AOPs and size distribution. The variability of AOPs is also investigated on a shorter time resolution but ignoring the year-to-year variability (Section 3.4, 3.5, 3.6). Thanks to the long-term measurement I would expect a work focusing on trends and multiple-year variability of AOPs. However, the analysis of trends is disconnected from the seasonal and diurnal variability and the consequent RFE. Therefore, I have some troubles in understanding what is the topic or scientific question acting as a glue between the sections, which in some cases (Section 3.3.1, 3.5 and 3.6) appear to be self-standing. I would thus suggest the authors to better exploit their long time series and focus on the long-term evolution/variability of AOPs including trends, impacts on seasons and, potentially, diurnal variability. For instance, Section 3.3.1 is based on 2 months measurements only, what is the long-term implications of NPF on the aerosol optical properties, and is this short period representative of the 10 considered years? Moreover, Section 3.5 provides the diurnal variability of AOPs. Despite the fact of a weak variability, was the boundary layer dynamic changing within the 10-year period? It is hard to understand the relevance and implications of such variability. Similar reasoning applies to the monthly variability, did summer and winter experienced a change from 2006 to now?

The calculation of the forcing efficiency is an extremely interesting topic, and up to me a decadal trend of RFE might represent the core of the entire manuscript together with Section 3.2 and 3.3. However, for the RFE estimations, the authors assumed the atmospheric (RH, cloud) and environmental (surface albedo, day length) variables as constant, which are not even specific for the SMEAR II station. Due to strong seasonality and, potentially, year-to-year variability of such variables, the final RFE estimations are unrealistic. I would suggest the authors implement constants representative, at least, of Southern Finland or, better, to use seasonal dependent variables. Other than that, any conclusion on climatic impacts of aerosol at SMEAR II will be highly questionable and of low interest.

SPECIFIC COMMENTS

P3L11: why MAAP and PSASP are introduced if only data from the AE31 are used?

P3L13: this is irrelevant for the present manuscript.

P3L23: I would say that, since Luoma et al. 20xx is not available, a better description should be provided here.

P3L31: can the author exclude the influence of hygroscopic growth?

P4L4: which instruments, all of them?

P4L11: is not clear why the truncation correction was not applied for the back-scattering. Does it mean that back-scattering can be affected by systematic error compared to total scattering? Was this assessed? Was it negligible? The authors present more than 10 years of data, more care in the presentation of the data correction is mandatory.

P4L19: multiple correction procedures were used or only the Collaud Coen et al. (2010) as stated later? If the correction of Collaud Coen was used, I honestly do not see the reason to cite all the other algorithms. Generally, I would not recommend the frequent self-citation of works that are not ready yet.

P6L15: I expect that BC from biomass burning and traffic has a different chemical composition. Isn't it in contrast with lines 13-15? If AlphaAbs is simultaneously affected by size, chemical composition, sources and mixing, to what purpose is AlphaAbs used here?

P6L22: Equation 5 is quite different from Haywood and Shine (1995), is this the original source of the equation? What is the wavelength of RFE? Is then the aerosol optical depth measured or everything is calculated from Equation 5? Though I have quite some doubts on the choices of constants (see comments on Section 3.7), a better description of the equations and its limits should be provided, together with the motivations at the base of the choices of the constants and the subsequent uncertainties.

P10L5-17: this part of the section mostly describes the technical aspects of the measurements. I would suggest to move them in the method section. Potentially into a new subsection discussing the data coverage and how the data set was reduced/validated.

P11L3: this is the first and last time $\omega 0$ was discussed in Section 3.3. I am wondering if the four panels in Figure 4 showing $\omega 0$ are needed at all.

P11L21-24: the inverse proportionality between ïĄąsca and GMD is supposedly caused by the bimodal size distribution of the aerosol and the substantial presence of accumulation particles. Despite supported by a reference, there is no direct explanation of the physical causes behind such proportionality. Since this is contrary to

expectations, as stated by the authors, a deeper reasoning and explanation should be provided.

P12L7-10: The diameter of the particles is the driver for both AlphaSca and b, I have some difficulties in understanding the relevance of the findings described here.

P13L2-3: here is stated that long-range transport brings pollution to the station, but 70% of black carbon comes from local and regional sources (P13L7). These two statements are contradictory.

P13L4-5: I am not sure to understand the relevance of the polar dome here.

P13L30-33: you have the size distribution data, why should you make a hypothesis on size distribution from optical properties?

P15L15-18: Here you need to be careful with the instrumental error. Do you mean that absorption was close to the detection limit of the instrument or that dominant presence of non-absorbing particles caused a decrease of light transmitted through the filter and apparent absorption (Müller et al., 2011)?

P15L23-24: the RFE trends are not described, discussed or interpreted. This recalls my major comments. The manuscript is loaded with data that are never discussed. Provide an interpretation or remove Fig.10a. By the way, add to all figures the panel reference.

P15L26-28: since RFE is calculated from b and w0 and all the environmental variables are kept constant, RFE must change with b and w0. As follow up to the second major comment, the authors are required to provide a deeper interpretation of their results.

P15L29: how the monthly RFE should be interpreted if the atmospheric and environmental parameters are kept constant? Moreover, it appears that the constants are not representative of SMEAR II. So, what should we really learn out of RFE?

P16L5-8: the problem here is that the aerosol optical depth is affected by RH and

subsequent hygroscopic growth. So, all your RFE are systematically underestimated by an unknow factor. However, it is unclear if optical depth is measured or calculated.

P16L9-10: Nessler et al. (2005) suggested that water uptake does not enhance absorption coefficient of BC.

P16L11-13: from this work, it is impossible to quantify the change of radiative forcing, nor the effects on the climate. First, RFE trends are not discussed: Second, the absolute values of RFE, as admitted by the authors, are far from being realistic. Moreover, why should we use RFE as "an indicator of how the properties of the aerosol particles have been changing" if the changes of aerosol particles have been measured (Section 3.2, 3.3, 3.4)?

P17L1-2: Fig 1 and Fig. 2 show a net decrease of aerosol number concentration, but is this due to the implementation of new emission policies only? How did precipitation and air circulation changes from 2006? I would recommend the authors to consider all possibilities and base their final conclusions on their data and existing literature.

F1: this figure is too crowded, I do not think that showing both PM1 and PM10 as any relevance (see major comments).

F2: here 5 panels are used to show that the total number of particles decreases and the size distribution is shifted to the smaller diameters. I would say that two panels will do efficiently the job. For example, one panel showing the total particle number concentration (Nfine+Ncoarse) and a second panel showing the ratio between Nfine and Ncoarse or the GMD. Note that Nfine is never defined in the text, is this accumulation+Aitken+nucleation? Please provide a description.

F1-2 As a follow up of my previous comments, I would find a way to merge together a reduced version of Figure 1 and 2, with the goal to focus on the relationship between physical and optical properties described in the text.

F3 This Figure is mentioned only once at P9L26, it does not appear to provide a key

insight into the understanding of data interpretation. I would thus recommend to move it to the supplementary.

F5: the size distribution of PM10 contains all the necessary data to investigate the size distribution in PM1. This is clear in panels (c) and (d), where the size distributions below 1um are exactly the same. This recalls my general comments, is a separated discussion of PM1 and PM10 really necessary?

F6: The figure shows the nucleation events and the related change in the real part of the refractive index. However, I think that it is largely overcrowded. On page 12, lines 26-27 sufficiently describe the absence of change in the observed AOPs. Due to the low relevance of AOPs variability in this context, I would suggest removing the third and fourth panels from the top. Finally, I am wondering what is the relevance of 2 months data over a 10 year period.

---

## Referee Comment (RC2) · Anonymous Referee #2 · 17 Dec 2018

GENERAL COMMENTS

This manuscript reports on the optical and microphysical properties of aerosols collected for over a decade at the SMEAR-II atmospheric monitoring station in Finland. These data are valuable in determining long-term trends and variabilities in aerosol properties, which are useful to climate modelers. The statistical distributions of aerosol properties presented in this paper should also be useful to GCM and CTM modelers for model initialization and validation exercises. As such, I think this paper is appropriate for inclusion in ACP and warrants publication after attention to the comments listed below.

The paper is well organized and there are only a few places were the English usage could be improved. The methodology used by the authors is excellent and of high

[Figure]

quality, and the data presented are in general valid and relevant. I do have some comments that suggest some additional thought be given to provide better explanations of the observations, and I think a better discussion of how drying the sampled aerosols might influence the RFE results is warranted (see comments below). The paper is a bit long but the amount of data being presented from over a decade at this site and the necessary discussions warrant a longer paper. In looking for possible ways to decrease the length of the paper, the only thing I see is to remove the size distribution discussion. While it is interesting in its own right and assists in the interpretation of the AOP data, it is not strictly necessary in this paper. I will leave that decision up to the authors and the editor.

In order to better interpret trends and variability, some estimates of the measurement uncertainty should be provided. I would point the authors to the work of Sherman et al. (2015, ACP), who put a great deal of effort into estimating measurement uncertainties for aerosol optical properties. There is no need to repeat this exercise in detail, but at the very least this reference should be included and some mention of the measurement uncertainties for the TSI nephelometer should be provided.

It was a bit disappointing to find little or no discussion in sections 3.1-3.3 on the relevance and importance of the measurements and their long-term trends and variability. A considerable amount of discussion is presented in sections 3.4-3.7 to explain the seasonal and diurnal variability, etc., and I would like to see more of this in sections 3.2 and 3.3. For the trends, for example, it would be useful to know how these trends compared with other long-term trends in Europe. The Pandolfi paper is cited and is an excellent place to start. Some additional information can be found in Collaud-Coen et al. (2013, ACP), and this paper should also be cited when comparing the optical property (scattering and absorption) measurements. There is also little discussion on the importance of measuring the optical properties in two different size ranges (PM1 and PM10). What does it tell you about sources, ageing, human contribution, etc., if the PM1 fraction for a given parameter is almost as large as the PM10 fraction? What

does it mean if the PM1/PM10 ratio is changing over time? The authors went to the trouble of adding this additional set of PM1 measurements in 2010 and have several tables and graphs in this paper showing the results. They need to say why they are important and what we learn from them.

The purpose of the lengthy discussion of the comparison of the optical and microphysical properties on page 11 is not clear to me. The manuscript title indicates that this manuscript is about the aerosol optical properties, so why are there size distribution data included in the results and discussions? They are of course useful for interpreting the optical properties, so they have value, and good agreement between measured optical properties and ones calculated from the microphysical measurements give increased confidence in the findings of the study. Perhaps the authors can state that more clearly. The size distribution results could also go in the supplemental materials section if length of the paper becomes a concern.

The RFE calculations in this paper use the global average constants of Haywood and Shine rather than ones estimated or derived for the local area. This is probably OK for trend analysis but the magnitude of the forcing is wrong, especially when considering seasonal variations. For example, the constant used for the global average surface albedo (0.15) does not represent that of the boreal forest around SMEAR-II station over all seasons. . . it should be significantly higher in winter due to snow cover and (I would guess) lower in summer. Also, the measurement relative humidity (RH) and the ambient RH were generally different (this occurred most frequently in the summers). The authors state that if the sample RH was above 40%, the data were flagged and marked as invalid. This implies that the SSA and b values are only accurate when the ambient RH was low (i.e., close to the measurement RH), and that the RFE results are only appropriate for times when the ambient RH was low. Aerosol hygroscopic growth is generally thought to increase the ambient light scattering coefficient much more so than the ambient light absorption coefficient, which would lead to a higher single-scattering albedo and, most likely, a more negative top-of-the-atmosphere RFE value

(i.e., stronger cooling effect). If the ambient RH was higher in many cases than the measurement RH and these measurements were removed from the data set, the reported data set is biased toward a smaller (less negative forcing) cooling effect. Given that the RFE values are most likely not representative of the SMEAR-II region (they use the global average constants) or actual atmospheric conditions, I question their value in this manuscript. If they are to be kept, the authors should re-emphasize that the RFE results are technically meaningful only in the trend analysis (in Table 3) and that the calculations are for dried aerosols using global average constants and thus considerable caution should be used when trying to interpret seasonal variation in RFE at SMEAR-II (Fig. 10). The RFE results could also be moved into the supplemental materials if length of manuscript is a concern.

SPECIFIC COMMENTS

Pg. 1, Line 14: Replace the words 'affected to' with 'influenced'.

Pg.1, Lines 20-21: 'For the aerosol particles to have a cooling (warming) effect, the reflectivity of the particles must be higher (lower) than the albedo of the surface...'. What is the definition of 'reflectivity' the authors are using (or is it being used in a qualitative sense here)? For aerosol particles, are the authors referring to aerosol single-scattering albedo (SSA) or some other reflective properties of the particles? It is not technically correct to state that '...the aerosol particles ... have a cooling (warming) effect (if) the SSA of the particles (is) higher (lower) than the albedo of the surface...'. Solar photons can be elastically scattered in the forward direction, which does not appreciably cool the surface or lower atmosphere. I would recommend removing this sentence as it is not really necessary anyway, but if kept in the manuscript the authors should state how they are defining the term 'reflectivity' and how that is being compared to surface albedo.

Pg. 2, Line 6: Replace 'concentration' with 'mass and/or volume'. Extensive AOPs are not dependent on the concentration of the particles but on the amount of aerosol

present. Freshly formed particles may have extremely high concentrations in the atmosphere and show very low scattering values.

Pg. 2, Line 8: Replace 'concentration' with 'amount of aerosol'. Same explanation as above.

Pg. 2, Line 11: Eliminate '. . . and not only on the amount of scattering and absorption.'

Pg. 2, Lines 27-28: Why is it important to measure the AOP's of PM1 particles? This should be stated in the manuscript somewhere.

Pg. 3, Line 13 and Line 23: When will the Luoma et al. manuscript in preparation be available? Will it be available by the time this manuscript is published? If not, other references on how the various instruments compare would be appropriate.

Pg. 3, Lines 19-22: The reported AOP's will vary depending on the measurement conditions. The direct aerosol radiative forcing effects at SMEAR-II, however, depend on the ambient conditions of T, P and RH, which were not usually the same as the measurement conditions. A discussion of how this would affect the results is appropriate. Are your seasonal results biased by a) eliminating the high ambient RH periods (which occur more frequently in the summer) before the driers were installed in 2013, or b) accepting these periods after 2013 with high ambient RH but reduced measurement RH? Some discussion of the fraction of data flagged as invalid due to high ambient RH before 2013 is warranted, as is the fraction deemed acceptable (with significant drying) after the driers were installed. This way the reader can understand if this was a frequent or merely occasional occurrence.

Pg. 4, Line 2: How warm does the sample air to the APS instrument get? Does this heating to above room temperature remove any volatile species other than water (e.g., ammonium nitrate)?

Pg. 4, Lines 11-12: 'We did not apply the truncation correction to the backscattering, since the backscattering measurements were much noisier, especially at the red

wavelength.' OK then the determination of b is wrong, as is the calculation of the up-scatter fraction, and the question is how far off are your values from the fully truncation-corrected values. An estimate of the uncertainty or error that enters the calculation of b due to not applying the truncation correction to the sigma-bsca values should be given. I agree that the sigma-bsca values are quite noisy at 1-minute resolution. At what resolution were you recording the raw data (1 second?,1 minute?, 10 minutes?, I don't see this listed in the manuscript)? Could you have averaged the sigma-bsca values to hourly or longer resolution before applying the corrections? This would perhaps help to beat down the noise a little.

Pg. 4, Lines 18-23: Which algorithm(s) or recommendations in Collaud Coen et al. (2010) were used? In that paper they evaluated four previous aethalometer correction schemes (Weingartner, Arnott, Schmid and Virkkula) and they also made new recommendations on the applicability of each in different circumstances.

Pg. 5, Line 12: Replace the word 'direction' with 'hemisphere'.

Pg. 5, all equations: The subscript font is quite small. Possibly it will look better in the published version.

Pg. 4-6, Section on Data Processing: Somewhere in this manuscript the authors need to give some estimate of the measurement uncertainties of the instruments they are using. I recommend looking at the work of Sherman et al. (2015, ACP) to see how they calculated the measurement uncertainties. It is a lot of work so I do not recommend that you try to repeat those analyses, but you should be able to reference their Table S2 'Total and precision fractional uncertainties (%) of measured PM1 and PM10 aerosol optical properties (AOPs) $\sigma$sp, $\sigma$bsp, and $\sigma$ap and calculated AOPs (e.g., the intensive AOPs) for 1-hour averaging time. Uncertainties are expressed as 95% confidence intervals.' and state the uncertainties relevant to your report.

Pg. 6, Line 8: '...the absorption would be dependent on wavelength as lambdaˆ-1...'. Rephrase as '...the absorption would have a wavelength dependence of approximately

lambdaˆ-1. . .'.

Pg. 6, Lines 28-29: While adjusting the AOP's to a common set of conditions is appropriate (and indeed necessary) to evaluate trends and to compare properties at different sites, you need the measurements at ambient conditions to determine the effects of aerosols on perturbing the surface radiation balance (i.e., their direct radiative/climate forcing effect). It would be good to provide some estimate or limit as to how different the AOP's are for dried vs. ambient air. Perhaps an example calculation, where the AOP's are adjusted to ambient conditions using some assumed conditions of T, P and RH, would help. I am sure there are studies of Finnish/Scandinavian/northern European aerosols where the aerosol hygroscopic growth was measured or calculated. These results could be used as a very rough scaling factor to calculate the AOP's at SMEAR-II at ambient atmospheric conditions. Otherwise the reader will not know if the presented dry aerosol RFE results are even close to those for real atmospheric conditions at SMEAR-II.

Pg. 7, Line 15: Replace 'describes' with 'provides information on'.

Pg. 8, Line 3: Replace 'chapter' with 'section'.

Pg. 8, Line 15-16: 'Naturally, the different methods used in the absorption data processing also affected the optical properties, which are dependent on the sigma-abs, such as omega-0 and k.' How much of a difference in omega-0 or k can be attributed to the different data processing methods? Is it a large or small difference? Could you provide an example where the same processing is used in two different time periods that shows how large of an effect this is?

Pg. 9, Line 6: Replace 'marked' with 'included'.

Pg. 9, Section 3.2, second paragraph: The 13%/year decrease in the ïĄşabs value at SMEAR-II is an important finding and should be emphasized here! Has this been observed at other sites in Finland and/or Europe? Can you provide a hypothesis as

to why this happened over the last decade at SMEAR-II station? Could it be more local or regional/continental scale effects? Is it due to less soot aerosols? Or possibly decreasing amounts of BrC?

Pg. 13, Line 3: Replace 'means' with 'suggests'.

Pg. 14, Section 3.6, second paragraph: The difference in the PM1/PM10 scattering ratio between Virkkula (2011) at 85% and the current study at 75% is a little concerning. There could have been long term changes in the environment at SMEAR-II region that might partially explain this, or it could be a difference in sampling conditions. Was there any RH measurement made at or near the impactors (as opposed to inside the nephelometer)? You need an RH measurement taken near the impactors to ensure you have a proper size cut (i.e., without the possible artifact you mention).

Pg. 15, Section 3.7: It needs to be stressed that the RFE calculations are for dry or semi-dry (RH<40%) aerosols.

Pg. 15, Line 23: '..., which makes the RFE decrease.' After decrease, add the parenthetical phrase '(i.e., become more negative)'.

Pg. 15, Line 26: Replace 'ine' with 'in'.

Pg. 16, Lines 5-13: This is a good explanation! The authors state that while the magnitude of the RFE perturbation cannot be precisely determined using this methodology, the trends probably can, and the RFE estimates they provide are most likely a lower limit to the true cooling effect.

Pg. 16., Lines 18-23, and Fig. 12: This is a discussion of systematic variability of aerosol optical properties. This type of systematic variability has been observed before. The earliest paper I know of that discussed this was Delene and Ogren (2002, J. Atmos. Sci, Fig. 8) which should be referenced. This was also in Sherman et al. (2015, ACP, Figs. 10a, 10b, 10d). Their results are consistent with those presented in this paper.

Pg. 27, Fig. 3: The largest decrease over time is for the larger accumulation mode

particles (i.e., 0.4-0.7 micrometer diam). Any ideas why?

Pg. 29, Fig. 5: The text in the legends are very small. This may, however, be acceptable to the technical editor.

Pg. 31, Fig. 7: Why are there breaks in the whiskers and some whiskers not attached to the boxes? Is this a plotting artifact or is additional explanation necessary as to what the whiskers are meant to display?

Pg. 33, Fig. 9: It appears that the whiskers are drawn as dashed lines with relatively long dashes and breaks. These should either be changed to solid lines or else changed to broken lines with smaller breaks in them.

Pg. 35, Fig. 11: Caption '. . .1000 grid points in total.' Should this be '10,000 grid points in total.'?

---

## Author Comment (AC1) · 31 Mar 2019

**RESPONSE BY THE AUTHOR**

First of all, than you for your comments! They were of great help in improving this study. I first reply on some major comments you both had and then, I reply to each of the comments separately.

**1 MAJOR CHANGES AND RESPONSE TO COMMON COMMENTS**

**1.1 SHORTENING THE MANUSCRIPT**

We analyzed a long data set in this study and we presented a lot of figures in the first manuscript. To emphasize the important parts of the manuscript, such as the trend analysis and RFE, we removed some of the figures as you recommended. For example, we got rid of the PM1 panels on some of the figures, since they gave no extra information that was relevant for this study. We also removed the section about the new particle formation events and aerosol optical properties. We agreed that it did not fit in the topic of this manuscript. I moved the old Fig. 3 to the supplementary material, since it was referred only once in the text. I moved the old Sect. 3.5 (Diurnal variation) to the supplementary material. The diurnal variation was a bit separate from the rest of the manuscript and it did not present any new information to Virkkula et al. (2011). I also moved the old Fig. 11 to the supplementary since after adding the seasonality and ambient RH to the RFE analysis, this figure did not feel important anymore. The number of figures decreased from 12 to 8 and the number of subfigures decreased as well. However, the supplementary material grew from one figure to nine figures since we answered to some of your questions without increasing the number of figures in the main article.

Since we added more description in the method section and improved the discussion we got now more text. We also added one table, which describes how the trend varies between different seasons.

**1.2 SIZE DISTRIBUTION ANALYSIS**

We are still keeping the size distribution analysis, since we found it interesting to study how the aerosol optical properties and their trends are related to the size distribution. For example, in Pandolfi et al., (2018) they were not using size distribution data and they had to assume what kind of changes in the size distributions cause the different types of trends for the $\alpha_{sca}$ observed at different stations. The study on the size distribution helps us to understand how the size dependent $b$ and $\alpha_{sca}$ vary between PM1 and PM10 measurements. However, we moved part of the size distribution analysis (old Fig. 3) to the supplementary material.

**1.3 CALCULATION OF RFE**

Another thing you both pointed out was the calculation of RFE by using global average values. We worked more with the topic and determined a more realistic RFE at SMEAR II. What I found difficult was determining the $b$ for moist conditions and I had to make rather rough estimates there. This would be an interesting topic to study more.

To emphasize the meaning of RFE: as stated by Sherman et al. (2015), the RFE ($\Delta F \, \delta^{-1}$) provides a means for comparing the intrinsic forcing efficiency of aerosols measured at different sites. The RFE describes the change that aerosol particles would have on the top-of-atmosphere radiative forcing ($\Delta F$) per unit of aerosol optical

depth ($\delta$). Since AOD is unitless, the unit of RFE is W/m2. The RFE is an intensive property and it does not depend on the amount of aerosols. If we wanted to know the $\Delta F$, we would need measurements of $\delta$, which, on the contrary, is an extensive property and depended on the amount of aerosols.

We have now determined three different types of RFE values:

1) $RFE_{H\&S}$ was calculated by using the constant values suggested by Haywood and Shine (1995). $RFE_{H\&S}$ was derived for dry particles. Here the subscript "H&S" refers to Haywood and Shine (1995).

2) $RFE_S$ was calculated by using seasonal averages for the environmental parameters ($D$, $A_C$, $R_S$). So here we let the fractional day length to vary; we used more realistic values for the surface reflectance according to Kuusinen et al., (2012) and took the snow cover into account; and we determined an average cloud fraction for each month. More detailed description is provided in the manuscript in Sect. 2.3.3 (p. 8 – 10). We added a figure in the supplementary material (Fig. S1a-c) describing the seasonal variability of these parameters. $RFE_S$ was derived for dry particles. Here the subscript "S" refers to "seasonal".

3) $RFE_{S,moist}$ was calculated similarly to $RFE_S$, but taking the ambient RH into account. Here we determined average RH for each month and we derived the AOPs to the average humidity. The seasonality of RH is presented in Fig. S1d. In determining the $\omega_0$ for humid conditions, we assumed that the absorption does not depend on the RH. The scattering was converted to humid conditions using the parametrization provided by Zieger et al. (2015), which is presented in Sect. 2.3.1. The parametrization was given only for total scattering so we could not use it for backscattering and determine $b$ with this parametrization. Instead we assumed that the $b$ has a linear dependency on RH. Fierz-Schmidhauser et al. (2010) observed that the $b$ decreased 30 % when the RH increased from dry conditions to 85 %, which we used in this study. Here the subscript "S,moist" refers to "seasonal" and ambient RH, which was > 50 % for each month. This is described in Sect. 2.3.3 (p. 10).

The results for different RFE values are presented in Sect. 3.6, Table 1., and Fig. 7.

**1.4 DATA PROCESSING**

You both commented that I should apply the truncation correction also to backscattering data. This has now been done.

I also made a small change to the Aethalometer flow correction (Fig. S2 added in the supplement), and changed the $C_{ref}$ value from 3.35 to 3.19. Thus there are small changes (less than 5 %) in the data presented in the article.

In this new version of the manuscript we have discussion about how the RH affects the scattering. Therefore we present the parametrization of the scattering enhancement factor in the Sect. 2.3.1. I have also added a better description about the Aethalometer correction algorithm used here. Since we were not able to submit my second manuscript, which would have presented the $C_{ref}$ at SMERA II, I added a short description about $C_{ref}$.

**2 RESPONSE TO MAJOR AND SPECIFIC COMMENTS**

The comments by the referees are listed with bolded font and the response by the authors are written with normal font. If the answer refers to a text that was added in the manuscript, the quotation is italicized.

**GENERAL COMMENT**

**The here presented manuscript describes the variability of several aerosol optical properties (AOPs) measured in southern Finland for more than a decade. The multi-year variation of AOPs is presented together with a detailed analysis of AOPs variability on a shorter timescale. Due to its time coverage, the dataset presented here is of great relevance and might help to understand how aerosol changed in a North European background site during the last ten years. The scheme and structure of the manuscript are linear and follows a logical order. However, the amplitude of the dataset generates a certain overloading of the manuscript, meaning that the results are not always properly discussed within a climatologic perspective but simply described. As a consequence, is difficult to identify the overall scientific message of the work. I truly believe that the paper covers the topic of interest of ACP, but I would recommend the authors to improve the discussion and interpretation of their results in order to better transmits their message to the readers. Hopefully, the major and specific comments reported below will help the authors to improve their work.**

We have now worked with the manuscript and improved the discussion especially in the sections that concern trend and RFE. We have also decreased the number figures in order to make the manuscript more readable and less crowded.

**MAJOR COMMENTS**

**I have the strong feeling that the manuscript is overloaded with figures, especially multipanels figures. First of all, due to the similarity between PM1 and PM10 (Figure 1, 2, 4), the discussion and presentation of results become particularly redundant in Section 3.2, 3.3, 3.4. The subsequent effect is that the discussion often focusses on the differences between the two aerosol fractions rather than on the reasons leading to the multi-year trends or seasonality. I thus suggest the authors show and describe PM10. This will lighten the paper and give more space for the climatologic interpretation of the results. Moreover, it appears that a considerable number of figures is poorly described or is not essential to the understanding of the results. I thus suggest the authors to reconsider the absolute relevance of certain graphs and to remove them from the manuscript or move them to the supplementary. More details can be found in the specific comments.**

I have now modified the multipanel figures so that they include only PM10 data, if the PM1 data presented no additional information. Some of the figures, which did not seem that relevant anymore, I have moved to the supplementary material.

**The dataset allows the investigation of multi-year variability and trends of AOPs and size distribution. The variability of AOPs is also investigated on a shorter time resolution but ignoring the year-to-year variability (Section 3.4, 3.5, 3.6). Thanks to the long-term measurement I would expect a work focusing on trends and multiple-year variability of AOPs. However, the analysis of trends is disconnected from the seasonal and diurnal variability and the consequent RFE. Therefore, I have some troubles in understanding what is the topic or scientific question acting as a glue between the sections, which in some cases (Section 3.3.1, 3.5 and 3.6) appear to be self-standing. I would thus suggest the authors to better exploit their long time series and focus on the long-term evolution/variability of AOPs including trends, impacts on seasons and, potentially, diurnal**

**variability. For instance, Section 3.3.1 is based on 2 months measurements only, what is the long-term implications of NPF on the aerosol optical properties, and is this short period representative of the 10 considered years? Moreover, Section 3.5 provides the diurnal variability of AOPs. Despite the fact of a weak variability, was the boundary layer dynamic changing within the 10-year period? It is hard to understand the relevance and implications of such variability. Similar reasoning applies to the monthly variability, did summer and winter experienced a change from 2006 to now?**

There is now a figure in the supplementary material (Fig. S3) that better describes the year-to-year variability of scattering, absorption and particulate volume for each season. The figure presents the time series of monthly medians separately for each season. It can be seen from the figure that the year-to-year variability is the highest in winter, when the amount of pollution is highly depended on the meteorological conditions, which is discussed in the supplementary.

To connect the trend analysis to the seasonal variation we determined trends separately for each season. The results are presented in Table 4 of the manuscript.

I agree with the Sections 3.3.1 and 3.5 being self-standing and we chose to remove 3.3.1 from this manuscript. Sect. 3.5 I chose to move in the supplementary.

**The calculation of the forcing efficiency is an extremely interesting topic, and up to me a decadal trend of RFE might represent the core of the entire manuscript together with Section 3.2 and 3.3. However, for the RFE estimations, the authors assumed the atmospheric (RH, cloud) and environmental (surface albedo, day length) variables as constant, which are not even specific for the SMEAR II station. Due to strong seasonality and, potentially, year-to-year variability of such variables, the final RFE estimations are unrealistic. I would suggest the authors implement constants representative, at least, of Southern Finland or, better, to use seasonal dependent variables. Other than that, any conclusion on climatic impacts of aerosol at SMEAR II will be highly questionable and of low interest.**

We have now worked more with this topic. See my answer 1.3 in the beginning of this document.

**SPESIFIC COMMENTS**

**P3L11: Why MAAP and PSAP are introduced if only data from the AE31 are used?**

I removed PSAP from the introduction but I left MAAP since I added a short description about the multiple scattering correction factor ($C_{ref}$) (see my response to comment about P3L23), which was determined by comparing Aethalometer and MAAP measurements.

**P3L13: This is irrelevant for the present manuscript.**

I removed this sentence.

**P3L23: I would say that, since Luoma et al. 20xx is not available, a better description should be provided here.**

I removed the self-citation from the manuscript. We had too optimistic expectations about the timetable with the other manuscript. We hope to submit the manuscript by the end of this year. I have also added a better description about determining the $C_{ref}$:

*"The $C_{ref}$ was determined by comparing the Aethalometer data, that was corrected only for the filter loading artefact, against the reference absorption coefficient ($\sigma_{abs,ref}$) measured by the MAAP.*

$$C_{ref} = \frac{\sigma_{ATN}}{L \cdot \sigma_{abs,ref}}.$$  *(6)*

*The resulted median value for $C_{ref}$ was 3.19, with a standard deviation of 0.67. "*

**P3L31: Can the author exclude the influence of hygroscopic growth?**

When the particles enter the electric charger, the sample air is not dried. However, since the sheath air is dried, the aerosol particles are also dry in the DMA of the DMPS setup. I added this explanation to the manuscript:

*"There is no active drying system in the TDMPS sample line to prevent particle losses. However, the sheath flows, which are used in the TDMPS system, are dried (RH < 40 %) so the particles are sampled in dry conditions."*

**P4L4: Which instruments, all of them?**

I meant only instruments that measure optical properties. For other instruments we should not have this problem. I fixed the sentence to:

*"In the present work, if the internal RH in any of the optical instruments exceeded 40 %, the data from that instrument were excluded from further analysis if not stated otherwise."*

**P4L11: It is not clear why the truncation correction was not applied for the backscattering. Does it mean that back-scattering can be affected by systematic error compared to total scattering? Was this assessed? Was it negligible? The authors present more than 10 years of data, more care in the presentation of the data correction is mandatory.**

Now, also the backscattering is corrected.

**P4L19: Multiple correction procedures were used or only the Collaud Coen et al. (2010) as stated later? If the correction of Collaud Coen was used, I honestly do not see the reason to cite all the other algorithms. Generally, I would not recommend the frequent self-citation of works that are not ready yet.**

I removed the self-citation and the citations to the other correction algorithms that were not used here. I added a description of the correction algorithm in the text:

*"Here, we corrected the Aethalometer data by using the correction algorithm described by Collaud Coen et al. (2010)*

$$\sigma_{abs,i} = \frac{\sigma_{ATN,i} - a_{s,i}\bar{\sigma}_{sca,s,i}}{C_{ref}\, L_{s,i}},$$ (3)

*where*

$$L_{s,i} = \left(\frac{1}{l\left(1-\bar{\omega}_{0,s,i}\right)+1} - 1\right) \cdot \frac{ATN_i}{50\ \%} + 1,$$ (4)

*and*

$$a_{s,i} = \bar{\zeta}_{sca,s,i}^{d-1} \cdot c \cdot \lambda^{-\bar{\alpha}_{sca,s,i} \cdot (d-1)}.$$ (5)

*In Eqs. 3 and 4, the subscript i indicates the number of the measurement and the subscript s indicates the average properties of the aerosol particles that are embedded in the filter spot. The over lined parameters are mean values from the start of the filter spot to the ith measurement. In Eq. 3, the $\sigma_{ATN}$ is the attenuation coefficient reported by the Aethalometer, a is the scattering correction parameter, $C_{ref}$ is the multiple scattering correction factor, and L is the loading correction function. In Eq. 4, the $\omega_o$ is the single scattering albedo (see Sect. 2.3.3) and the ATN is the light attenuation through the filter spot in percentages. In Eq. 5 the $\zeta_{sca}$ is the proportionality constant of the wavelength power law dependence of $\sigma_{sca}$ and $\alpha_{sca}$ is the Ångström exponent of the $\sigma_{sca}$ (see Sect. 2.3.3). For l, d, and c we used values 0.74, 0.564 and $0.329 \cdot 10^{-3}$ respectively. For scattering correction, we used measured $\sigma_{sca}$ values that were interpolated and extrapolated to the AE-31 wavelengths. Note that most of the symbols used for the variables are different from Collaud Coen et al. (2010). The reason is that in the present work the symbols are used for other variables below."*

**P6L15: I expect that BC from biomass burning and traffic has a different chemical composition. Isn't it in contrast with lines 13-15? If $\alpha_{abs}$ is simultaneously affected by size, chemical composition, sources and mixing, to what purpose is $\alpha_{abs}$ used here?**

I guess the Aethalometer model assumes that the $\alpha_{abs}$ is only depended on the chemical composition, which then depends on the source. In Hyytiälä, the BC particles are typically aged so they probably have a coating that affects the $\alpha_{abs}$ and because of this the Aethalometer model might not be functioning in Hyytiälä. We added some discussion about the aethalometer model in the discussion section:

*"Also, the $\alpha_{abs}$ is typically associated with the source of the BC and it is often used to quantify whether the BC is traffic or wood burning related (Sandradewi et al., 2008; Zotter et al., 2017) so that high $\alpha_{abs}$ is a sign of wood burning. In the source apportionment, $\alpha_{abs}$ close to one indicates that the BC is sourced from traffic. Since we observed relatively higher $\alpha_{abs}$ in winter, the results are in line with the assumption of domestic wood burning that takes place during winter. However, in summer, $\alpha_{abs}$ was often < 1, which would yield an unphysical fraction (over a 100 %) of traffic related BC. Values below 1 could have been caused by large BC particles ($D_p$ > 100 nm) that have a purely scattering coating (Lack and Cappa, 2010). It must be noted that the $\alpha_{abs}$ depends also on the correction algorithm. For example, if the $\sigma_{abs}$ was corrected with the algorithm proposed by Arnott et al. (2005), the mean ± SD of $\alpha_{abs}$ would have been 1.36 ± 0.51 (see Table S2). Using the $\alpha_{abs}$, which was determined by using the correction by Arnott et al. (2005), the results for the source apportionment would be different and they would show higher fraction of BC from wood burning. Further investigation of the complex nature of $\alpha_{abs}$ is omitted here."*

**P6L22: Equation 5 is quite different from Haywood and Shine (1995), is this the original source of the equation? What is the wavelength of RFE? Is then the aerosol optical depth measured or everything is calculated from Equation 5? Though I have quite some doubts on the choices of constants (see comments on Section 3.7), a better description of the equations and its limits should be provided, together with the motivations at the base of the choices of the constants and the subsequent uncertainties.**

It is the same equation as the Eq. 3 in Haywood & Shine (1995) but divided by the AOP. This was derived by Sheridan and Ogren (1999) (Eq. 8), to which I was missing a citation. Here we have not used the measurements of AOD. Haywood & Shine used the constants for calculating the $\Delta F$ independent of wavelength and Sheridan and Ogren (1999) used these same values in calculating the RFE at 550 nm. In addition to the H&S constants, we now calculated the RFE by using seasonally varying environmental parameters (see my 1.3 at the beginning of this document). Using seasonally varying parameters give some estimate about how realistic the RFE calculated by using the constant values is.

**P10L5-17: This part of the section mostly describes the technical aspects of the measurements. I would suggest to move them in the method section. Potentially into a new subsection discussing the data coverage and how the data set was reduced/validated.**

I moved this part to a new section "2.3.4 Data coverage" and described the data coverage better. I also added a table in the supplementary material where the data coverage from each month is presented.

*"If averaged over the whole measurement period, 81 % of the nephelometer data and 70 % of the aethalometer data were considered valid. All the AOPs had some gaps in the data (see Fig. 1). Most of these gaps in the time series of AOPs during the summers of 2009 and 2010 were due to too high RH. The gap in 2010 was due to maintenance and installation of the dryers and the switching inlet system. Some additional $\sigma_{bsca}$ data were missing, due to malfunction of the backscatter shutter of the integrating nephelometer. Dirty optics, malfunctions and maintenance caused the gaps in the $\sigma_{abs}$ data in 2012 and 2015.*

*Until March 2010, the integrating nephelometer and the aethalometer measured sample air that was not dried with any external dryers. During winter, the relative humidity (RH) remained below 40 %, since the sample air warmed up to room temperature (about 22 °C). Sometimes in summer, the RH of the sample increased to over the 40 % limit. If the RH was above 40 %, the data were flagged as invalid and they were omitted from the data analysis if not stated otherwise. About 25 % of all the data before March 2010 had to be removed due to too high RH. Almost all of the removed data was from summer and fall months (June – October) and if regarding only these months, 46 % of the data were flagged. If the  moist data was included the overall data coverage would increase to 89 % and 77 % for scattering and absorption data, respectively. Monthly data coverage is presented in Table S1. After the installation of the Nafion-dryers in March 2010, the humidity caused no further problems."*

**P11L3: This is the first and last time $\omega_0$ was discussed in Section 3.3. I am wondering if the four panels in Figure 4 showing $\omega_0$ are needed at all.**

We added discussion concerning the $\omega_o$ panel:

*"Kulmala et al. (2016) estimated that fresh eBC particles observed at SMEAR II are in the size range of 80 – 120 nm. That estimate was calculated in a simplified way from the relationship between particle number*

*concentrations and BCe concentrations. A better estimate is obtained from the size dependence of $\omega_o$. The darkest aerosol has $\omega_o < 0.6$ and GMD in the range of about 30 – 70 nm (Fig. 3b, 3f, and 3j). This has been shown to be the range of fresh BC (e.g., Kittelson, 1998; Casati et al., 2007; Zhang et al., 2008) which suggests the source of BC is not far, probably within some kilometers only."*

**P11L21-24: The inverse proportionality between $\alpha_{sca}$ and GMD is supposedly caused by the bimodal size distribution of the aerosol and the substantial presence of accumulation particles. Despite supported by a reference, there is no direct explanation of the physical causes behind such proportionality. Since this is contrary to expectations, as stated by the authors, a deeper reasoning and explanation should be provided.**

This was more and a detailed explanation is given in the Sect. S6 in the supplementary material. This was also described in the main text:

*"To study the reasons behind this relationship we generated first unimodal size distributions with two geometric standard deviations GSD = 1.5 and 2.0 and calculated both $\sigma_{sca}$ and $\sigma_{bsca}$ at $\lambda$ = 450, 550, and 700 nm with the Mie code with m = 1.517 + 0.19i and the $\alpha_{sca}$ and b from them. For unimodal size distributions the $\alpha_{sca}$ decrease with increasing GMD as is shown by the lines in Fig. 3c. Schuster et al. (2006) showed that the relationship may be the opposite for bimodal size distributions. Schuster et al. (2006) explained this behavior by that adding a larger or coarse particle size mode to a fine particle mode that is inefficiently scattering - for instance nucleation and Aitken mode particles – the larger mode contributes more efficiently to the Ångström exponent than the fine mode. The contribution of the particles smaller than 100 nm to GMD is larger than that of the larger particle modes which leads to the observed relationship. To study this in more detail we generated also bimodal size distributions. The analysis presented in the supplement (S4) shows that the $\alpha_{sca}$ of bimodal size distributions can be calculated as a linear combination of the $\alpha_{sca}$ of the modes, weighted by the fractions of $\sigma_{sca}$ of the respective modes. This explains the increase of $\alpha_{sca}$ with growing GMD."*

**P12L7-10: The diameter of the particles is the driver for both $\alpha_{sca}$ and b, I have some difficulties in understanding the relevance of the findings described here.**

The point here was to show that the b and $\alpha_{sca}$ are sensitive to different size ranges and that the bimodal size distribution for PM10 particles can make the examination of b and $\alpha_{sca}$ a bit complicated. For PM1 particles the variation of b and $\alpha_{sca}$ is easier to understand since the size distribution is closer to a unimodal size distribution:

*"There was a negative correlation between the GMD and PM10 b (Fig. 3d) as expected, but the correlation was rather weak. On the contrary, the correlation between the VMD$_{tot}$ and PM10 b was slightly positive (Fig. 3h). The negative correlation of $\alpha_{sca}$ with VMD$_{tot}$ and the positive correlation of b with VMD$_{tot}$ for the PM10 particles indicates that the $\alpha_{sca}$ and b were sensitive to different size ranges. The $\alpha_{sca}$ decreased when there are more coarse particles present, but for the b the coarse particles seem to have no expected effect and the b increased with increasing VMD$_{tot}$. Fig. 4a. shows that when the VMD > 1500 nm, the peak of DV/dlogD$_p$ in the accumulation mode was much lower and tilted towards the smaller diameters than compared to the situations where the VMD < 1000 nm. This is in line with Collaud Coen et al. (2007), who stated that in the Jungfraujoch data, b was sensitive to particles smaller than 400 nm and that the sensitivity of the $\alpha_{sca}$ was at its maxima for particle diameters between 500 and 800 nm.*

*For the PM1 particles, the measured $\alpha_{sca}$ and b were well in line with the modeled values (Figs. 3k and l), since the coarse mode particles were removed prior to the measurements, the shape of the size distribution was closer to a unimodal size distribution, and the $VMD_{fine}$ described better how the accumulation mode shifted."*

**P13L2-3: Here is stated that long-range transport brings pollution to the station, but 70% of black carbon comes from local and regional sources (P13L7). These two statements are contradictory.**

Here, I meant to say that the local and regional BC emissions are also important in winter. I formulated this paragraph to:

*"Hyvärinen et al. (2011) observed increased equivalent black carbon (eBC, the BC concentration determined optically from $\sigma_{abs}$ measurements) concentrations at SMEAR II in winter, when the long-range transport brings pollution from the central and eastern Europe. However, Hienola et al. (2013) estimated that about 70 % of the measured eBC at SMEAR II is emitted from local or regional sources or transported from Finnish cities so also the local and regional emissions have a significant role in the elevated eBC concentrations."*

**P13L4-5: I am not sure to understand the relevance of the polar dome here.**

I formulated this sentence and removed the reference to polar dome. I meant that the southern air masses are more common at SMEAR II in winter, which brings pollution from central or eastern Europe.

**P13L30-33: You have the size distribution data, why should you make a hypothesis on size distribution from optical properties?**

I added a figure about the seasonality of the size distribution in the supplementary material so no hypothesis needs to be done.

*"The seasonal variation in $\alpha_{sca}$ and b depends on the seasonal variation in the size distribution of the particles. Both $\alpha_{sca}$ and b were maximal in summer and minimal in winter, suggesting that in summer, the particle population consisted of smaller particles than in winter. Closer investigation on the size distribution, which is presented in Fig. S3 and S4, reveals that in winter, the $VMD_{tot}$ was experiencing it minimum due to a lack of coarse mode particles. This is in contrast with the observation or smaller $\alpha_{sca}$ and b. In fact, the seasonal variation of $\alpha_{sca}$ and b was explained by the seasonal variation of accumulation mode* and $VMD_{fine}$, which is a good indicator for the shifting accumulation mode. *In winter, the accumulation mode was shifted towards larger sizes and the median of $VMD_{fine}$ was about 350 nm. In summer the situation was the opposite and $VMD_{fine}$ was about 250 nm."*

**P15L15-18: Here you need to be careful with the instrumental error. Do you mean that absorption was close to the detection limit of the instrument or that dominant presence of non-absorbing particles caused a decrease of light transmitted through the filter and apparent absorption (Müller et al., 2011)?**

I formulated this part. I meant the apparent absorption caused by scattering and how it causes relative large uncertainty since the scattering is high and absorption is low.

*"The deviation of the $\sigma_{abs}$ PM1/PM10 ratio clearly varied seasonally. In summer, the variation was considerably higher than in winter. In the correction algorithm, which was used for the absorption data (Eq. 3), part of the $\sigma_{sca}$ is subtracted from $\sigma_{abs}$ as an apparent absorption (Muller et al., 2011). This subtraction causes relatively high uncertainty when the $\sigma_{abs}$ is low and $\sigma_{sca}$ is high like it is in summer. This uncertainty is emphasized for PM10 measurements, since the $\sigma_{sca}$ is relatively higher than $\sigma_{abs}$ if compared to PM1 measurements. The uncertainty in the measurements also explains why there were so many values above 1 measured in the PM1/PM10 $\sigma_{abs}$ ratio."*

**P15L23-24: The RFE trends are not described, discussed or interpreted. This recalls my major comments. The manuscript is loaded with data that are never discussed. Provide an interpretation or remove**

I added more discussion in Sect. 3.6 about the trends and seasonal variation of RFE.

**Fig.10a. By the way, add to all figures the panel reference.**

Fixed this.

**P15L26-28: Since RFE is calculated from *b* and $\omega_0$ and all the environmental variables are kept constant, RFE must change with *b* and $\omega_0$. As follow up to the second major comment, the authors are required to provide a deeper interpretation of their results.**

We added more discussion about the effect of environmental variables and about the effect of the RH. See my answer 1.3 at the beginning of the document. Taking the seasonality of the environmental parameters into account amplifies the seasonal variation of RFE. The effect of RH is not as pronounced as the effect of using the seasonally varying environmental parameters.

**P15L29: How the monthly RFE should be interpreted if the atmospheric and environmental parameters are kept constant? Moreover, it appears that the constants are not representative of SMEAR II. So, what should we really learn out of RFE?**

More discussion about this in Sect. 3.6. As stated by Sherman et al. (2015), the RFE provides a means for comparing the intrinsic forcing efficiency of aerosols measured at different sites, this is the reason for calculating it by using the same constants that have been used in other publications.

**P16L5-8: The problem here is that the aerosol optical depth is affected by RH and subsequent hygroscopic growth. So, all your RFE are systematically underestimated by an unknown factor. However, it is unclear if optical depth is measured or calculated.**

Our study used only in-situ measurements of AOPs and we have not measured the AOD. I have emphasized this in the text:

*"The RFE (or ΔFδ$^{-1}$) describes only the efficiency of the aerosol particles in cooling or warming the climate per unit of aerosol optical depth (δ). Eq. 11 assumes that the properties of the aerosol particles are uniform in the atmospheric column that is rarely the case in reality. In ambient air, we should also take into account the variability in RH as a function of height. At the top of the boundary layer we typically have RH values close to 100 %. Here, we determined the RFE by using the RH measured near the ground (16 m). The simplified RFE does not give an absolute value for the aerosol forcing, however, it can still indicate how the changes in AOPs affect the climate."*

**P16L9-10: Nessler et al. (2005) suggested that water uptake does not enhance absorption coefficient of BC.**

I added a citation to this study and took this finding into account when determining the effect of ambient RH on the $\omega_o$ and further on to RFE.

**P16L11-13: From this work, it is impossible to quantify the change of radiative forcing, nor the effects on the climate. First, RFE trends are not discussed: Second, the absolute values of RFE, as admitted by the authors, are far from being realistic. Moreover, why should we use RFE as "an indicator of how the properties of the aerosol particles have been changing" if the changes of aerosol particles have been measured (Section 3.2, 3.3, 3.4)?**

I added here the RFE calculated by using more realistic values (see my answer 1.3). I also improved the discussion. The point of determining the RFE is not to quantify the radiative forcing (for that we would need a lot more parameters, AOD for example). However it describes how the efficiency of aerosol particles to cool (or warm) the climate has changed during the measurement period.

**P17L1-2: Fig 1 and Fig. 2 show a net decrease of aerosol number concentration, but is this due to the implementation of new emission policies only? How did precipitation and air circulation changes from 2006? I would recommend the authors to consider all possibilities and base their final conclusions on their data and existing literature.**

I formulated this:

*"The extensive AOPs, as well as the aerosol number and volume concentration, tended to decrease. Our observation was in line with the other studies conducted in Europe and North America that also observed decreasing trends for the extensive AOPs (Collaud Coen et al., 2013; Pandolfi et al., 2018; Sherman et al., 2015), number concentration (Asmi et al., 2013) and aerosol optical depth (Li et al., 2014). This uniform decreasing trend in the amount of aerosol particles suggests that the anthropogenic emissions of particulate matter and gases that take part in secondary aerosol formation has been decreasing in Europe and North America The observed tendency for b and $\alpha_{sca}$ to increase indicated that the particle size distribution was moving towards smaller diameters. A more detailed investigation revealed that the number of larger accumulation mode particles decreased relatively the fastest, which also supports the assumed decrease in pollution."*

**F1: This figure is too crowded, I do not think that showing both PM1 and PM10 as any relevance (see major comments).**

I removed the PM1 column from here.

**F2: Here 5 panels are used to show that the total number of particles decreases and the size distribution is shifted to the smaller diameters. I would say that two panels will do efficiently the job. For example, one panel showing the total particle number concentration (Nfine+Ncoarse) and a second panel showing the ratio between Nfine and Ncoarse or the GMD. Note that Nfine is never defined in the text, is this accumulation+ Aitken+nucleation? Please provide a description.**

I removed two of the panels. Now there are panels for total particle number concentration ($N_{tot}$), total particle volume ($V_{tot}$) and $VMD_{tot}$. We kept these parameters since $N$ describes well the overall decrease in aerosol particles and $V$ describes the amount of optically active aerosol matter. VMD was used instead of GMD, since it is more sensitive to changes in the optically active size ranges.

**F1-2 As a follow up of my previous comments, I would find a way to merge together a reduced version of Figure 1 and 2, with the goal to focus on the relationship between physical and optical properties described in the text.**

I see this point, but I could not merge these figures due to technical reasons (the figure became too crowded and the fonts too small). I would also like to keep the optical parameters separated from the size distribution parameters.

**F3 This Figure is mentioned only once at P9L26, it does not appear to provide a key insight into the understanding of data interpretation. I would thus recommend to move it to the supplementary.**

We removed this figure from the main manuscript according to your recommendation. However, we kept it in the supplementary material since it proves that the relatively highest decrease occurred in the larger side of the accumulation mode and thus supports the observed trends in increasing $b$ and $\alpha_{sca}$.

**F5: The size distribution of PM10 contains all the necessary data to investigate the size distribution in PM1. This is clear in panels (c) and (d), where the size distributions below 1 um are exactly the same. This recalls my general comments, is a separated discussion of PM1 and PM10 really necessary?**

In Figs. 4a and b we used different VMD values as limit values for averaging the size distributions. These values differ for PM1 and PM10 as well as do the $b$ and $\alpha_{sca}$. So if we want to study how the $b$ and $\alpha_{sca}$ are related to the VMD, we need to investigate these separately for PM1 and PM10. However, since GMD is practically the same for PM1 and PM10, we now show the average size distribution limited by the GMD for the PM10 only. We also removed the panel where the PM1 AOPs were compared against GMD in the old Fig. 4.

**F6: The figure shows the nucleation events and the related change in the real part of the refractive index. However, I think that it is largely overcrowded. On page 12, lines 26-27 sufficiently describe the absence of change in the observed AOPs. Due to the low relevance of AOPs variability in this context, I would suggest removing the third and fourth panels from the top. Finally, I am wondering what is the relevance of 2 months data over a 10 year period.**

We removed this figure and the section discussing about it from the manuscript, since we realized that it is too much for this article.

REFEREE 2

**GENERAL COMMENTS**

**This manuscript reports on the optical and microphysical properties of aerosols collected for over a decade at the SMEAR-II atmospheric monitoring station in Finland. These data are valuable in determining long-term trends and variabilities in aerosol properties, which are useful to climate modelers. The statistical distributions of aerosol properties presented in this paper should also be useful to GCM and CTM modelers for model initialization and validation exercises. As such, I think this paper is appropriate for inclusion in ACP and warrants publication after attention to the comments listed below.**

**The paper is well organized and there are only a few places were the English usage could be improved. The methodology used by the authors is excellent and of high quality, and the data presented are in general valid and relevant. I do have some comments that suggest some additional thought be given to provide better explanations of the observations, and I think a better discussion of how drying the sampled aerosols might influence the RFE results is warranted (see comments below).**

**The paper is a bit long but the amount of data being presented from over a decade at this site and the necessary discussions warrant a longer paper. In looking for possible ways to decrease the length of the paper, the only thing I see is to remove the size distribution discussion. While it is interesting in its own right and assists in the interpretation of the AOP data, it is not strictly necessary in this paper. I will leave that decision up to the authors and the editor.**

We have now kept the size distribution section since they explain the behavior of AOPs. However, we cut some other elements (Sect. 3.3.1 and 3.5) from the manuscript to shorten it a bit and to emphasize important results.

**In order to better interpret trends and variability, some estimates of the measurement uncertainty should be provided. I would point the authors to the work of Sherman et al. (2015, ACP), who put a great deal of effort into estimating measurement uncertainties for aerosol optical properties. There is no need to repeat this exercise in detail, but at the very least this reference should be included and some mention of the measurement uncertainties for the TSI nephelometer should be provided.**

We cited this work concerning the uncertainty of the Nephelometer. Since they used PSAP in measuring absorption, we determined the uncertainty of the Aethalometer in a similar manner to Backman et al. 2017. We determined that the uncertainty of the $\sigma_{abs}$ was about 23 %.

**It was a bit disappointing to find little or no discussion in sections 3.1-3.3 on the relevance and importance of the measurements and their long-term trends and variability. A considerable amount of discussion is presented in sections 3.4-3.7 to explain the seasonal and diurnal variability, etc., and I would like to see more of this in sections 3.2 and 3.3. For the trends, for example, it would be useful to know how these trends compared with other long-term trends in Europe. The Pandolfi paper is cited and is an excellent place to start. Some additional information can be found in Collaud-Coen et al. (2013, ACP), and this paper should also be cited when comparing the optical property (scattering and absorption) measurements.**

      We have now added discussion to these sections, especially to Sect. 3.2.

**There is also little discussion on the importance of measuring the optical properties in two different size ranges (PM1 and PM10). What does it tell you about sources, ageing, human contribution, etc., if the PM1 fraction for a given parameter is almost as large as the PM10 fraction? What does it mean if the PM1/PM10 ratio is changing over time? The authors went to the trouble of adding this additional set of PM1 measurements in 2010 and have several tables and graphs in this paper showing the results. They need to say why they are important and what we learn from them.**

      We added a motivation to measure PM1 to the introduction:

*"The measurements of AOPs were started for aerosol particles smaller than 10 µm in diameter (PM10). The PM10 measurements are sensitive to coarse particles that are typically primary and originated from natural sources, such as soil dust and sea salt. To obtain additional information about submicron particles, parallel measurements of AOPs for PM1 were launched in June 2010. Motivation to measure also PM1 particles is that secondary aerosols (both natural and anthropogenic), and anthropogenic primary aerosols are typically submicron particles. Having measurements for different cut-offs makes the measurements also more comparable between different stations, since stations might use different cut-off sizes."*

**The purpose of the lengthy discussion of the comparison of the optical and microphysical properties on page 11 is not clear to me. The manuscript title indicates that this manuscript is about the aerosol optical properties, so why are there size distribution data included in the results and discussions? They are of course useful for interpreting the optical properties, so they have value, and good agreement between measured optical properties and ones calculated from the microphysical measurements give increased confidence in the findings of the study. Perhaps the authors can state that more clearly. The size distribution results could also go in the supplemental materials section if length of the paper becomes a concern.**

      We have kept the size distribution study in the manuscript since the size distributions explain most of the optical properties. The size distribution measurements are independent from the optical measurements so together they increase confidence to the results. We even added some discussion of the size dependence of single-scattering albedo as wished by the other reviewer. Another addition is an explanation of the increasing Ångström exponent with increasing geometric mean diameter as also wished by the other reviewer. That analysis is in the supplement.

**The RFE calculations in this paper use the global average constants of Haywood and Shine rather than ones estimated or derived for the local area. This is probably OK for trend analysis but the magnitude of the forcing is wrong, especially when considering seasonal variations. For example, the constant used for the global average surface albedo (0.15) does not represent that of the boreal forest around SMEAR-II station over all seasons… it should be significantly higher in winter due to snow cover and (I would guess) lower in summer.**

As stated by Sherman et al. (2015), the purpose of determining the RFE is to provide a means for comparing the intrinsic aerosol forcing efficiency of aerosols measured at different sites. We calculated the RFE by using the constant values to have results comparable with other studies in very different types of environments (e.g. Sheridan and Ogren, 1999; Andrews et al., 2011; Sherman et al., 2015; Shen et al., 2018) and to study how the RFE changes with varying $\omega_0$ and $b$. However, we have now determined RFE also by using more realistic environmental parameters also. See my answer 1.3 at the beginning of this document.

**Also, the measurement relative humidity (RH) and the ambient RH were generally different (this occurred most frequently in the summers). The authors state that if the sample RH was above 40%, the data were flagged and marked as invalid. This implies that the SSA and b values are only accurate when the ambient RH was low (i.e., close to the measurement RH), and that the RFE results are only appropriate for times when the ambient RH was low. Aerosol hygroscopic growth is generally thought to increase the ambient light scattering coefficient much more so than the ambient light absorption coefficient, which would lead to a higher single scattering albedo and, most likely, a more negative top-of-the-atmosphere RFE value (i.e., stronger cooling effect). If the ambient RH was higher in many cases than the measurement RH and these measurements were removed from the data set, the reported data set is biased toward a smaller (less negative forcing) cooling effect. Given that the RFE values are most likely not representative of the SMEAR-II region (they use the global average constants) or actual atmospheric conditions, I question their value in this manuscript. If they are to be kept, the authors should re-emphasize that the RFE results are technically meaningful only in the trend analysis (in Table 3) and that the calculations are for dried aerosols using global average constants and thus considerable caution should be used when trying to interpret seasonal variation in RFE at SMEAR-II (Fig. 10). The RFE results could also be moved into the supplemental materials if length of manuscript is a concern.**

We have now estimated the $\omega_o$ and $b$ for ambient RH as well and taken this into account in the RFE calculations. See my answer 1.3 at the beginning of this document for a more detailed description.

**SPECIFIC COMMENTS**

**Pg. 1, Line 14: Replace the words 'affected to' with 'influenced'.**

Fixed.

**Pg.1, Lines 20-21: 'For the aerosol particles to have a cooling (warming) effect, the reflectivity of the particles must be higher (lower) than the albedo of the surface…'. What is the definition of 'reflectivity' the authors are using (or is it being used in a qualitative sense here)? For aerosol particles, are the authors referring to**

aerosol single-scattering albedo (SSA) or some other reflective properties of the particles? It is not technically correct to state that '…the aerosol particles… have a cooling (warming) effect (if) the SSA of the particles (is) higher (lower) than the albedo of the surface…'. Solar photons can be elastically scattered in the forward direction, which does not appreciably cool the surface or lower atmosphere. I would recommend removing this sentence as it is not really necessary anyway, but if kept in the manuscript the authors should state how they are defining the term 'reflectivity' and how that is being compared to surface albedo.**

I reformulated this part.

**Pg. 2, Line 6: Replace 'concentration' with 'mass and/or volume'. Extensive AOPs are not dependent on the concentration of the particles but on the amount of aerosol present. Freshly formed particles may have extremely high concentrations in the atmosphere and show very low scattering values.**

Replaced.

**Pg. 2, Line 8: Replace 'concentration' with 'amount of aerosol'. Same explanation as above.**

Replaced this as well.

**Pg. 2, Line 11: Eliminate '…and not only on the amount of scattering and absorption.'**

Eliminated.

**Pg. 2, Lines 27-28: Why is it important to measure the AOP's of PM1 particles? This should be stated in the manuscript somewhere.**

We added a motivation to measure PM1 particles in the introduction (see my answer to one of the major comments).

**Pg. 3, Line 13 and Line 23: When will the Luoma et al. manuscript in preparation be available? Will it be available by the time this manuscript is published? If not, other references on how the various instruments compare would be appropriate.**

I had too optimistic expectations about the timetable with that manuscript… I have removed self-citation from the manuscript and modified the text.

**Pg. 3, Lines 19-22: The reported AOP's will vary depending on the measurement conditions. The direct aerosol radiative forcing effects at SMEAR-II, however, depend on the ambient conditions of T, P and RH, which were not usually the same as the measurement conditions. A discussion of how this would affect the results is appropriate. Are your seasonal results biased by a) eliminating the high ambient RH periods (which occur more frequently in the summer) before the driers were installed in 2013, or b) accepting these periods after 2013**

**with high ambient RH but reduced measurement RH? Some discussion of the fraction of data flagged as invalid due to high ambient RH before 2013 is warranted, as is the fraction deemed acceptable (with significant drying) after the driers were installed. This way the reader can understand if this was a frequent or merely occasional occurrence.**

The RFE does not depend on T or p, since it is an intensive property. But it does depend on the RH and I have now taken this into account in the paper. I added median values of other AOPs including the high RH conditions as well. I also added a more detailed description about the fraction of invalidated data in the new "2.3.4 Data coverage" section.

I now did the trend analysis for a data set, where the high humidity conditions were included. The trends did not change remarkably compared to the data set, where the moist conditions were excluded. I had a typo in the old manuscript concerning the installation of the driers. The driers were installed already in 2010, not in 2013, so there were only four years of measurements without the drier.

**Pg. 4, Line 2: How warm does the sample air to the APS instrument get? Does this heating to above room temperature remove any volatile species other than water (e.g., ammonium nitrate)?**

The sample air is heated to 40 °C and for example ammonium nitrate would be evaporated, however, the concentration of ammonium nitrate is very low in Hyytiälä so this should not affect in our study. Alternatively, there are many other volatile compounds at SMEAR II. These compounds take part in the secondary aerosol particle formation and the resulting particles are typically smaller than what we measure with the APS. Thus we believe that heating the sample up to 40 °C has no significant effect here.

**Pg. 4, Lines 11-12: 'We did not apply the truncation correction to the backscattering, since the backscattering measurements were much noisier, especially at the red wavelength.' OK then the determination of *b* is wrong, as is the calculation of the upscatter fraction, and the question is how far off are your values from the fully truncation corrected values. An estimate of the uncertainty or error that enters the calculation of *b* due to not applying the truncation correction to the $\sigma_{bsca}$ values should be given. I agree that the s $\sigma_{bsca}$ values are quite noisy at 1-minute resolution. At what resolution were you recording the raw data (1 second?,1 minute?, 10 minutes?, I don't see this listed in the manuscript)? Could you have averaged the $\sigma_{bsca}$ values to hourly or longer resolution before applying the corrections? This would perhaps help to beat down the noise a little.**

Backscattering data has now been corrected as well.

**Pg. 4, Lines 18-23: Which algorithm(s) or recommendations in Collaud Coen et al. (2010) were used? In that paper they evaluated four previous aethalometer correction schemes (Weingartner, Arnott, Schmid and Virkkula) and they also made new recommendations on the applicability of each in different circumstances.**

We used the new recommendation presented in that paper. I added a better description about the correction algorithm we used.

**Pg. 5, Line 12: Replace the word 'direction' with 'hemisphere'.**

Replaced.

**Pg. 5, All equations: The subscript font is quite small. Possibly it will look better in the published version.**

I have now increased the font a bit for so that it is easier to review.

**Pg. 4-6, Section on Data Processing: Somewhere in this manuscript the authors need to give some estimate of the measurement uncertainties of the instruments they are using. I recommend looking at the work of Sherman et al. (2015, ACP) to see how they calculated the measurement uncertainties. It is a lot of work so I do not recommend that you try to repeat those analyses, but you should be able to reference their Table S2 'Total and precision fractional uncertainties (%) of measured PM1 and PM10 aerosol optical properties (AOPs) $\sigma_{sp}$, $\sigma_{bsp}$, and $\sigma_{ap}$ and calculated AOPs (e.g., the intensive AOPs) for 1-hour averaging time. Uncertainties are expressed as 95% confidence intervals.' and state the uncertainties relevant to your report.**

We have now added the uncertainty of the Nephelometer in the manuscript and we estimated the uncertainty of the Aethalometer in Sect. 2.3.2. An estimation for the intensive AOPs is presented in the supplement.

**Pg. 6, Line 8: '...the absorption would be dependent on wavelength as lambdaˆ-1…'. Rephrase as '…the absorption would have a wavelength dependence of approximately lambdaˆ-1…'.**

Rephrased this.

**Pg. 6, Lines 28-29: While adjusting the AOP's to a common set of conditions is appropriate (and indeed necessary) to evaluate trends and to compare properties at different sites, you need the measurements at ambient conditions to determine the effects of aerosols on perturbing the surface radiation balance (i.e., their direct radiative/climate forcing effect). It would be good to provide some estimate or limit as to how different the AOP's are for dried vs. ambient air. Perhaps an example calculation, where the AOP's are adjusted to ambient conditions using some assumed conditions of T, P and RH, would help. I am sure there are studies of Finnish/Scandinavian/northern European aerosols where the aerosol hygroscopic growth was measured or calculated. These results could be used as a very rough scaling factor to calculate the AOP's at SMEAR-II at ambient atmospheric conditions. Otherwise the reader will not know if the presented dry aerosol RFE results are even close to those for real atmospheric conditions at SMEAR-II.**

We did more analysis on RFE, where we took the seasonality of the $D$, $R_S$, and $A_C$ into account. We also determined the RFE for ambient conditions by calculating the $b$ and $\omega_0$ for moist conditions. See my answer 1.3 for a more detailed description.

**Pg. 7, Line 15: Replace 'describes' with 'provides information on'.**

Replaced.

**Pg. 8, Line 3: Replace 'chapter' with 'section'.**

Replaced.

**Pg. 8, Line 15-16: 'Naturally, the different methods used in the absorption data processing also affected the optical properties, which are dependent on the sigma-abs, such as ω₀ and k.' How much of a difference in $\omega_0$ or k can be attributed to the different data processing methods? Is it a large or small difference? Could you provide an example where the same processing is used in two different time periods that shows how large of an effect this is?**

Table S2 in the supplementary material presents various absorption depended AOPs that were determined from absorption data that was corrected using the correction algorithm by Arnott et al. (2005). In this correction algorithm we used the same $C_{ref}$ as Virkkula et al. (2011). Compared to Virkkula et al. (2011), we used a corrected spot size and flow, and we invalidated situations when the RH > 40 % (Virkkula et al. (2011) invalidated situation when the RH > 50 %). The RH limits for acceptable data were taken from WMO GAW recommendations. In the GAW guidelines of 2003 the recommendation was to maintain RH < 50% but the limit was later lowered to RH < 40% (WMO/GAW, 2003, 2016).

The absorption at 520 nm was similar for both corrections but at other wavelengths the were differences. At lower wavelengths the $\sigma_{abs}$ was higher for the data that was corrected by the Arnott et al. (2005) algorithm. At higher wavelengths the situation was the opposite. Thus there was a notable change is in the $\alpha_{abs}$ that describes the wavelength dependency of $\sigma_{abs}$ and the $\alpha_{abs}$ was 1.4 for the $\sigma_{abs}$ data that was corrected by the Arnott et al. (2005) algorithm. For other $\sigma_{abs}$ depended parameters there were no significant differences at green wavelength. We did not do the analysis for the *k* since running the iteration for the whole data set takes a long time and we expected to see no large differences.

**Pg. 9, Line 6: Replace 'marked' with 'included'.**

Replaced.

**Pg. 9, Section 3.2, second paragraph: The 13%/year decrease in the $\sigma_{abs}$ value at SMEAR-II is an important finding and should be emphasized here! Has this been observed at other sites in Finland and/or Europe? Can you provide a hypothesis as to why this happened over the last decade at SMEAR-II station? Could it be more local or regional/continental scale effects? Is it due to less soot aerosols? Or possibly decreasing amounts of BrC?**

I do not think that there has been this steep decrease in other rural or remote stations or at least I could not find any citations. The trend of absorption in PM1 aerosol was calculated for only 7.5 years of data and it is more sensitive to extreme values in the time series. Here I believe the steep slope was caused by few extreme high values measured at the beginning of 2012. I find the trends calculated for the PM10 data more reliable since it is calculated for over a ten year period. I added some discussion about this difference in the results.

*"For the PM1 $\sigma_{abs}$, we observed a very steep decrease (-12 %yr⁻¹), which was probably caused by very high $\sigma_{abs}$ measured in January and February in 2012. Also the data gaps in winter 2013 and 2015 could have affected the*

*trends. The time series, of which the trends were determined for the PM1 measurements, were only 7.5 years long. Trends, which are determined for shorter time series are more sensitive to year-to-year variability. This kind of extreme values can induce relatively large trends, which is why trend analysis for short time series (less than ten years) should be treated with caution."*

**Pg. 13, Line 3: Replace 'means' with 'suggests'.**

Replaced.

**Pg. 14, Section 3.6, second paragraph: The difference in the PM1/PM10 scattering ratio between Virkkula (2011) at 85% and the current study at 75% is a little concerning. There could have been long term changes in the environment at SMEAR-II region that might partially explain this, or it could be a difference in sampling conditions. Was there any RH measurement made at or near the impactors (as opposed to inside the nephelometer)? You need an RH measurement taken near the impactors to ensure you have a proper size cut (i.e., without the possible artifact you mention).**

Unfortunately we do not have any RH measurements near the impactor or the inlet. The impactor is inside the measurement cottage so the air has some time to warm up to room temperature (about 22°C). Of course it does not help during the summer, when the temperature outside is similar to the room temperature. The difference that we observe between these two studies could have been caused by some technical issues like this.

Another thing is that Virkkula et al. (2011) determined the ratio from scattering calculated with a Mie model from size distributions measured with the DMPS and the APS. In the present work it was determined from the scattering measured with the nephelometer with the alternating PM1-PM10 inlet. In the present work we omitted the PM1/PM10 ratio calculation from Mie modeling, however, so the ratios are not strictly comparable. Both approaches have fairly large uncertainties associated with the large particles.

**Pg. 15, Section 3.7: It needs to be stressed that the RFE calculations are for dry or semi-dry (RH<40%) aerosols.**

I have tried to stress this in text. But now there are also RFE calculated for more realistic conditions and ambient moisture.

**Pg. 15, Line 23: '…, which makes the RFE decrease.' After decrease, add the parenthetical phrase '(i.e., become more negative)'.**

Added the phrase.

**Pg. 15, Line 26: Replace 'ine' with 'in'.**

Replaced.

**Pg. 16, Lines 5-13: This is a good explanation! The authors state that while the magnitude of the RFE perturbation cannot be precisely determined using this methodology, the trends probably can, and the RFE estimates they provide are most likely a lower limit to the true cooling effect.**

**Pg. 16., Lines 18-23, and Fig. 12: This is a discussion of systematic variability of aerosol optical properties. This type of systematic variability has been observed before. The earliest paper I know of that discussed this was Delene and Ogren (2002, J. Atmos. Sci, Fig. 8) which should be referenced. This was also in Sherman et al. (2015, ACP, Figs. 10a, 10b, 10d). Their results are consistent with those presented in this paper.**

I added the citations here and included some discussion too.

*"These relationships were also observed in a study of AOPs at the Station for Observing Regional Processes of the Earth System (SORPES), a measurement station in Nanjing China (Shen et al., 2018). Also, Delene and Ogren (2002) and Sherman et al. (2015) observed similar systematic variability between $\sigma_{sca}$, $\omega_0$, b, and $RFE_{H\&S}$ at several North American measurement stations; when the $\sigma_{sca}$ increases, the $\omega_0$ increases and the b decreases. Sherman et al. (2015) suggested that this variability could be caused by deposition of larger particles, which typically absorb less light. Delene and Ogren (2002) observed that $RFE_{H\&S}$ increases (i.e. becomes less negative) with increasing $\sigma_{sca}$, but Sherman et al. (2015) did not observe this trend. "*

**Pg. 27, Fig. 3: The largest decrease over time is for the larger accumulation mode particles (i.e., 0.4-0.7 micrometer diam). Any ideas why?**

I added some discussion here. The larger accumulation mode particles could be aged pollution particles that have been grown by $SO_2$ for example. The emissions of $SO_2$ have decreased, which would support this claim.

*"The results, which are presented in Fig. S3, pointed out that relatively greatest decrease occurred for accumulation mode particles that were 500 – 800 nm in diameter. On average, the volume size distribution of accumulation mode particles peaks around 300 nm (see Figs. S3 and S4) so the greatest decrease occurred at the larger sizes of the accumulation mode. The decrease in this size range might be caused by decrease in long-range transported pollution. Aged pollution particles might be grown by other substances, such as $SO_2$ in the atmosphere so their sizes are larger than freshly emitted or formed particles. $SO_2$ emissions have decreased in Europe (Tørseth et al., 2012), which supports this assumption. A trajectory analysis by Virkkula et al. (2011) showed that $\alpha_{sca}$ was clearly higher in air masses from continental Europe than from the North Atlantic and but also that the highest $\alpha_{sca}$ values were measured in air masses sources from within southern Finland, which would suggest that larger particles are not from nearby the station."*

**Pg. 29, Fig. 5: The text in the legends are very small. This may, however, be acceptable to the technical editor.**

I have modified the figure and also made the legends bigger.

**Pg. 31, Fig. 7: Why are there breaks in the whiskers and some whiskers not attached to the boxes? Is this a plotting artifact or is additional explanation necessary as to what the whiskers are meant to display?**

I fixed this. The whiskers were drew with dashed lines, which caused them to be not attached to the boxes like you noticed in your next comment.

**Pg. 33, Fig. 9: It appears that the whiskers are drawn as dashed lines with relatively long dashes and breaks. These should either be changed to solid lines or else changed to broken lines with smaller breaks in them.**

Fixed this by drawing the whiskers with solid lines.

**Pg. 35, Fig. 11: Caption '…1000 grid points in total.' Should this be '10,000 grid points in total.'?**

Yes it should! I fixed the number.

---

## Author Comment (AC2) · 1 Apr 2019

**Over a ten-year record of aerosol optical properties at SMEAR II**

Krista Luoma[1], Aki Virkkula[1,2], Pasi Aalto[1], Tuukka Petäjä[1] and Markku Kulmala[1]

[1]Institute for Atmospheric and Earth System Research, University of Helsinki, Helsinki, 00014, Finland
[2]Finnish Meteorological Institute, Helsinki, 00560, Finland

5   *Correspondence to*: Krista Luoma (krista.q.luoma@helsinki.fi), Aki Virkkula (aki.virkkula@fmi.fi)

**Abstract.** The aerosol optical properties (AOPs) describe the ability of aerosols to scatter and absorb radiation at different wavelengths. Since the aerosol particles interact with the radiation from the sun, they also have an impact on the climate. Our study focuses on the long-term trends and seasonal variations of different AOPs measured at a rural background station in Northern Europe. To explain the observed variations in the AOPs, we also analyzed changes in the aerosol size distribution.

10   AOPs of particles smaller than 10 µm (PM10) and 1 µm (PM1) have been measured at SMEAR II, in Southern Finland, since 2006 and 2010, respectively. For the PM10 particles the mean values of the scattering and absorption coefficients, single-scattering albedo, and backscatter fraction at $\lambda$ = 550 nm were 15.2 Mm$^{-1}$, 2.1 Mm$^{-1}$, 0.87 and 0.14. The scattering and absorption Ångström exponents at the wavelength ranges 450–700 nm and 370–950 nm were 1.80 and 0.95, respectively. Statistically significant trends were found for example for the PM10 scattering and absorption coefficients, single-scattering

15   albedo, and backscatter fraction, and the slopes of these trends were -0.32 Mm$^{-1}$, -0.086 Mm$^{-1}$, 2.2·10$^{-3}$, and 1.3·10$^{-3}$ per year. The tendency for the extensive AOPs to decrease correlated well with the decrease in aerosol number and volume concentration. The tendency for the single-scattering albedo and backscattering fraction to increase indicates that the aerosol size distribution consist of less larger particles and that aerosols absorb relatively less light than before. The trends of the single-scattering albedo and backscattering fraction influenced the effective aerosol forcing efficiency, indicating that the

20   aerosol particles are scattering the radiation more effectively back into space.

**1 Introduction**

Aerosol particles directly affect the climate by scattering and absorbing the shortwave radiation from the sun (ARI, *aerosol–radiation interaction*) (Charlson et al., 1992). Aerosol particles can either have a warming or cooling effect on the climate, depending on the optical properties of the aerosol particles and the surface below the aerosol layer. 
[revised manuscript text omitted]

**2.2.2 Size distribution measurements**

In addition to the AOPsoptical measurements, measurements of the particle size distribution data were used in the analyses below. The size distributions measurements were conductedmeasured with a Twin Differential Mobility Particle Sizer (TDMPS) in the size range 3–1000 nm (Aalto et al., 2001) and a TSI Aerodynamic Particle Sizer (APS, Model 3321) in the size range 0.53–10 µm. In the overlapping range of the TDMPS and the APS the number concentrations from the TDMPS were used up to 700 nm. The TDMPS, APS, integrating nephelometer and the aethalometer are located in the same measurement building. The TDMPS and APS have their own individual measurement lines. In the TDMPS measurement line, there is an inlet that removes particles larger than 1 µm. There is no active drying system in the TDMPS sample line to prevent particle losses. However, the sheath flows, which are used in the TDMPS system, are dried (RH < 40 %) so the particles are sampled in dry conditions. In the APS measurement line there is a pre-impactor that removes particles larger than 10 µm. The APS has its own dryer that heats up the sample air to 40 °C. This temperature might evaporate some semivolatile compounds, for instance ammonium nitrate but this is mainly an issue of urban sites (e.g. Bergin et al. 1997), whereas at the forest site in Hyytiälä low-volatile organic compounds are common (Ehn et al., 2014). Nevertheless, semivolatile aerosol particles are typically secondary particles smaller than 1 µm in diameter so evaporation of them does not have a large effect on the APS measurements.

**2.3 Data processing**

The data used in this study were measured between 21 June 2006 and 31 December 2017. All the optical data were quality assured manually and averaged for 1 h periods. The aerosol hygroscopic growth is often significant when RH increases above ~45 ± 5% and therefore the World Meteorological Organization and Global Atmosphere Watch (WMO and GAW) recommend for aerosol monitoring stations to keep sample air RH lower than that. In the GAW guidelines of 2003 the recommendation was to maintain RH < 50% but the limit was later lowered to RH < 40% (WMO/GAW, 2003, 2016). In the present work, if the internal RH in any of the optical instruments exceeded 40 %, the data from that instrument were excluded from further analysis if not stated otherwise. Note that Virkkula et al. (2011) followed the earlier RH recommendation: they calculated AOPs using data measured at RH < 50%. In addition, they also presented results from data measured at all RH. This affects comparisons of the results presented in this work.

All the optical data were also converted from ambient conditions to the standard temperature and pressure (STP) conditions (1013 hPa, 0 °C).

**2.3.1 Scattering data**

Both total scattering and backscattering coefficients measured with the nephelometer were corrected for the truncation error according to Anderson and Ogren (1998). The truncation correction uses the Ångström exponent (see Sect. 2.3.3) calculated from the uncorrected data.

Sherman et al. (2015) presented a well documented analysis for determining the uncertainty of the different AOPs. They determined a total fractional uncertainty of 9.2 % and 8.9 % (8.0 % and 8.1 %) for PM10 (PM1) $\sigma_{sca}$ and $\sigma_{bsca}$.

To test, if excluding the moist data had a large effect on the AOPs and their trends, we included the periods of high humidity (RH > 40 %) in some of the analyses. However, in these cases we corrected the scattering data, which was flagged due to too high RH, to dry conditions by using the scattering enhancement factor $f$(RH). $f$(RH) describes the increase of $\sigma_{sca}$ with increasing RH

$$f(\text{RH}) = \frac{\sigma_{sca}(\text{RH})}{\sigma_{sca}(\text{RH= dry})}. \tag{1}$$

$f$(RH) is the ratio of $\sigma_{sca}$ measured at high RH and at dry conditions. The $f$(RH) can be described by empirical relationship

$$f(\text{RH}) = q\left(1 - \frac{\text{RH}}{100\,\%}\right)^{-\gamma}, \tag{2}$$

with a parametrization presented by Zieger et al. (2015) for aerosol particles measured at SMEAR II in summer. They determined mean values for $q$ and $\gamma$ that were $0.96 \pm 0.07$ and $0.24 \pm 0.07$ at red wavelength (450 nm), $1.01 \pm 0.05$ and $0.25 \pm 0.07$ at green wavelength (525 nm), and $1.01 \pm 0.05$ and $0.30 \pm 0.08$ at red wavelength (635 nm). We used this parametrization, when the RH was higher than 40 %. Zieger et al. (2015) presented parameterization for total scattering only so we did not correct the $\sigma_{bsca}$ to dry condition.

This parametrization was also used for calculating the radiative forcing efficiency (see Sect. 2.3.3) in ambient RH.

[revised manuscript text omitted]

10 Schmidhauser et al. (2010) observed about 30 % decrease in $b$ when the RH increased to 85 % at the Jungfraujoch measurement station. We used this observation as a linear approximation to estimate the how the $b$ changes with varying RH. The estimated $b$ was then used in calculating the $\beta$ for moist conditions. The seasonally averaged RH was determined from RH measurements conducted at the height of 16 m. The lowest mean RH occurred in May (~62 %) and the highest in November (~95 %).

15 The estimated uncertainties for the intensive AOPs are presented in Sect. S3 in the supplementary material. The uncertainties were calculated according to Sherman et al. (2015).

**2.3.4 Data coverage**

If averaged over the whole measurement period, 81 % of the nephelometer data and 70 % of the aethalometer data were

20 considered valid. All the AOPs had some gaps in the data (see Fig. 1). Monthly data coverage of $\sigma_{sca}$ and $\sigma_{abs}$ are presented in Table S1. Most of the gaps in the time series of AOPs during the summers of 2006 to 2010 were due to too high RH. The gap in 2010 was due to maintenance and installation of the dryers and the switching inlet system. Some additional $\sigma_{bsca}$ data were missing, due to malfunction of the backscatter shutter of the integrating nephelometer. Dirty optics, malfunctions and maintenance caused the gaps in the $\sigma_{abs}$ data in 2012 and 2015.

Until March 2010, the integrating nephelometer and the aethalometer measured sample air that was not dried with any external dryers. During winter, the relative humidity (RH) remained below 40 %, since the sample air warmed up to room temperature (about 22 °C). Sometimes in summer, the RH of the sample increased to over the 40 % limit. If the RH was above 40 %, the data were flagged as invalid and they were omitted from the data analysis if not stated otherwise. About 25 % of all the data

30 before March 2010 had to be removed due to too high RH. Almost all of the removed data was from summer and fall months (June – October) and if regarding only these months, 46 % of the data were flagged. 
[revised manuscript text omitted]
 mean $\omega_0$ was 0.84 for Puijo tower (at 637 nm), 0.88 for SMEAR II (at 550 nm), and 0.92 for the remote Pallas station (at 550 nm). . Highest $\sigma_{sca}$ Pandolfi et al. (2018) observed in central and Eastern Europe.

The differences between the optical properties of the PM1 and PM10 particles are explained by the differences in concentrations, size distributions and chemical compositions. If only the PM10 data overlapping with the PM1 measurements were taken into account, the median values of $\sigma_{sca}$, $\sigma_{abs}$, $\omega_0$, $b$, $\alpha_{sca}$, $\alpha_{abs}$, $n$, and $k$ would have been 9.5 6 Mm$^{-1}$, 1.3 Mm$^{-1}$, 0.89, 0.15 14, 1.92, 0.96 97, 1.525 and 0.014 ($\sigma_{sca,}$, $\omega_0$, $b$, $\alpha_{sca}$, $n$ and $k$ at 550 nm, $\sigma_{abs}$ at 520 nm), respectively. The extensive variables ($\sigma_{sca}$, $\sigma_{bsca}$ and $\sigma_{abs}$) were smaller for the PM1 measurements, since there was less particle volume interacting with the radiation. Due to the differences in the median $\omega_0$ and $n$, the PM1 particles absorbed more light relative to scattering than the PM10 particles. The $\alpha_{sca}$ and $b$ are related to the sizes of the particles, so they were naturally different between the PM1 and PM10 particles. For the smaller PM1 particles, the $\alpha_{sca}$ and $b$ were larger than for the PM10 particles. However, $b$ does not have as large a difference between the PM1 and PM10 particles as $\alpha_{sca}$.

The average values of the PM10 particles given in Table 1 are calculated by excluding the periods when the RH > 40 %. If these periods of $\sigma_{sca}$ and $\sigma_{abs}$ measurements were included in the analysis and the moist scattering data were corrected to dry conditions by using the Eqs. 1 and 2, we would get median values of $\sigma_{sca} = 10.3$ Mm$^{-1}$, $\sigma_{abs} = 1.5$ Mm$^{-1}$, $\omega_0 = 0.88$, $b = 0.15$, $\alpha_{sca} = 1.91$, $\alpha_{abs} = 0.98$, and RFE$_{H\&S} = -23$ for PM10 ($\sigma_{sca,}$, $\omega_0$, $b$, and RFE$_{H\&S}$ at 550 nm, $\sigma_{abs}$ at 520 nm, $\alpha_{abs}$ at 370 nm/950 nm and $\alpha_{sca}$ at 450 nm/700 nm). The differences are not large compared to values presented in Table 1, so omitting the moist data periods from the data set does not seem to have a large effect on the median AOPs in this data set.

**3.2 Trends**

The long time series of the PM10 and PM1 AOPs were used to determine the trends for the optical properties. For the PM10 trend analysis we used data from about 10.5 years and for the PM1 trend analysis we used about 7.5 years long time series. The slopes of the trends and the trend statistics are presented in Table 3. The table also presents the trends as percentages, which were calculated by dividing the slope by the overall median value of the variable. The trends are also plotted in Fig. 1, where the monthly medians of the PM10 AOPs at SMEAR II used in this analysis are presented. The monthly medians are included in Fig. 1Fig. 1 only if the month had at least 14 days of valid data.

In the extensive properties, the trends were negative. The slopes of the trends for PM10 $\sigma_{sca}$, $\sigma_{bsca}$ and $\sigma_{abs}$ were -0.32, -0.038, and -0.088 -086 Mm$^{-1}$yr$^{-1}$, respectively. The decrease in the extensive properties were due to decrease in the total particle number concentration ($N_{tot}$) and total volume of the particles ($V_{tot}$) that can be seen in the combined TDMPS and APS data presented in Figs. 2a and bFigs. 2a and b and in Table 3. The relative decrease in $V_{tot}$ (-4 %yr$^{-1}$) was rather similar to that of $\sigma_{sca}$ (-3 %yr$^{-1}$). Also, Pandolfi et al. (2018) showed a statistically significant trend for $\sigma_{sca}$ (-0.588 Mm$^{-1}$yr$^{-1}$) measured at SMEAR II. They reported negative trends at other European sites as well and they determined that the average decrease was about -35 % for a ten-year period, which is a bit larger reduction than that observed at SMEAR II (-30 % for a ten-year period). The results are in line with the decrease in particle number concentration observed in European countries (Asmi et al., 2013). Also the remotely measured decreasing trend for aerosol optical depth ($\delta$) supports the decreasing trends in Europe (Li et al., 2014). Decreasing trends for $\sigma_{sca}$ are not only observed in Europe; Collaud Coen et al. (2013) and Sherman et al. (2015) reported negative trends for $\sigma_{sca}$ in North America as well.

The observed relative decrease in $\sigma_{abs}$ (-6 %yr$^{-1}$) was about twice as large than that of $\sigma_{sca}$ (-3 %yr$^{-1}$). The differences in the trends indicates that during the measurement period, the amount of absorbing material, such as BC and BrC, decreased relatively faster than the amount of scattering material (e.q. sulfate). It is also possible that the decrease in non-absorbing compounds decreased the $\sigma_{abs}$ since a non-absorbing coating around an absorbing particle can act as a lens, which increases absorption. The study by Collaud Coen et al. (2013), which included also $\sigma_{abs}$ data, observed negative trends for both $\sigma_{sca}$ and $\sigma_{abs}$ at the Bondville measurement station in Illinois, USA. There the trends of $\sigma_{sca}$ and $\sigma_{abs}$ were similar in magnitude (about -3 % yr$^{-1}$). Sherman et al. (2015) did not observe this decreasing $\sigma_{abs}$ trend later.

For the PM1 $\sigma_{abs}$, we observed a very steep decrease (-12 % yr$^{-1}$), which was probably caused by very high $\sigma_{abs}$ measured in January and February in 2012. Also, the data gaps in winter 2013 and 2015 could have affected the trends. The time series, of which the trends were determined for the PM1 measurements, were only 7.5 years long. Trends, which are determined for shorter time series, are more sensitive to year-to-year variability. This kind of extreme values can induce relatively large trends, which is why trend analysis for short time series (less than ten years) should be treated with caution.

Since the aerosol particles were absorbing less light than before, there was a tendency for the $\omega_0$ to increase. As shown by the increase in $\omega_0$ and the decrease in the extensive properties, the air measured at SMEAR II was less polluted than before. The higher $\omega_0$ indicates that the measurements were less affected by particles produced by traffic emissions or incomplete combustion. Li et al. (2014) reported mostly positive trends that were determined by remote measurements conducted in Europe. The decreasing trend for $k$ supports the tendency for $\omega_0$ to increase, since the negative trend for the imaginary part of $m$ means that particles absorb less light. The $\alpha_{abs}$, which is also related to the chemical composition of the particles, showed no significant trend for either the PM1 or PM10 particles. The negative trend for the interpolated $n$ was only significant for the PM1 particles. The tendency for the interpolated $n$ to decrease could have been caused by changes in the chemical composition.

The trends of the $b$ and $\alpha_{sca}$ were also investigated. These trends describe how the size distribution of the aerosol particles has changed. For the PM10 $b$ and $\alpha_{sca}$ the trends were positive, but for the PM10 $\alpha_{sca}$ however, the $p$ value was 0.07 so there was only a weak evidence for the positive trend in PM10 $\alpha_{sca}$. For the PM1 the trends for both, $b$ and $\alpha_{sca}$, were positive and statistically significant. Increasing $b$ and $\alpha_{sca}$ indicates that the mean size of the size distribution was moving towards smaller particles. The shift of in the size distribution towards smaller diameters is also observed in the negative trend of the volume mean diameter (VMD$_{tot}$), presented in Fig. 2c and in Table 3, supporting the increase in $b$ and $\alpha_{sca}$.

Also, Pandolfi et al. (2018) observed increasing trends for $b$ at SMEAR II and other European stations. For the $\alpha_{sca}$, however, they observed both positive and negative trends at different stations. Pandolfi et al. (2018) suspected that the variation was caused by differing trends of the coarse and accumulation mode particle concentration. Li et al. (2014) observed negative trends for the $\alpha_{sca}$ across the Europe and they suggested the trends were caused by a decrease in fine particle emissions.

Since the trends of $b$ and $\alpha_{sca}$ for the PM10 and PM1 measurements were similar, the trends in $\alpha_{sca}$ and $b$ may indicate that the concentration of larger particles in the accumulation mode was decreasing, since a decrease in coarse particle concentration only could not cause the decreasing trend of PM1 $\alpha_{sca}$. The changes in the size distribution were investigated by determining a trend for each TDMPS and APS measurement channels. The results, which are presented in Fig. S4, pointed out that relatively greatest decrease occurred for accumulation mode particles that were 500 – 800 nm in diameter. On average, the volume size distribution of accumulation mode particles peaks around 300 nm (see Figs. S6 and S7) so the greatest decrease occurred at the larger sizes of the accumulation mode. The decrease in this size range might be caused by decrease in long-range transported pollution. Aged pollution particles might be grown by other substances, such as $SO_2$ in the atmosphere so their sizes are larger than freshly emitted or formed particles. $SO_2$ emissions have decreased in Europe (Tørseth et al., 2012), which supports this assumption. A trajectory analysis by Virkkula et al. (2011) showed that $\alpha_{sca}$ was clearly higher in air masses from continental Europe than from the North Atlantic and but also that the highest $\alpha_{sca}$ values were measured in air masses sources from within southern Finland, which would suggest that larger particles are not from nearby the station.

The installation of the Nafion-dryers in 2010 could have caused an artificial decrease in $\sigma_{sca}$ or $\sigma_{abs}$ since the dryers increase the deposition of the particles and may decrease the sizes of hygroscopic particles. However, the trends were similar for the PM10 and PM1 particles. During the PM1 measurements, there were no large changes in the measurement line, so the observed trends were probably not caused by any technical changes in the measurement line.

A lot of summer time data measured before 2010, were marked invalid due to too high humidity and it could have affected the trend analysis. To test this hypothesis, we used Eqs. 1 and 2 to correct the $\sigma_{sca}$ to dry conditions and included this data in the trend analysis. The $\sigma_{bsca}$ was not corrected to dry conditions. Also, moist (RH > 40 %) absorption data was included in this test. Including the originally omitted data in the trend analysis, we observed statistically significant ($p$-value < 0.05) trends for the PM10 $\sigma_{sca}$, $\sigma_{abs}$, $\omega_0$, and RFE with the slopes of -4 % yr$^{-1}$, -5 % yr$^{-1}$, 0.2 % yr$^{-1}$, and 0.5 % yr$^{-1}$ respectively. Still, there were decreasing trends for extensive properties and positive trends for $\omega_0$. However the difference between the $\sigma_{sca}$ and $\sigma_{abs}$ trends decreased from 3 % to 1 % if compared against the trends that were determined only for the dry conditions. Including the moist data and acquiring longer data sets in the trend analysis suggests that the relative difference between the trends of $\sigma_{sca}$ and $\sigma_{abs}$ might not be that large. Not correcting the $\sigma_{bsca}$ to dry conditions probably explains why we do not see a significant trend for the $b$ here.

In addition to the general trends, we also investigated how the trends of $\sigma_{sca}$ and $\sigma_{abs}$ varied between different seasons. In this analysis, the periods of RH > 40 % were included ($\sigma_{sca}$ corrected to dry conditions according to Eqs. 1 and 2) in order to avoid the data gaps in summer and autumn before 2010. The trends were determined separately for spring (March, April, May), summer (June, July, August), autumn (September, October, November), and winter (December, January, February). The trend calculations were conducted by using the monthly medians (see timeseries in Fig. S3). The results are presented in Table 4.

Table 4 shows that $\sigma_{sca}$ and $\sigma_{abs}$ had a decreasing trend for each season, but for the autumn the trends were not significant. Both $\sigma_{sca}$ and $\sigma_{abs}$ experience the fastest absolute decrease in winter when the energy consumption is the highest and pollution sources are more pronounced; on the opposite, the trends are the least negative in summer. In spring, the absolute trends were less negative than compared to winter. However, for the $\sigma_{abs}$ we observed that the relative trend in spring (-9 % yr$^{-1}$) was steeper than in winter (-8 % yr$^{-1}$).

**3.3 Aerosol optical properties and size distribution**

To obtain a better view on how the shape of the size distribution affected the AOPs, the various AOPs were compared against the GMD and VMD. The results of the comparison are shown in Fig. 3. The GMD was mostly affected by the small nucleation

and Aitken mode particles, which are high in number concentration; the accumulation mode particles also had some effect on the GMD. Since only the smallest particles affect the GMD it is practically the same for the fine ($D_p < 1$ µm) and total ($D_p < 10$ µm) particle size distribution. Thus we present the comparison of GMD and AOPs only for the PM10 particles (Figs. 3a – d). The VMD, however, was heavily affected by the size distribution of the accumulation and coarse mode particles, since they predominated in the particle volume size distribution. This explains why there was notable differences for the PM10 (Figs. 3e – h) and PM1 (Figs. 3i – l) particles, when their AOPs were compared against the VMD calculated for particles smaller than 10 µm (VMD$_{tot}$) and VMD calculated for particles smaller than 1 µm (VMD$_{fine}$), respectively.

The $\sigma_{sca}$ correlated positively with the GMD due to the changes in particle concentration in the accumulation mode. The median number and volume size distribution for situations when GMD was below 50 nm or above 100 nm are presented in Fig. 4c. There was a clear difference in the number and volume size distribution in the accumulation mode when the GMD limit was varied. From the number size distribution, it can be seen that GMD increased due to a larger accumulation mode and lack of particles in the nucleation and Aitken modes. Nucleation and Aitken mode particles are mainly produced and grown by condensing vapors and since larger particles in the accumulation mode act as a condensation sink for vapors, the smaller particle modes do not tend to exist when accumulation mode particles are present.

For the PM10 particles, there was a negative correlation between the $\sigma_{sca}$ and VMD, but when the coarse particles were ignored, i.e. for PM1 particles the correlation became positive. The negative correlation for the PM10 particles is caused by the changes in the accumulation and coarse mode particle concentration. This is shown in further detail in Fig. 4a, where the median volume size distribution is presented for situations in which the VMD$_{tot}$ > 1500 nm, 500 nm < VMD$_{tot}$ < 1000 nm and VMD$_{tot}$ < 500 nm. When the VMD$_{tot}$ was high, there was a strong coarse mode but the accumulation mode was clearly smaller than in the other situations. Even though the VMD$_{tot}$ was high, the lack of accumulation mode particles decreased the scattering. From Fig. 3aFig. 4a, it can be seen that the $\sigma_{sca}$ became maximal when the VMD$_{tot}$ was about 500–1000 nm. In this VMD range, the coarse mode was slightly smaller but the accumulation mode clearly increased, thus increasing the scattering. When the VMD$_{tot}$ < 500 nm, the coarse mode was almost completely missing that caused the $\sigma_{sca}$ to decrease, even though there was a large accumulation mode present.

Kulmala et al. (2016) estimated that fresh eBC particles observed at SMEAR II are in the size range of 80 – 120 nm. That estimate was calculated in a simplified way from the relationship between particle number concentrations and BCe concentrations. A better estimate is obtained from the size dependence of $\omega_o$. The darkest aerosol has $\omega_o < 0.6$ and GMD in the range of about 30 – 70 nm (Fig. 3b, 3f, and 3j). This has been shown to be the range of fresh BC (e.g., Kittelson, 1998; Casati et al., 2007; Zhang et al., 2008) which suggests the source of BC is not far, probably within some kilometers only.

The size-dependent properties $\alpha_{sca}$ and $b$ for PM10 acted rather differently when compared with the GMD and VMD$_{tot}$. When the GMD was higher, the $\alpha_{sca}$  increased with growing GMD (Fig. 3c), which  is in contrast with the expectation that the $\alpha_{sca}$ would decrease when the size distribution is dominated by larger particles. The observation that the $\alpha_{sca}$ increased with an increasing GMD is in line with the analyses made for AOPs and size distributions measured in

5    Guangzhou, China by Garland et al. (2008), at SMEAR II by Virkkula et al. (2011), and in Nanjing, China by Shen et al. (2018). To study the reasons behind this relationship we generated first unimodal size distributions with two geometric standard deviations GSD = 1.5 and 2.0 and calculated both $\sigma_{sca}$ and $\sigma_{bsca}$ at $\lambda$ = 450, 550, and 700 nm with the Mie code with $m$ = 1.517 + 0.19i and the $\alpha_{sca}$ and $b$ from them. For unimodal size distributions the $\alpha_{sca}$ decrease with increasing GMD as is shown by the lines in Fig. 3c.

10     Schuster et al. (2006) showed that the relationship may be the opposite for  bimodal size distribution. Schuster et al. (2006) explained this behavior by that adding a larger or coarse particle size mode to a fine particle mode that is inefficiently scattering  - for instance nucleation and Aitken mode particles – the larger mode contributes more efficiently to the Ångström exponent than the fine mode. The contribution of the particles smaller than 100 nm to GMD is larger than that

15   of the larger particle modes which leads to the observed relationship. To study this in more detail we generated also bimodal size distributions. The analysis presented in the supplement (S6) shows that the $\alpha_{sca}$ of bimodal size distributions can be calculated as a linear combination of the $\alpha_{sca}$ of the modes, weighted by the fractions of $\sigma_{sca}$ of the respective modes. This explains the increase of $\alpha_{sca}$ with growing GMD.

20   In addition, at SMEAR II the size distribution typically consist of not only two but  multiple modes (Dal Maso et al., 2005; Saarikoski et al., 2005) that explains the observed relationship. An additional qualitative analysis of this relationship is given in Fig. 4c, where the median number and volume size distributions are plotted for situations in which the GMD was < 50 nm and > 100 nm. By comparing these two situations, it can be seen that when the GMD > 100 nm the accumulation mode was much larger than when GMD < 50 nm. Since the coarse mode is rather similar for both cases, the $\alpha_{sca}$ varied due to changes

25   in the accumulation mode. For the $\alpha_{sca}$ and VMD, the correlation was negative (Fig. 3g) that supports the expectations. However, the $\alpha_{sca}$ measured for the PM10 particles was much higher than that modeled for the unimodal distributions, which can also be explained by the multiple modes of the real size distributions.

There was a negative correlation between the GMD and PM10 $b$ (Fig. 3d) as expected, but the correlation was rather weak. On the contrary, the correlation between the VMD$_{tot}$ and PM10  $b$ was slightly positive (Fig. 3h). The  negative correlation of $\alpha_{sca}$ with VMD$_{tot}$ and the  positive correlation of $b$ with VMD$_{tot}$ for the PM10 particles indicates that the

$\alpha_{sca}$ and $b$ were sensitive to different size ranges. The $\alpha_{sca}$ decreased when there are more coarse particles present, but for the $b$ the coarse particles seem to have no expected effect and the $b$ increased with increasing $VMD_{tot}$. Fig. 4a. shows that when the $VMD > 1500$ nm, the peak of $DV/dlogD_p$ in the accumulation mode was much lower and tilted towards the smaller diameters than compared to the situations where the $VMD < 1000$ nm. This is in line with Collaud Coen et al. (2007), who stated that in the Jungfraujoch data, $b$ was sensitive to particles smaller than 400 nm and that the sensitivity of the $\alpha_{sca}$ was at its maxima for particle diameters between 500 and 800 nm.

For the PM1 particles, the measured $\alpha_{sca}$ and $b$ were well in line with the modeled values (Figs. 3k and l), since the coarse mode particles were removed prior to the measurements, the shape of the size distribution was closer to a unimodal size distribution, and the $VMD_{fine}$ described better how the accumulation mode shifted.

**3.4 Seasonal variation**

[revised manuscript text omitted]

25 The seasonal variation in $\alpha_{sca}$ and $b$ depends on the seasonal variation in the size distribution of the particles. Both $\alpha_{sca}$ and $b$ were maximal in summer and minimal in winter, suggesting that in summer, the particle population consisted of smaller particles than in winter. Closer investigation on the size distribution, which is presented in Fig. S6 and S7, reveals that in winter, the $\text{VMD}_{tot}$ was experiencing it minimum due to a lack of coarse mode particles. This is in contrast with the observation or smaller $\alpha_{sca}$ and $b$. In fact, the seasonal variation of $\alpha_{sca}$ and $b$ was explained by the seasonal variation of accumulation mode

30 and $\text{VMD}_{fine}$, which is a good indicator for the shifting accumulation mode. In winter, the accumulation mode was shifted towards larger sizes and the median of $\text{VMD}_{fine}$ was about 350 nm. In summer the situation was the opposite and $\text{VMD}_{fine}$ was about 250 nm.

**3.5 Variation between the PM10 and PM1 measurements**

[revised manuscript text omitted]

The evolution of the PM1/PM10 ratios were also investigated but we observed no statistically significant trends for either $\sigma_{sca}$ or $\sigma_{abs}$.

**3.6 Radiative forcing efficiency**

For the aerosol radiative forcing efficiency (RFE) the mean values, trends, and seasonal variation were also investigated. The statistics of the $RFE_{H\&S}$, $RFE_S$ and $RFE_{S,moist}$ are presented in Table 1 and their time series and seasonal variation are presented in Figs. 7a and b7.

In general, the aerosols, measured at SMEAR II, tend to have a cooling effect on the climate (RFE < 0) as seen in Table 1. By using the global average values suggested by Haywood and Shine (1995), the mean $RFE_{H\&S}$ was -22 Wm$^{-2}$. This is about 12 % less negative than the mean $RFE_{H\&S}$ (about -25 Wm$^{-2}$) determined by Sherman et al. (2015) for different North American stations. The difference is explained by higher mean $\omega_0$ (about 0.91) observed by Sherman et al. (2015), the mean $b$ (about 0.14) was similar if compared to average values observed at SMEAR II. Also, a mean $RFE_{H\&S}$ -25 Wm$^{-2}$ was determined at SORPES in China (Shen et al., 2018). Shen et al. (2018) observed a notably higher mean $\omega_0$ (0.93 at 520 nm) than what we observed at SMEAR II (0.87), but for the $b$ the situation was the opposite and it was lower at SORPES (0.12 at 525 nm) than at SMEAR II (0.14 at 550 nm). This would suggest that for dry particles the variation of $\omega_0$ is more pronounced than the variation of $b$ in context of calculating the $RFE_{H\&S}$. This is also observed at SMEAR II (Fig. S9). If the seasonal variation of $D$, $A_C$, and $R_S$ were taken into account, the mean $RFE_S$ (-34 Wm$^{-2}$) was more negative than $RFE_{H\&S}$.

Both, the $\omega_0$ and $b$ tended to increase, which makes the $RFE_{H\&S}$ to decrease (i.e., become more negative). The decreasing $RFE_{H\&S}$ means that the properties of dry aerosol particles have changed so that they cool the climate more efficiently. The trends for the $RFE_{H\&S}$, $RFE_S$ and $RFE_{S,moist}$ are presented in Table 3 as well. Since we used seasonal averages in calculating the $RFE_S$ and $RFE_{S,moist}$, their trends are also depended only on the changes of the $\omega_0$ and $b$ and thus their trends are also decreasing and similar in magnitude as the trend for $RFE_{H\&S}$. However, in reality the trend of RFE does not depend only on the $\omega_0$ and $b$. For example, a decrease in the snow cover due to global warming would decrease the $R_S$ and make the decrease of RFE steeper. Here, we omitted further analysis on the effect that the changes of $A_C$, $R_S$, $T_{at}$ and RH have on the RFE.

The seasonal variation in the $RFE_{H\&S}$ followed the seasonal cycles of the $\omega_0$ and $b$. The $RFE_{H\&S}$ was minimal in summer and maximal in winter. Since $b$ was lowest (forward-scattering particles) and the $\omega_0$ is also low (dark particles) in winter, the particles clearly did not have as strong a cooling effect as in summer when particles are smaller and light colored. If the seasonal changes of $D$, $A_C$, and $R_S$, were taken into account, the seasonal variability of $RFE_S$ is amplified remarkably compared to $RFE_{H\&S}$ as seen in Fig. 7b. In winter, the $D$ is lower and the $A_C$ is higher, which are shown in Fig. S1, causing the aerosol particles to have less effect (RFE closer to zero) than in summer. During winter the higher $R_S$ causes the aerosol particles to be less cooling or even warming. We chose to use the $R_S$ determined for a boreal forest according to the surroundings of SMEAR II. However, the area around the station consists also of fields and lakes, which in winter, would act as smooth snow fields. Even for snow containing impurities the $R_S$ is notably higher ($> 0.7$) than $R_s$ for snow covered boreal forest (Warren and Wiscombe, 1980). Using $R_S = 0.7$ for winter time data, would increase the $RFE_S$.

Taking the effect of RH into account increases the $\omega_0$ since the aerosols scatter more light due to hygroscopic growth. However, the same effect decreases the $b$ since the particles grow in size and scatter relatively less light backwards (Birmili et al., 2009). The seasonality of RH is presented in Fig. S1d and on average the RH is higher in winter than in summer. Fig. 7b shows that that the $RFE_{S,moist}$ is less negative in summer compared to $RFE_S$ since the effect of RH on $b$ overcomes the effect on $\omega_0$. Fierz-Schmidhauser et al. (2010) also observed this kind of behavior at the Jungfraujoch station. In winter the situation is the opposite and $RFE_{S,moist}$ is more negative than $RFE_S$. However, in winter, the effect of RH is small due to the small $D$ and large $A_C$. In general, the observed effect of the RH on RFE is smaller than the effect of taking the seasonal variation of $D$, $A_C$, and $R_S$ into account.

The RFE (or $\Delta F \delta^{-1}$) describes only the efficiency of the aerosol particles in cooling or warming the climate per unit of aerosol optical depth ($\delta$). Eq. 13 assumes that the properties of the aerosol particles are uniform in the atmospheric column that is rarely the case in reality. In ambient air, we should also take into account the variability in RH as a function of height. At the top of the boundary layer we typically have RH values close to 100 %. Here, we determined the RFE by using the RH measured near the ground (16 m). The simplified RFE does not give an absolute value for the aerosol forcing; however, it can still indicate how the changes in AOPs affect the climate.

Even if the RFE was very negative, the influence of aerosol particles on the climate would be small if the $\delta$ was small. The $\delta$ is highly dependent on the $\sigma_{sca}$ and $\sigma_{abs}$; the more there are scattering and absorbing material in the atmosphere, the higher the $\delta$. This is analyzed in further detail in Fig. 8, where the $\omega_0$ is presented as a function of $b$. In Fig. 8Fig. 12 the $RFE_{H\&S}$ is presented with isolines and the $\sigma_{sca}$ is presented by color-coding. Fig. 8Figure 12 shows that when the $RFE_{H\&S}$ is most negative, the median $\sigma_{sca}$ is actually experiencing its lowest value. When the $RFE_{H\&S}$ is closest to zero, the median $\sigma_{sca}$ is the highest. It

is also seen that when the $b$ is high and the particle size distribution consists of smaller particles, the particles are most efficient at cooling the atmosphere even though the average $\omega_0$ is the lowest.

These relationships were also observed in a study of AOPs at the Station for Observing Regional Processes of the Earth System (SORPES), a measurement station in Nanjing China (Shen et al., 2018). Also, Delene and Ogren (2002) and Sherman et al. (2015) observed similar systematic variability between $\sigma_{sca}$, $\omega_0$, $b$, and $RFE_{H\&S}$ at several North American measurement stations; when the $\sigma_{sca}$ increases, the $\omega_0$ increases and the $b$ decreases. Sherman et al. (2015) suggested that this variability could be caused by deposition of larger particles, which typically absorb less light. Delene and Ogren (2002) observed that $RFE_{H\&S}$ increases (i.e. becomes less negative) with increasing $\sigma_{sca}$, but Sherman et al. (2015) did not observe this trend.

**4 Summary and cConclusions**

In this study, we presented over 11-year long time series of AOPs measured at SMEAR II, a station in southern Finland. With the long time series, it was possible to see statistically significant trends, seasonal variation, and different types of causalities between the optical properties. We compared the AOPs with the aerosol size distribution measurements conducted at the station and observed in detail how the AOPs are dependent on the shape of the size distribution. By comparing the AOPs and size distribution, we were able to determine the $m$ values that can be used in modeling the $\sigma_{sca}$ and $\sigma_{abs}$ from size distribution measurements.

The extensive AOPs, as well as the aerosol number and volume concentration, tended to decrease. Our observation was in line with the other studies conducted in Europe and North America that also observed decreasing trends for the extensive AOPs (Collaud Coen et al., 2013; Pandolfi et al., 2018; Sherman et al., 2015), number concentration (Asmi et al., 2013) and aerosol optical depth (Li et al., 2014). This uniform decreasing trend in the amount of aerosol particles suggests that the anthropogenic emissions of particulate matter and gases that take part in secondary aerosol formation has been decreasing in Europe and North America. The observed tendency for $b$ and $\alpha_{sca}$ to increase together with the decreasing extensive properties indicated that the particle size distribution was moving towardsconsisted of less larger particles smaller diameters. A more detailed investigation revealed that the number of larger accumulation mode particles decreased relatively the fastest, which also supports the assumed decrease in pollution.

There are were clearly seasonal variations in the AOPs. The largest differences occur during summer and winter. The seasonal variations in the extensive properties and, $\omega_0$ and size distribution revealed that in winter the particles have a larger contribution from the anthropogenic sources than during summer.

Since the aerosol particles are smaller and less dark than before, their RFE tended to decrease (i.e. became more negative), which means that the ability of aerosols to cool the climate per unit $\delta$ increased. However, since the extensive properties and particle number concentration are decreasing, which means that the $\delta$ decreases as well, the total aerosol forcing is probably also decreasing. We determined the RFE to dry aerosol particles by using global average values suggested by Haywood and Shine (1995). To test the sensitivity of RFE to environmental parameters ($D$, $R_S$, and $A_C$), we calculated the RFE also by using more realistic and seasonally averaged environmental parameters. We also determined the RFE for ambient RH, since it is affected by the hygroscopic growth of aerosols. We observed that at SMEAR II the environmental parameters had a higher impact on the RFE than the ambient RH. Here we only studied the effect of AOPs on the RFE. Taking the long-term trend of environmental parameters into account would probably have a large effect on the trend of the RFE.

**Data availability**

[revised manuscript text omitted]
. Subplots a) – d) describe the correlation between the PM10 AOPs and GMD; subplots e) – h) describe the correlation between the PM10 AOPs and VMD$_{tot}$; and the subplots i) – l) describe the correlation between the PM1 AOPs and VMD$_{fine}$. The correlation coefficients of the linear regressions are given in each subfigure. The color-coding represents the number of data points in a grid point. In each subfigure, there are 100 grid points on both axes, making 10 000 grid points in total. The orange and black lines represent the values calculated from the unimodal size distributions, which were generated for different GMDs with geometric standard deviation GSD = 2.0 and 1.5 nm. The scattering was modeled from the generated size distribution at wavelengths 450, 550, and 700 nm with a refractive index $m = 1.517 + 0.19i$.**

[Figure]

**Figure 4: Median volume and number size distributions for the various VMD and GMD limits. The median $b$ and $\alpha_{sca}$ for the VMD and GMD limits are given in each legend box. The vertical grid lines represent the typical diameter limits for the nucleation, Aitken, accumulation and coarse particle modes (same as in  ). a) Volume size distribution for different PM10 VMD$_{tot}$ limits. b) Volume size distribution for different PM1 VMD$_{fine}$ limits. c) Volume and number size distribution for different PM10 _and_ GMD limits. The c) subfigure also represents volume and number size distribution for different PM1 and GMD limits as well, since the GMD is practically the same for PM10 and PM1 particles.**

[Figure]

**Figure 5: Seasonal variation in the aerosol optical properties for PM10 particles. The boxes represent the 25th and 75th percentiles and the whiskers the 10th and 90th percentiles of the data. The orange line is the median and the mean is presented with a black circle.**

[Figure]

**Figure 6: Seasonal variation in the PM1/PM10 ratio for a) $\sigma_{sca}$ and b) $\sigma_{abs}$. The explanation for the boxplots are the same as in Fig. 5.**

[Figure]

**Figure 7: Variations in the different radiative forcing efficiencies at SMEAR II in 2006 – 2018. a) Time series of the RFE$_{H\&S}$, RFE$_S$, and RFE$_{S,moist}$. The monthly medians are presented if the month had at least 14 days of valid data. b) Seasonal variation of the RFE$_{H\&S}$, RFE$_S$, and RFE$_{S,moist}$ as overall monthly medians.**

[Figure]

**Figure 8: Relationships between $\omega_0$, $b$ and RFE$_{H\&S}$. The RFE$_{H\&S}$ is shown as the dashed isolines in the background. The boxes represent the data measured at SMEAR II and they are colored by the median $\sigma_{sca}$. The explanation for the boxplots is the same as in Fig. 5.**

---

## Author Comment (AC3) · 1 Apr 2019

**Over a ten-year record of aerosol optical properties at SMEAR II**

Krista Luoma[1], Aki Virkkula[1,2], Pasi Aalto[1], Tuukka Petäjä[1], Markku Kulmala[1]

[1]Institute for Atmospheric and Earth System Research, University of Helsinki, Helsinki, 00014, Finland
[2]Finnish Meteorological Institute, Helsinki, 00560, Finland

5  *Correspondence to*: Krista Luoma (krista.q.luoma@helsinki.fi), Aki Virkkula (aki.virkkula@fmi.fi)

**S1. Data coverage**

The data coverage for each month is presented in Table S1. The data coverage is presented separately for $\sigma_{sca}$ and $\sigma_{abs}$. The Table S1 shows clearly how the data coverage improved from the beginning of the measurements to 2017. The data was quality assured by the author. Data was invalidated if the instrument had mechanical problems or if the RH in the istrument exceeded 10    40 %.

**Table S1. Data coverage of the extensive AOPs. The data coverage is presented as percentages for each month.**

| Data coverage (%) | | Jan | Feb | Mar | Apr | May | Jun | Jul | Aug | Sep | Oct | Nov | Dec |
|---|---|---|---|---|---|---|---|---|---|---|---|---|---|
| **2006** | $\sigma_{sca}$ | - | - | - | - | - | 29 | 54 | 25 | 28 | 64 | 99 | 42 |
| | $\sigma_{abs}$ | - | - | - | - | - | 27 | 52 | 25 | 27 | 64 | 100 | 43 |
| **2007** | $\sigma_{sca}$ | 0 | 54 | 100 | 99 | 76 | 81 | 3 | 9 | 46 | 51 | 71 | 89 |
| | $\sigma_{abs}$ | 0 | 53 | 99 | 99 | 76 | 81 | 2 | 08 | 46 | 58 | 83 | 91 |
| **2008** | $\sigma_{sca}$ | 100 | 99 | 100 | 93 | 99 | 45 | 14 | 10 | 57 | 68 | 98 | 97 |
| | $\sigma_{abs}$ | 99 | 96 | 100 | 92 | 99 | 44 | 13 | 10 | 55 | 67 | 97 | 98 |
| **2009** | $\sigma_{sca}$ | 100 | 100 | 100 | 100 | 34 | 0 | 0 | 15 | 33 | 83 | 97 | 93 |
| | $\sigma_{abs}$ | 97 | 99 | 100 | 100 | 38 | 0 | 0 | 14 | 32 | 85 | 98 | 89 |
| **2010** | $\sigma_{sca}$ | 100 | 97 | 28 | 0 | 0 | 76 | 93 | 92 | 100 | 15 | 0 | 56 |
| | $\sigma_{abs}$ | 86 | 80 | 28 | 0 | 0 | 77 | 92 | 90 | 100 | 15 | 0 | 57 |
| **2011** | $\sigma_{sca}$ | 96 | 98 | 87 | 79 | 84 | 90 | 31 | 85 | 100 | 84 | 100 | 97 |
| | $\sigma_{abs}$ | 76 | 0 | 74 | 100 | 98 | 100 | 63 | 100 | 100 | 85 | 100 | 92 |
| **2012** | $\sigma_{sca}$ | 98 | 99 | 99 | 100 | 100 | 97 | 98 | 100 | 98 | 100 | 100 | 93 |
| | $\sigma_{abs}$ | 98 | 100 | 99 | 100 | 82 | 28 | 0 | 0 | 0 | 0 | 0 | 0 |
| **2013** | $\sigma_{sca}$ | 97 | 51 | 79 | 100 | 100 | 100 | 100 | 100 | 100 | 100 | 100 | 96 |
| | $\sigma_{abs}$ | 0 | 0 | 79 | 100 | 100 | 100 | 100 | 100 | 99 | 99 | 99 | 89 |
| **2014** | $\sigma_{sca}$ | 77 | 92 | 98 | 100 | 100 | 100 | 100 | 100 | 100 | 100 | 100 | 97 |
| | $\sigma_{abs}$ | 70 | 91 | 98 | 100 | 100 | 99 | 100 | 100 | 100 | 99 | 100 | 0 |
| **2015** | $\sigma_{sca}$ | 100 | 100 | 100 | 100 | 100 | 100 | 100 | 100 | 100 | 99 | 41 | 69 |
| | $\sigma_{abs}$ | 0 | 0 | 0 | 0 | 89 | 95 | 100 | 99 | 100 | 91 | 43 | 69 |
| **2016** | $\sigma_{sca}$ | 100 | 100 | 100 | 100 | 100 | 100 | 100 | 84 | 100 | 100 | 100 | 97 |
| | $\sigma_{abs}$ | 94 | 100 | 100 | 99 | 99 | 100 | 100 | 87 | 99 | 100 | 100 | 100 |
| **2017** | $\sigma_{sca}$ | 100 | 100 | 100 | 92 | 100 | 100 | 100 | 98 | 100 | 100 | 100 | 38 |
| | $\sigma_{abs}$ | 100 | 100 | 99 | 65 | 95 | 99 | 99 | 96 | 100 | 100 | 81 | 14 |

[Figure]

**Figure S1: Seasonality of a) the fractional day length ($D$),  b) the surface reflectance ($R_S$), c) the cloud fraction ($A_C$), and d) the relative humidity (RH). In calculating the RFE$_S$ and RFE$_{S,moist}$, we used daily values for $D$ and $R_S$, and monthly means for $A_C$ and RH.**

**S2. Seasonal environmental variables for calculating the RFE**

The seasonal variability of the environmental parameters ($D$, $R_S$, $A_C$, and RH) used in calculating the seasonal radiative effective forcing (RFE$_S$ and RFE$_{S,moist}$) are presented in Fig. S1. The fractional daylength ($D$) was calculated for the latitude of 61°N and the seasonal variation of $D$ is presented in Fig. S1a. The surface reflectance ($R_S$) was determined by using the surface reflectance measurements by Kuusinen et al., (2012) and the seasonal variation of $R_S$ is presented in Fig. S1b. The cloud fraction was measured ($A_C$) by a ceilometer that was deployed to a nearby airport that is located about 25 km from SMEAR II. The monthly means were calculated by using data from 2010 to 2017 and the seasonal variation is presented in Fig. S1c. The relative humidity (RH) measurements were conducted with a RH sensor (Rotronic model MP102H) at 16 m height at SMEAR II. We used measurements from 2012 to 2017 in calculating the monthly means that are presented in Fig. S1d. For the $D$ and $R_S$ we used daily values and for the $A_C$ and RH the monthly means were used in calculating the RFE

**S2. Aethalometer data processing**

**S2.1 Flow correction**

The flow reported by the Aethalometer was corrected by using the weekly flow measurements conducted at SMEAR II with a Gilian flow meter. For correcting the flow we used a three-month moving average of the measured flow. The corrected flow is presented in Fig. S2.

[Figure]

**Figure S2: The Aethalometer flow (Q) correction. The black circles represent the flow measurements that were conducted almost every week at SMEAR II. The gray line is the flow that was reported by the Aethalometer and the orange line represents the corrected flow that was used in the data analysis.**

**S2.2 Difference between correction algorithms**

In Table S2, we present values for PM10 AOPs that depend on the $\sigma_{abs}$. In Table S2, the absorption data was corrected with the correction algorithm that was suggested by Arnott et al., (2005) with a $C_{ref} = 3.688$ at $\lambda = 520$ nm in a similar manner to Virkkula et al. (2011). However, the results may vary from Virkkula et al., (2011), since we used a spot size correction and a flow correction.

By comparing Table S2 to the Table 1 in the main article, we see that there is no large difference between the measured $\sigma_{abs}$ at 520 nm. Since the $\sigma_{abs}$ is rather similar, there is no notable difference in $\omega_0$ either. There is a larger difference, however, in $\sigma_{abs}$ at other wavelengths. This causes the $\alpha_{abs}$ to be remarkably higher than the $\alpha_{abs}$ that was determined for data, which was corrected with the algorithm described in the main article. We also did the trend analysis for the data corrected with the algorithm by Arnott et al., (2005). The slope of the $\sigma_{abs}$ statistically significant trend was -0.085 $Mm^{-1}yr^{-1}$ (-6 %$yr^{-1}$), which was similar to the trend determined with the new algorithm by Collaud Coen et al., (2010).

**Table S2. PM10 AOPs derived from Aethalometer data that was corrected with the algorithm described by Arnott et al. (2005).**

| PM10 | $\lambda$ (nm) | mean ± SD | 1 % | 10 % | 25 % | 50 % | 75 % | 90 % | 99 % |
|---|---|---|---|---|---|---|---|---|---|
| $\sigma_{abs}$ (Mm$^{-1}$) | 370 | 3.3 ± 3.9 | 0.2 | 0.6 | 1.1 | 2.1 | 4.1 | 7.3 | 19.4 |
| | 520 | 2.1 ± 2.4 | 0.1 | 0.4 | 0.7 | 1.4 | 2.6 | 4.7 | 12.0 |
| | 950 | 1.0 ± 1.1 | -0.1 | 0.1 | 0.3 | 0.6 | 1.2 | 2.2 | 5.3 |
| $\omega_0$ | 450 | 0.88 ± 0.08 | 0.63 | 0.79 | 0.84 | 0.89 | 0.93 | 0.95 | 0.99 |
| | 550 | 0.87 ± 0.09 | 0.62 | 0.78 | 0.84 | 0.89 | 0.92 | 0.95 | 0.99 |
| | 700 | 0.85 ± 0.09 | 0.56 | 0.75 | 0.81 | 0.87 | 0.91 | 0.95 | 0.99 |
| $\alpha_{abs}$ | 370/520 | 1.30 ± 0.60 | 0.16 | 0.85 | 1.10 | 1.30 | 1.46 | 1.68 | 2.86 |
| | 370/950 | 1.36 ± 0.51 | 0.28 | 0.92 | 1.16 | 1.34 | 1.49 | 1.71 | 3.31 |
| | 470/950 | 1.43 ± 0.63 | 0.12 | 0.98 | 1.23 | 1.40 | 1.55 | 1.81 | 3.86 |
| RFE$_{H\&S}$ (Wm$^{-2}$) | 550 | -22 ± 8 | -33 | -29 | -27 | -23 | -19 | -15 | -3 |

**S3. Uncertainty analysis**

We determined the uncertainties for the intensive PM10 AOPs using the equations presented in the supplementary material by Sherman et al. (2015). The absolute and fractional uncertainties are presented in Table S4. Here we used fractional uncertainties of 9.2 %, 8.0 %, and 23 % for the PM10 $\sigma_{sca}$, $\sigma_{bsca}$, and $\sigma_{abs}$, respectively. Since the uncertainties depend on the measured values, we used the mean values presented in Table 1 of the main article.

**Table S3: Uncertainties for different intensive AOPs. Fractional uncertainty is the absolute uncertainty divided by the mean value of the AOP. The uncertainties for $\omega_0$, $b$, and RFE$_{H\&S}$ were determined at 550 nm. The uncertainty for $\alpha_{sca}$ was determined for the wavelength range 450–700 nm, and the uncertainty for the $\alpha_{abs}$ was determined for the wavelength range 370–950 nm.**

| | Absolute uncertainty | Fractional uncertainty (%) |
|---|---|---|
| $\Delta\omega_0$ | 0.018 | 2.1 |
| $\Delta b$ | 0.003 | 2.2 |
| $\Delta\alpha_{sca}$ | 0.044 | 2.5 |
| $\Delta\alpha_{abs}$ | 0.26 | 27.7 |
| $\Delta$RFE$_{H\&S}$ (Wm$^{-2}$) | 1.42 | 6.5 |

**S4. Seasonality of the trends**

Fig. S3 presents the time series of the $\sigma_{sca}$, $\sigma_{abs}$, $V_{tot}$, and $V_{fine}$ monthly medians separately for spring, summer, autumn, and winter. Fig. S3 reveals the the year-to-year variablity between different seasons and it seems that in winter the variation from the fitted trend line is the highest. This is probably due to changes in the meteorological conditions. For example, according to

the statistics provided by the FMI (FMI: http://ilmatieteenlaitos.fi/vuositilastot, in Finnish only, last access: 25 March 2019) winter 2008 (December 2007 – February 2008) was exeptionally warm and the air masses arriving to Finland were mostly from the South and South-West that explains the low concenrations. On the contrary, high concentrations were measured in winter 2010, which was according to the reports by the FMI notably colder than average. It also seems from Figs. S3c, h, and

5    g that the concentration in winter increased from 2006 to 2010 after which it started to decrase. For other seasons we do not observe this kind of variation.

We did a similar analysis for the seasonal trends of $V_{tot}$ and $V_{fine}$ as we did for the $\sigma_{sca}$ and $\sigma_{abs}$ in the main article. The results are presented in Table S3. For $V_{tot}$ we observed a significant decreasing trend for all seasons and for $V_{fine}$ we observed

10    significant trends for spring, summer, and winter. For the $V_{tot}$ the relative trends were rather similar for all the seasons; the relative trends of $V_{fine}$ had more variation between the seasons. The variation of $V_{fine}$ relative trends was similar to that of the $\sigma_{sca}$ and $\sigma_{abs}$; the trends were most negative in winter and spring, and least negative in summer. This analysis would suggest that the variation of the $\sigma_{sca}$ and $\sigma_{abs}$ seasonal trends was due to varying trends in fine particle concentration.

[Figure]

15    **Figure S3: Monthly median values of a) – d) $\sigma_{sca}$, e) – h) $\sigma_{abs}$, and i) – l) $V_{tot}$ (black) and $V_{fine}$ (gray). and their trends. If the trend was statistically significant, the line is uniform and if the $p$ value of the trend was > 0.05, the line is dashed.**

**Table S4: The seasonal trend for $V_{tot}$ and $V_{fine}$.**

| | Trend (yr$^{-1}$) | | Lower (yr$^{-1}$) | Upper (yr$^{-1}$) | p-value | Trend (yr$^{-1}$) | | Lower (yr$^{-1}$) | Upper (yr$^{-1}$) | p-value |
|---|---|---|---|---|---|---|---|---|---|---|
| **Spring** | -0.10 | -4 % | -0.20 | -0.04 | < 0.01 | -0.06 | -4 % | -0.15 | 0.00 | 0.07 |
| **Summer** | -0.11 | -3 % | -0.20 | -0.03 | < 0.01 | -0.07 | -3 % | -0.14 | -0.02 | < 0.01 |
| **Autumn** | -0.07 | -3 % | -0.11 | -0.02 | < 0.01 | -0.02 | -2 % | -0.07 | 0.01 | 0.23 |
| **Winter** | -0.11 | -4 % | -0.21 | -0.01 | < 0.05 | -0.09 | -5 % | -0.18 | -0.00 | < 0.05 |

**S5. The trend of size distribution**

The trend for the size distribution was determined by applying the seasonal Kendall test to each channel of the TDMPS and APS. The results are shown in Fig. S4 that presents statistically significant decreasing trends for most of the measurement channels. The relative trend was the most negative (about -5 % yr$^{-1}$) for particles that were about 500 – 800 nm in diameter. Fig. S6 shows that the particle volume size distribution typically has a peak around 200 – 400 nm so the largest decrease occurs on the larger side of the accumulation mode.

[Figure]

**Figure S4: Trend analysis for the size distribution. The solid line represents the average trend in percentages. The gray bars mark the size ranges, in which the trend was statistically significant (p-value < 0.05). The typical borders of the nucleation, Aitken, accumulation and coarse particle modes are marked with vertical lines.**

**S6. Scattering Ångström exponent in simulated bimodal size distributions**

Two sets of simulations were done so that in the first one the geometric standard deviations (GSD) of both modes were 1.5 and the number concentrations (N) of the small and large particle modes were $N_{small}$ = 1000 cm$^{-3}$ and $N_{large}$ = 10 cm$^{-3}$ , respectively. The small particle mode $GMD_{small}$ varied from 50 to 300 nm, the large particle mode was set constant to $GMD_{large}$ = 300 nm. In the second set we changed both the number concentrations and the widths of the modes: $N_{small}$ = 1000 cm$^{-3}$ and $N_{large}$ = 1 cm$^{-3}$ , and $GSD_{small}$ = 1.3 and $GSD_{large}$ = 2.0. We used the Mie code and the refractive index m = 1.517 + 0.019i and calculated $\sigma_{sca}$ at $\lambda$ = 450 nm, 550 nm and 700 nm and $\alpha_{sca}$ for both the full bimodal size distribution as a function of its GMD, and for the two modes separately as a function of the GMD of the modes. In addition, we calculated the mode-scattering-weighted average $\alpha_{sca}$ from

$$\alpha_{sca,swa} = \frac{\sum \sigma_{sca550,i}\, \alpha_{sca,i}}{\sigma_{sca550}} = \frac{\sigma_{sca550,small}\, \alpha_{sca,small} + \sigma_{sca550,large}\, \alpha_{sca,large}}{\sigma_{sca550}} \qquad (S1)$$

where $\sigma_{sca550,small}$, $\sigma_{sca550,large}$ and $\sigma_{sca550}$ are the scattering coefficients of the small particle mode, the large particle mode and the full size distribution, respectively at 550 nm and $\alpha_{sca,small}$ and $\alpha_{sca,large}$ the scattering Ångström exponents of the two modes. The results are shown in Fig S3. In both simulations $\alpha_{sca}$ first increases with growing GMD, reaches a maximum at GMD $\approx$ 130 nm – 150 nm and then starts decreasing. The small particle mode $\alpha_{sca,small}$ has values close to 4 for small GMDs and then it decreases as a function of increasing GMD in line with the expected relationship. The $\alpha_{sca,swa}$ follows very closely the $\alpha_{sca}$ which suggests that the latter can be calculated as a linear combination of scattering-weighted $\alpha_{sca}$ of modes. This also explains the increase of $\alpha_{sca}$ with growing GMD: for the smallest GMDs of the small particle mode the $\alpha_{sca,small}$ is high but since the fraction of $\sigma_{sca550,small}$ of total $\sigma_{sca550}$ is small, the contribution of $\alpha_{sca,small}$ is small.

[Figure]

**Figure S5: Simulated scattering Ångström exponent of bimodal size distributions as a function of the geometric mean diameter (GMD). The geometric standard deviations (GSD) and number concentrations of the modes were a) small particle mode: GSD = 1.5, N = 1000; large particle mode: GSD = 1.5, N = 10, b) small particle mode: GSD = 1.3, N = 1000; large particle mode: GSD = 2, N = 1. The small particle mode GMD varied from 50 to 300 nm, the large particle mode GMD = 300 nm.**

**S7. The seasonal variation of the size distribution**

Fig. S6 presents the mean aerosol particle volume size distribution, and the median $\alpha_{sca}$ and $b$ for different seasons. Fig. S7 presents the seasonal variation of the geometrical mean diameter (GMD), the volumetric mean diameter for fine particles ($D_p$ < 1 µm, $VMD_{fine}$) and the volumetric mean diameter for all particles ($D_p$ < 10 µm, $VMD_{tot}$). The seasonal variation of the size distribution helps interpreting the seasonal variation of $\alpha_{sca}$ and $b$ that are sentive to different size ranges.

[Figure]

**Figure S6:** Averaged volume size distribution for winter (December – February), spring (March – May), summer (June – August) and autumn (September – November). Also, the averaged $\alpha_{sca}$ and $b$ for the seasons are presented.

[Figure]

 **Figure S7:** The seasonal variation and statistics for a) the GMD, b) the VMD$_{fine}$, and c) VMD$_{tot}$.

**S8. Diurnal variation of AOPs**

The diurnal variations of AOPs at SMEAR II were also studied, as shown in Fig. S8. However, the diurnal variations were weak and not nearly as clear as the seasonal variation. This was expected, since SMEAR II is located in a rather remote area further away from anthropogenic activities. Since the meteorological conditions at the SMEAR II station vary widely from season to season, the daily variation was determined separately for spring (March–May), summer (June–August), autumn (September–November) and winter (December–February). The diurnal variation was similar to the PM1 particles, so we do not present that separately.

For the extensive properties, the daily variation was similar in spring and summer, when both the $\sigma_{sca}$ and $\sigma_{abs}$ experienced a decrease during the day. The plausible explanation is boundary layer mixing that dilutes the air. The diurnal variation in both the $\sigma_{sca}$ and $\sigma_{abs}$ was smallest in autumn. In winter, the $\sigma_{sca}$ was maximal before noon but it did not decrease significantly in the afternoon that is clearly different from the diurnal cycles of the $\sigma_{sca}$ in spring and summer. In winter, the variation was much weaker, which can be explained by the weaker solar radiation and consequently weaker boundary layer mixing. In winter, there were also more often temperature inversions that caused air pollutants to accumulate in the boundary layer. The maximum $\sigma_{abs}$ in winter was observed in the evening at about 18–20 local time, whereas the maximum $\sigma_{sca}$ was before noon. In winter, the extensive properties increase slightly during the day and decrease during the late night and early morning hours. The daily variation in the $\omega_0$ is contrasted with the variation in $\sigma_{abs}$ in every season. In spring, summer, and autumn $\omega_0$ was the highest during the day, while in winter $_{it}$ peaked in the early morning.

The clearest diurnal variation was seen for $\sigma_{abs}$, which had an effect on $\omega_0$ and $k$ that can be observed in Fig. S6. For $n$ the diurnal variation is barely visible, but it is the opposite to $k$. For the size depended properties $b$ and $\alpha_{sca}$, there is no daily variation whatsoever. For the $\alpha_{abs}$, there is no variation during the winter, but during other seasons, the $\alpha_{abs}$ experiences a small decrease during the daytime. The variation of the $\alpha_{abs}$ is strongest during the summer and during the other seasons the variation is rather small. In the summer, there is more organic material present that can condensate on BC particles and thus cause variation in the $\alpha_{abs}$.

[Figure]

**Figure S8: Diurnal variation of different optical properties for different seasons for PM10 particles. The solid black line represents the median value and the dashed lines are the 25th and 75th percentiles.**

**S9. Radiative forcing efficiency**

5  The relationship between the $RFE_{H\&S}$, $\omega_0$ and $b$ is shown in Fig. S9. It can be seen that the correlation with $\omega_0$ is much stronger than with $b$. This can be interpreted such that at SMEAR II, the $RFE_{H\&S}$ for dry particles was much more dependent on the chemical composition described by the $\omega_0$ and not as much on the size distribution described by $b$. This situation looks probably different if the ambient RH was taken into account. In the main results we saw that the $RFE_{S,moist}$ was less negative than $RFE_S$. So in the moist condition the variability in $b$ overcame the variability of $\omega_0$.

[Figure]

**Figure S9: RFE$_{H\&S}$ as a function of a) single-scattering albedo ($\omega_0$) and b) backscatter fraction ($b$) at $\lambda$ = 550 nm. The coloring indicates the concentration of the data points in a single grid point. In each figures, there are 100 grid points on both axes, making 10 000 grid points in total.**

---

## Editor Decision (ED1)

Editor's Review of manuscript acp-2018-981

GENERAL REMARKS

From the review by two referees and from my editor's review, the manuscript is still considered presenting high-relevance data and analyses of high interest for the research community. The authors have responded to the concerns raised by both referees, but still in an insufficient manner. I have sent the manuscript for a second review to one of the referees, who confirmed that many of the technical issues have been answered but the presentation quality is still considered poor.

Serious objections against publication arise still from the manner, the scientific results are presented. The core of the manuscript is the analysis of the long-term time series which is of high relevance. However, there is no red line to follow in this manuscript. The presentation of results oscillates between different foci which makes it hard to follow.

To help making the manuscript acceptable for publication, I suggest the following way forward:

(1) The time-series of PM10 data is well described but the reader has enormous difficulties identifying if the authors focus on dry or humid conditions. The confusion starts with the description of the sampling conditions in Section 2.2.1. Instead of combining all relevant information on sampling lines, aerosol drying etc. in this section the information is distributed among the Sections 2.2 and 2.3. This is in particular true for all topics related to relative humidity, humidity-driven particle growth and related impacts on aerosol optical properties.
Here, I suggest combining all information on particle sampling, including references to aerosol sampling at SMEAR, into Section 1. Already here it should be stated whether the manuscripts is focusing on dry or ambient aerosol conditions. Having said this, the specific treatment of samples at high RH conditions can be added. But it should be clear to the reader if the authors generally focus on dry or ambient conditions. The treatment of humid cases can then be introduced as special cases. Obviously, there have difficulties caused by the failed humidity control of the sampling during certain period, but the description of the difficulties and the resulting effects on the data analysis are presented at different positions of the manuscript. This needs to be presented in one section to allow the reader a clear assessment of the quality of data and deduced results.

(2) To which conditions (dry, humid) do the reported optical properties refer to? For instance, in Section 2.3, the authors start directly with the introduction of the humidity growth factor, but it is not explained why.
Here, a much clearer reasoning and description of the approach is needed. Having said this, it will be much clearer to follow the analyses. In particular, the discussion of the backscatter fraction needs to be removed at all since this factor was not corrected for humidity effects and thus treated completely different than the other optical properties. Thus, there is no way of comparing scattering and backscatter properties.

(3) The authors state correctly that trend analyses of time series of less than one decade duration have to be taken with care; see Section 3.2 on page 14. Why do the authors then present in detail the trend analysis of the PM1 time series? This is difficult to justify and contradicts with the statement made on trends from time series of less than 10 years duration.

Here, I suggest removing the PM1 data from the trend analysis but present them as comparison to the PM10 trend results. This would give less weight to the PM1 results in terms of trends but still allows showing the difference to the PM10 results.

(4) There are intensive properties discussed with respect to trends, but the trends are not significant. These facts need to be reflected in the description of the results; see Table 3 for quantities $\alpha_{scat}$, $\alpha_{abs}$, and refractive index real and imaginary parts. The interpretation of results needs to be softened since the trends are not sufficient.

(5) In the Figures there are clear statements missing to which fraction (PM10, PM1) the plots refer to. If the authors state at the very beginning on which fraction the manuscript is focusing at, these statements are no longer necessary.

(6) In Tables 1 and 2, the authors present statistical analyses of aerosol optical properties. Inspecting the tables in detail shows that there a large discrepancies between average values and median values. Such discrepancies are always a clear sign that Gaussian statistics is not applicable. I suggest reducing the statistical analysis to the robust analysis of median and percentiles. By doing this, any biases caused by extreme values are avoided and the results are much more stable.

I am well aware that this further major revision may cause another large amount of work. On the other hand the paper has great potential and the trend analysis is of high interest for ACP but the presentation of the material requires a much clearer discussion of different topics and analysis results to improve overall readability and the delivery of a clear scientific goal/interest.

In summary, I encourage the authors to undertake this effort and accept the additional burden since the material is clearly worth it. I ensure further contribution and backing of the process to help publishing the manuscript. In case of questions or discussion the authors may contact me directly at a.petzold@fz-juelich.de.

Besides these general topics, I strongly encourage the consideration of the specific comments given by one referee in his 2nd review.

SPECIFIC COMMENTS BY REFEREE #2, arranged by section

1 INTRODUCTION

The section is incomplete and not well structured.

The importance of AOPs for the estimation of the RF is not mentioned. Thus, the climatic motivations at the base of this study are unclear. The aerosol-cloud interaction is, presently, not of primary need. The description of the measurements and site (P2L16-30) is not of necessary, since it can be extensively described in the method section. Generally, I also find a serious lack of references. An introduction on aerosol trends, such as previous works , environmental policies, dimming, and brightening might underline the importance of your work.

2 MEASUREMENTS AND METHODS

All measurements and data analysis are fully described. However, I would suggest to restructure and reorganize the different sections, starting from the titles. Try to be a more specific and attract the interest of the reader avoiding general titles such as "The field site", which could be changed into a

more appealing "The SMEAR II boreal research station" (by the way here you have to insert the description given at P2L16-21). EFR needs a separate section. Sections 2.3.4 and 2.5 can be merged.

**3.1 OVERVIEW OF THE DATA**

Work on the titles and try to be a bit more original: "Characterisation of the boreal aerosol", or "Scandinavian background aerosol optical properties", etc... . It is important to define a target-topic for each section, I hardly see what you want to show here. The seasonal analysis can be used to describe the impact of different sources or the role of atmospheric processes on the AOPs.

**3.2 TRENDS**

A bit of rework on the red line and additional thinking on the climatic/environmental implications are needed.

**3.3 AEROSOL OPTICAL PROPERTIES AND SIZE DISTRIBUTION**

As already indicated by me and the second reviewer, I do not understand the goal of this section. I would definitely give more priority to the interpretation of trends rather than to the size distribution. Moreover, too many variables are discussed making the section quite chaotic. Potentially, a reduced/simplified discussion on the aerosol size distribution can be introduced in 3.1 as part of the aerosol characterization.

**3.4 SEASONAL VARIATION**

As said before, I would move and merge it with 3.1

**3.5 VARIATION BETWEEN THE PM10 AND PM1 MEASUREMENTS**

As already argued in the first review, this section is of scarce interest. In fact, a good part of the section is used to justify the differences from previous works (P20L27-P21L7) and the high uncertainty (P21L21-27).

**3.6 RADIATIVE FORCING EFFICIENCY**

The section reads nice and might represent the final outcome of the manuscript. However, I have some suggestions:

a) The title can be improved: "Seasonality and trend of ".

b) The text is very intricated, limiting the understanding.

c) Figure 7 tells that humidity is very important in summer (hence hygroscopicity is a critical property of the aerosol in order to asses RF) and that optical properties have a smaller impact of RF compared to environmental factors. Considering these two messages and the high number of assumptions for the RFE calculation, Fig. 8 is quite approximative. A full investigation of the impact of AOPs on RF would need a full separated paper. Thus, I would suggest excluding, this time, Fig. 8 and the subsequent (very short) discussion.

**4 SUMMARY AND CONCLUSIONS**

As a consequence of all the above comments, the conclusion section needs rethinking and rewriting.

---

## Author Response (AR2)

**AUTHOR'S RESPONSE TO EDITOR'S REVIEW**

Manuscript: ACP-2018-981

Thank you again for commenting and spending time with this manuscript. We first reply on some major comments you both had and then, we reply to each of the comments separately. The comments by the editor and the referee are marked with bold font.

**1 MAJOR CHANGES**

To make to manuscript easier to read, we did rearranging and rewriting, we also removed part of the study according to your recommendations. All the changes are marked in red and you can find the marked up manuscript at the end of this document. The green color indicates that the place of the text had changed.

We made some changes to the sections: 1) We separated data processing from data analysis. 2) We combined the sections about seasonality of AOPs and the differences between the PM1 and PM10 and moved it as a subsection for the "Characterization of boreal aerosol particles". 3) We removed the size distribution analysis from the manuscript.

Now the sections are:
    1 Introduction
    2 Measurements and methods
        2.1 The boreal research station SMEAR II
        2.2 Instrumentation
            2.2.1. Measurements of AOPs
            2.2.2 Size distribution measurements
        2.3 Data processing
            2.3.1 Corrections for the integrating nephelometer data
            2.3.2 Corrections for the Aethalometer data
        2.4 Data analysis
            2.4.1 Intensive optical properties
            2.4.2 Aerosol radiative forcing efficiency
            2.4.3 Properties calculated from particle size distribution
            2.4.4 Data coverage and long-term trend analysis
    3 Results and discussion
        3.1 Characterization of boreal aerosol particles
            3.1.1 Seasonality of the AOPs
        3.2 Long-term trends of the AOPs
        3.3 Seasonality and long-term trend of the radiative forcing efficiency
        3.4 Effect of excluding the moist data
    4 Summary and conclusions

We emphasized in several parts of the manuscript that the data analysis was conducted for dry data and that the moist measurements were only taken into account in few special cases. We improved the discussion

about including the moist data in some special cases, which should make the difference between the dry and the moist data more clear.

**2 RESPONSE TO EDITOR'S COMMENTS**

**2.1 EDITOR**

**GENERAL REMARKS**

**From the review by two referees and from my editor's review, the manuscript is still considered presenting high-relevance data and analyses of high interest for the research community. The authors have responded to the concerns raised by both referees, but still in an insufficient manner. I have sent the manuscript for a second review to one of the referees, who confirmed that many of the technical issues have been answered but the presentation quality is still considered poor.**

**Serious objections against publication arise still from the manner, the scientific results are presented. The core of the manuscript is the analysis of the long-term time series, which is of high relevance. However, there is no red line to follow in this manuscript. The presentation of results oscillates between different foci, which makes it hard to follow.**

**To help making the manuscript acceptable for publication, I suggest the following way forward:**

**(1) The time-series of PM10 data is well described but the reader has enormous difficulties identifying if the authors focus on dry or humid conditions. The confusion starts with the description of the sampling conditions in Section 2.2.1. Instead of combining all relevant information on sampling lines, aerosol drying etc. in this section the information is distributed among the Sections 2.2 and 2.3. This is in particular true for all topics related to relative humidity, humidity-driven particle growth and related impacts on aerosol optical properties.**

We have now written all the parts that describe the sampling under section "2.2.1 Measurements of AOPs". We also improved the section 2.3.1, where we present the hygroscopic growth correction factor for scattering. We added there few sentences about the motivation why we use the f(RH) in the special cases.

**Here, I suggest combining all information on particle sampling, including references to aerosol sampling at SMEAR, into Section 1. Already here it should be stated whether the manuscripts is focusing on dry or ambient aerosol conditions. Having said this, the specific treatment of samples at high RH conditions can be added. But it should be clear to the reader if the authors generally focus on dry or ambient conditions. The treatment of humid cases can then be introduced as special cases. Obviously, there have difficulties caused by the failed humidity control of the sampling during certain period, but the description of the difficulties and the resulting effects on the data analysis are presented at different positions of the manuscript. This needs to be presented in one section to allow the reader a clear assessment of the quality of data and deduced results.**

Now all the information, which considers the sampling and instrumentation are in Sect. 2.2.1. We thought that maybe it is better that the introduction (Sect. 1) describes the motivation of the study rather than the sampling and the site. To underline that the data analysis was conducted for dry data, we added a sentence in Sect. 1, which states that we present the results for dry aerosol particles if not stated otherwise.

**(2) To which conditions (dry, humid) do the reported optical properties refer to? For instance, in Section 2.3, the authors start directly with the introduction of the humidity growth factor, but it is not explained why. Here, a much clearer reasoning and description of the approach is needed. Having said this, it will be much clearer to follow the analyses. In particular, the discussion of the backscatter fraction needs to be removed at all since this factor was not corrected for humidity effects and thus treated completely different than the other optical properties. Thus, there is no way of comparing scattering and backscatter properties.**

All the optical properties refer to dry conditions if not stated otherwise and we have now described this better in different sections. We moved the description of hygroscopic growth in to Sect. 2.2.1, which describes the sampling and explains why measurements of dry aerosol are preferred.

Instead of the parametrization presented by Zieger et al. (2015), we chose to use the parametrization by Andrews et al. (2006) presented in Sect. 2.3.1. Andrews et al. presented the parameters for backscattering as well. There was no drastic change in the results due to the different parameters used.

**(3) The authors state correctly that trend analyses of time series of less than one decade duration have to be taken with care; see Section 3.2 on page 14. Why do the authors then present in detail the trend analysis of the PM1 time series? This is difficult to justify and contradicts with the statement made on trends from time series of less than 10 years duration. Here, I suggest removing the PM1 data from the trend analysis but present them as comparison to the PM10 trend results. This would give less weight to the PM1 results in terms of trends but still allows showing the difference to the PM10 results.**

I still kept the results of PM1 trends in the text and in the Table 3 as a comparison, but I decreased their importance in the text. Their trends have the same sign as PM10 trends so they support the decreasing trend and show that the trends are not caused by changes in the measurement line (installation of the Nafion dryers in 2010).

**(4) There are intensive properties discussed with respect to trends, but the trends are not significant. These facts need to be reflected in the description of the results; see Table 3 for quantities $\sigma_{scat}$, $\sigma_{abs}$, and refractive index real and imaginary parts. The interpretation of results needs to be softened since the trends are not sufficient.**

We softened the interpretation of the trends that were not statistically significant.

**(5) In the Figures there are clear statements missing to which fraction (PM10, PM1) the plots refer to. If the authors state at the very beginning on which fraction the manuscript is focusing at, these statements are no longer necessary.**

We added statements of the particle fraction to figures, which were missing it.

**(6) In Tables 1 and 2, the authors present statistical analyses of aerosol optical properties. Inspecting the tables in detail shows that there a large discrepancies between average values and median values. Such discrepancies are always a clear sign that Gaussian statistics is not applicable. I suggest reducing the statistical analysis to the robust analysis of median and percentiles. By doing this, any biases caused by extreme values are avoided and the results are much more stable.**

We have corrected this in the text and in figures so that I always refer to the median values (only when comparing the RFE between different stations, we use the mean, because one of the studies did not report median values) . However, we prefer to keep the mean values in the table because it shows that the extensive AOPs do not follow the Gaussian statistics. We also added the following statement in the beginning of Sect. 3.1: *"Tables 1 and 2 show that for most of the variables the mean and the median values were quite different, which means that the data were not normally distributed. Therefore, we use the medians, which are not as sensitive to extreme values as the mean, to describe the characteristics of the AOPs."*

**I am well aware that this further major revision may cause another large amount of work. On the other hand the paper has great potential and the trend analysis is of high interest for ACP but the presentation of the material requires a much clearer discussion of different topics and analysis results to improve overall readability and the delivery of a clear scientific goal/interest.**

**In summary, I encourage the authors to undertake this effort and accept the additional burden since the material is clearly worth it. I ensure further contribution and backing of the process to help publishing the manuscript. In case of questions or discussion the authors may contact me directly at [a.petzold@fz-juelich.de](mailto:a.petzold@fz-juelich.de).**

**Besides these general topics, I strongly encourage the consideration of the specific comments given by one referee in his 2nd review.**

Thank you for the comments. We also took the comments given by the referee into account and the answers to those comments are presented below.

**2.2 REFEREE**

Comments of the referee are arranged by the old manuscript sections.

**2.2.1 Introduction**

**The section is incomplete and not well structured.**

**The importance of AOPs for the estimation of the RF is not mentioned. Thus, the climatic motivations at the base of this study are unclear. The aerosol-cloud interaction is, presently, not of primary need. The description of the measurements and site (P2L16-30) is not of necessary, since it can be extensively described in the method section. Generally, I also find a serious lack of references. An introduction on aerosol trends, such as previous works, environmental policies, dimming, and brightening might underline the importance of your work.**

We modified this section by doing some rearranging and by taking these comments into account. We added a mention of the RF and importance of knowing the AOPs in calculating this parameter. We modified the emphasis on the part considering the aerosol-cloud interaction, since it was not relevant. We moved the description of the sampling and part of the description of the station into Sect. 2. I still kept a part about SMEAR II, since it presents the motivation to study AOPs in a boreal forest. We also added references and text where we introduce long-term trends.

**2.2.2 Measurements and methods**

**All measurements and data analysis are fully described. However, I would suggest to restructure and reorganize the different sections, starting from the titles. Try to be a more specific and attract the interest of the reader avoiding general titles such as "The field site", which could be changed into a more appealing "The SMEAR II boreal research station" (by the way here you have to insert the description given at P2L16-21).**

We modified the titles and reorganized the sections (see Sect. 1 in this document). For example, we divided the data processing and analysis into separate sections.

**EFR needs a separate section.**

We fixed this.

**Sections 2.3.4 and 2.5 can be merged.**

We combined these two sections.

**2.2.3 Overview of the data**

**Work on the titles and try to be a bit more original: "Characterisation of the boreal aerosol", or "Scandinavian background aerosol optical properties", etc... . It is important to define a target-topic for each section, I hardly see what you want to show here. The seasonal analysis can be used to describe the impact of different sources or the role of atmospheric processes on the AOPs.**

We changed the title and also worked with the text. We combined here the seasonal variation of PM1/PM10.

**2.2.4 Trends**

**A bit of rework on the red line and additional thinking on the climatic/environmental implications are needed.**

We did also some rearranging and rewriting here to make the text more readable.

**2.2.5 Aerosol optical properties and size distribution**

**As already indicated by me and the second reviewer, I do not understand the goal of this section. I would definitely give more priority to the interpretation of trends rather than to the size distribution. Moreover, too many variables are discussed making the section quite chaotic. Potentially, a reduced/simplified discussion on the aerosol size distribution can be introduced in 3.1 as part of the aerosol characterization.**

We now removed this section. However, we added a figure about the seasonality of the size distribution, which shows that the $b$ and $\alpha_{sca}$ depend on the accumulation mode.

**2.2.6 Seasonal variation**

**As said before, I would move and merge it with 3.1**

We created a subsection for the 3.1.

**2.2.7 Variation between the PM10 and PM1 measurements**

**As already argued in the first review, this section is of scarce interest. In fact, a good part of the section is used to justify the differences from previous works (P20L27-P21L7) and the high uncertainty (P21L21-27).**

We integrated this section into Sect. 3.1.1.

**2.2.8 Radiative forcing efficiency**

**The section reads nice and might represent the final outcome of the manuscript. However, I have some suggestions:**

**a) The title can be improved: "Seasonality and trend of ".**

We fixed this.

**b) The text is very intricated, limiting the understanding.**

We have now improved the text and made the differences between the $RFE_{H\&S}$, $RFE_S$, and $RFES_{,moist}$ more clear and easier to understand.

**c) Figure 7 tells that humidity is very important in summer (hence hygroscopicity is a critical property of the aerosol in order to asses RF) and that optical properties have a smaller impact of RF compared to environmental factors. Considering these two messages and the high number of assumptions for the RFE calculation, Fig. 8 is quite approximative. A full investigation of the impact of AOPs on RF would need a full separated paper. Thus, I would suggest excluding, this time, Fig. 8 and the subsequent (very short) discussion.**

This is true and we removed this figure and discussion from the manuscript.

**2.2.9 Summary and conclusions**

**As a consequence of all the above comments, the conclusion section needs rethinking and rewriting.**

We rewrote parts of the conclusion.

**Over a ten-year record of aerosol optical properties at SMEAR II**

Krista Luoma[1], Aki Virkkula[1,2], Pasi Aalto[1], Tuukka Petäjä[1] and Markku Kulmala[1]

[1]Institute for Atmospheric and Earth System Research, University of Helsinki, Helsinki, 00014, Finland
[2]Finnish Meteorological Institute, Helsinki, 00560, Finland

5    *Correspondence to*: Krista Luoma (krista.q.luoma@helsinki.fi), Aki Virkkula (aki.virkkula@fmi.fi)

**Abstract.** Aerosol optical properties (AOPs) describe the ability of aerosols to scatter and absorb radiation at different wavelengths. Since  aerosol particles interact with the sun's radiation , they  impact  the climate. Our study focuses on the long-term trends and seasonal variations of different AOPs measured at a rural boreal forest site in Northern Europe. To explain the observed variations in the AOPs, we also analyzed changes in the

10    aerosol size distribution. AOPs of particles smaller than 10 µm (PM10) and 1 µm (PM1) have been measured at SMEAR II, in Southern Finland, since 2006 and 2010, respectively. For  PM10 particles, the mean values of the scattering and absorption coefficients, single-scattering albedo, and backscatter fraction at $\lambda = 550$ nm were 9.8 Mm$^{-1}$, 1.3 Mm$^{-1}$, 0.88 and 0.14. The median scattering and absorption Ångström exponents at the wavelength ranges 450–700 nm and 370–950 nm were 1.88 and 0.99, respectively. We found Sstatistically significant trends  for the PM10

15    scattering and absorption coefficients, single-scattering albedo, and backscatter fraction, and the slopes of these trends were -0.32 Mm$^{-1}$, -0.086 Mm$^{-1}$, 2.2·10$^{-3}$, and 1.3·10$^{-3}$ per year. The tendency for the extensive AOPs to decrease correlated well with the decrease in aerosol number and volume concentration. The tendency for the  backscattering fraction and single-scattering albedo to increase indicates that the aerosol size distribution consist of less larger particles and that aerosols absorb relatively less light than before. The trends of the single-scattering albedo and backscattering fraction

20    influenced the  aerosol radiative forcing efficiency, indicating that the aerosol particles are scattering the radiation more effectively back into space.

**1 Introduction**

Aerosols affect the radiative balance of the atmosphere both directly by aerosol–radiation interactions (ARI), i.e., by scattering and absorbing solar radiation and by absorbing and emitting terrestrial infrared radiation, and indirectly by aerosol–cloud

25    interactions (ACI), i.e., by influencing the properties and processes of clouds (Charlson et al., 1992; Lohmann and Feichter, 2005; Ramanathan et al., 2001; Stocker, 2013) The uncertainty of the estimated radiative forcing of climate by ACI is larger than that by ARI but also the latter is substantial (Stocker, 2013) Both ARI and

30    ACI have been shown to be responsible of dimming, the reduction of solar radiation received at the surface of the Earth (Wild,

2009, 2012; Stocker et al., 2013). Dimming and brightening have been shown to be often reconcilable with the trends in anthropogenic emissions of aerosols and their precursors and atmospheric aerosol loadings (Wild, 2012).

Aerosol optical properties (AOPs) describe the ability of aerosol particles to absorb and scatter radiation at different wavelengths. Knowing how aerosol particles interact with radiation is essential in determining the direct effect that aerosols have on the climate. Aerosol particles can either have a warming or cooling effect on the climate, depending on the optical properties of the aerosol particles and the surface below the aerosol layer. 
[revised manuscript text omitted]

Note that Virkkula et al. (2011) followed the earlier RH recommendation: they calculated AOPs using data measured at RH < 50%. In addition, they also presented results from data measured at all RH. This affects comparisons of the results presented

5   in this work.

To test, if excluding the moist data had a large effect on the AOPs and their trends, we included the periods of high humidity (RH > 40 %) in some of the analyses. However, in these cases we corrected the scattering data, which was flagged due to too high RH, to dry conditions by using the scattering enhancement factor $f$(RH). $f$(RH) describes the increase of $\sigma_{sca}$ with

10   increasing RH

$$f(\text{RH}) = \frac{\sigma_{sca}(\text{RH})}{\sigma_{sca}(\text{RH}=\text{dry})}.$$   (1)

$f$(RH) is the ratio of $\sigma_{sca}$ measured at high RH and at dry conditions. The $f$(RH) can be described by empirical relationship

$$f(\text{RH}) = q \left(1 - \frac{\text{RH}}{100\,\%}\right)^{-\gamma}.$$   (2)

with a parametrization presented by   for aerosol particles measured at SMEAR II in summer. They determined mean values

15   for $q$ and $\gamma$ that were $0.96 \pm 0.07$ and $0.24 \pm 0.07$ at red wavelength (450 nm), $1.01 \pm 0.05$ and $0.25 \pm 0.07$ at green wavelength (525 nm), and $1.01 \pm 0.05$ and $0.30 \pm 0.08$ at red wavelength (635 nm). We used this parametrization, when the RH was higher than 40 %.   presented parameterization for total scattering only so we did not correct the $\sigma_{bsca}$ to dry condition.

This parametrization was also used for calculating the radiative forcing efficiency (see Sect. 2.3.3) in ambient RH.

[revised manuscript text omitted]
. There has not been measurements of hygroscopic growth parameters ($q$ and $\gamma$) for $\sigma_{bsca}$, so we could not use the same parametrization in calculating the $b$ to ambient RH. observed about 30 % decrease in $b$ when the RH increased to 85 % at the Jungfraujoch measurement station. We used this observation as a linear approximation to estimate the how the $b$ changes with varying RH. The estimated

 The seasonally averaged RH was determined from RH measurements conducted at the height of 16 m. The lowest mean RH occurred in May (~62 %) and the highest in November (~95 %).

More information about the seasonal $D$, $A_C$, $R_S$, and RH can be found in the supplementary material Sect. S2.

~~If averaged over the whole measurement period, 81 % of the nephelometer data and 70 % of the aethalometer data were considered valid. All the AOPs had some gaps in the data (see Fig. 1). Monthly data coverage of $\sigma_{sca}$ and $\sigma_{abs}$ are presented in Table S1. Most of the gaps in the time series of AOPs during the summers of 2006 to 2010 were due to too high RH. The gap in 2010 was due to maintenance and installation of the dryers and the switching inlet system. Some additional $\sigma_{bsca}$ data were missing, due to malfunction of the backscatter shutter of the integrating nephelometer. Dirty optics, malfunctions and maintenance caused the gaps in the $\sigma_{abs}$ data in 2012 and 2015.~~

~~Until March 2010, the integrating nephelometer and the aethalometer measured sample air that was not dried with any external dryers. During winter, the relative humidity (RH) remained below 40 %, since the sample air warmed up to room temperature (about 22 °C). Sometimes in summer, the RH of the sample increased to over the 40 % limit. If the RH was above 40 %, the data were flagged as invalid and they were omitted from the data analysis if not stated otherwise. About 25 % of all the data before March 2010 had to be removed due to too high RH. Almost all of the removed data was from summer and fall months (June – October) and if regarding only these months, 46 % of the data were flagged. If the moist data was included the overall data coverage would increase to 89 % and 77 % for scattering and absorption data, respectively. After the installation of the Nafion dryers in March 2010, the humidity caused no further problems.~~

**2.4.3 Properties calculated from particle size distribution**

Size distributions _were used_ to calculate differently weighted mean diameters. In this study, we used the geometric mean diameter (GMD) and the volume mean diameter (VMD). The GMD is the mean diameter that is weighted by the number concentration (*N*)

$$\text{GMD} = \exp\left(\frac{\sum N_i \ln D_{p,i}}{\sum N_i}\right), \tag{15}$$

while and the VMD is weighted by the particle volume (*V*)

$$\text{VMD} = \frac{\sum D_{p,i} V_i}{V_{tot}} = \frac{\sum N_i D_{\text{p},i}^4}{\sum N_i D_{\text{p},i}^3}. \tag{16}$$

Since the particle number concentration is focused the highest onfor the nucleation and Aitken mode sparticles, the GMD describes the distribution changes in the smallest sizes. The VMD, in contrast, is affected by the changes in the accumulation and coarse mode, since they contribute the most to the volume size distribution.

The measurements of the AOPs and size distribution can be combined by determiningto determine the complex refractive index ($m = n + ik$) that describes how much the particles scatter and absorb light. The $m$ and can be used to model $\sigma_{\text{sca}}$, $\sigma_{\text{bsca}}$ and $\sigma_{\text{abs}}$ from the size distribution measurements. Index 
[revised manuscript text omitted]

3.3 Aerosol optical properties and size distribution

To obtain a better view on how the shape of the size distribution affected the AOPs, the various AOPs were compared against the GMD and VMD. The results of the comparison are shown in Fig. 3. The GMD was mostly affected by the small nucleation and Aitken mode particles, which are high in number concentration; the accumulation mode particles also had some effect on the GMD. Since only the smallest particles affect the GMD it is practically the same for the fine ($D_p < 1$ µm) and total ($D_p < 10$ µm) particle size distribution. Thus, we present the comparison of GMD and AOPs only for the PM10 particles (Figs. 3a – d). The VMD, however, was heavily affected by the size distribution of the accumulation and coarse mode particles, since they predominated in the particle volume size distribution. This explains why there was notable differences for the PM10 (Figs. 3e – h) and PM1 (Figs. 3i – l) particles, when their AOPs were compared against the VMD calculated for particles smaller than 10 µm (VMD$_{tot}$) and VMD calculated for particles smaller than 1 µm (VMD$_{fine}$), respectively.

The $\sigma_{sca}$ correlated positively with the GMD due to the changes in particle concentration in the accumulation mode. The median number and volume size distribution for situations when GMD was below 50 nm or above 100 nm are presented in Fig. 4c. There was a clear difference in the number and volume size distribution in the accumulation mode when the GMD limit was varied. From the number size distribution, it can be seen that GMD increased due to a larger accumulation mode and lack of particles in the nucleation and Aitken modes. Nucleation and Aitken mode particles are mainly produced and grown by condensing vapors and since larger particles in the accumulation mode act as a condensation sink for vapors, the smaller particle modes do not tend to exist when accumulation mode particles are present.

For the PM10 particles, there was a negative correlation between the $\sigma_{sca}$ and VMD, but when the coarse particles were ignored, i.e. for PM1 particles, the correlation became positive. The negative correlation for the PM10 particles is caused by the changes in the accumulation and coarse mode particle concentration. This is shown in further detail in Fig. 4a, where the median volume size distribution is presented for situations in which the VMD$_{tot} > 1500$ nm, 500 nm $<$ VMD$_{tot} < 1000$ nm and VMD$_{tot} < 500$ nm. When the VMD$_{tot}$ was high, there was a strong coarse mode but the accumulation mode was clearly smaller than in the other situations. Even though the VMD$_{tot}$ was high, the lack of accumulation mode particles decreased the scattering. From Fig. 3a, it can be seen that the $\sigma_{sca}$ became maximal when the VMD$_{tot}$ was about 500–1000 nm. In this VMD range, the coarse mode was slightly smaller but the accumulation mode clearly increased, thus increasing the scattering. When the VMD$_{tot} < 500$ nm, the coarse mode was almost completely missing that caused the $\sigma_{sca}$ to decrease, even though there was a large accumulation mode present.

estimated that fresh eBC particles observed at SMEAR II are in the size range of 80 – 120 nm. That estimate was calculated in a simplified way from the relationship between particle number concentrations and BCe concentrations. A better estimate is obtained from the size dependence of $\omega_0$. The darkest aerosol has $\omega_0 < 0.6$ and GMD in the range of about 30 – 70 nm

(Fig. 3b, 3f, and 3j). This has been shown to be the range of fresh BC (e.g., Kittelson, 1998; Casati et al., 2007; Zhang et al., 2008) which suggests the source of BC is not far, probably within some kilometers only.

The size-dependent properties $\alpha_{sca}$ and $b$ for PM10 acted rather differently when compared with the GMD and VMD$_{tot}$. The $\alpha_{sca}$ increased with growing GMD (Fig. 3c), which is in contrast with the expectation that the $\alpha_{sca}$ would decrease when the size distribution is dominated by larger particles. The observation that the $\alpha_{sca}$ increased with an increasing GMD is in line with the analyses made for AOPs and size distributions measured in Guangzhou, China by , at SMEAR II by , and in Nanjing, China by . To study the reasons behind this relationship we generated first unimodal size distributions with two geometric standard deviations GSD = 1.5 and 2.0 and calculated both $\sigma_{sca}$ and $\sigma_{bsca}$ at $\lambda$ = 450, 550, and 700 nm with the Mie code with $m$ = 1.517 + 0.19i and the $\alpha_{sca}$ and $b$ from them. For unimodal size distributions the $\alpha_{sca}$ decrease with increasing GMD as is shown by the lines in Fig. 3c. showed that the relationship may be the opposite for bimodal size distributions. explained this behavior by that adding a larger or coarse particle size mode to a fine particle mode that is inefficiently scattering for instance nucleation and Aitken mode particles the larger mode contributes more efficiently to the Ångström exponent than the fine mode. The contribution of the particles smaller than 100 nm to GMD is larger than that of the larger particle modes, which leads to the observed relationship. To study this in more detail we generated also bimodal size distributions. The analysis presented in the supplement (S6) shows that the $\alpha_{sca}$ of bimodal size distributions can be calculated as a linear combination of the $\alpha_{sca}$ of the modes, weighted by the fractions of $\sigma_{sca}$ of the respective modes. This explains the increase of $\alpha_{sca}$ with growing GMD.

In addition, at SMEAR II the size distribution typically consists of not only two but multiple modes that explains the observed relationship. An additional qualitative analysis of this relationship is given in Fig. 4c, where the median number and volume size distributions are plotted for situations in which the GMD was < 50 nm and > 100 nm. By comparing these two situations, it can be seen that when the GMD > 100 nm the accumulation mode was much larger than when GMD < 50 nm. Since the coarse mode is rather similar for both cases, the $\alpha_{sca}$ varied due to changes in the accumulation mode. For the $\alpha_{sca}$ and VMD, the correlation was negative (Fig. 3g) that supports the expectations. However, the $\alpha_{sca}$ measured for the PM10 particles was much higher than that modeled for the unimodal distributions, which can also be explained by the multiple modes of the real size distributions.

There was a negative correlation between the GMD and PM10 $b$ (Fig. 3d) as expected, but the correlation was rather weak. On the contrary, the correlation between the VMD$_{tot}$ and PM10 $b$ was slightly positive (Fig. 3h). The negative correlation of $\alpha_{sca}$ with VMD$_{tot}$ and the positive correlation of $b$ with VMD$_{tot}$ for the PM10 particles indicates that the $\alpha_{sca}$ and $b$ were sensitive to different size ranges. The $\alpha_{sca}$ decreased when there are more coarse particles present, but for the $b$ the coarse particles seem to have no expected effect and the $b$ increased with increasing VMD$_{tot}$. Fig. 4a. shows that when the VMD > 1500 nm, the peak of D$V$/dlog$D_p$ in the accumulation mode was much lower and tilted towards the smaller diameters than compared to the

situations where the VMD < 1000 nm. This is in line with , who stated that in the Jungfraujoch data, $b$ was sensitive to particles smaller than 400 nm and that the sensitivity of the $\alpha_{sca}$ was at its maxima for particle diameters between 500 and 800 nm.

For the PM1 particles, the measured $\alpha_{sca}$ and $b$ were well in line with the modeled values (Figs. 3k and l), since the coarse mode particles were removed prior to the measurements, the shape of the size distribution was closer to a unimodal size distribution, and the VMD$_{fine}$ described better how the accumulation mode shifted.

For the PM1 $\sigma_{abs}$, we observed a very steep decrease (-12 % yr$^{-1}$), which was probably caused by very high $\sigma_{abs}$ measured in January and February in 2012. Also, the data gaps in winter 2013 and 2015 could have affected the trends. The time series, of which the trends were determined for the PM1 measurements, were only 7.5 years long. Trends, which are determined for shorter time series are more sensitive to year-to-year variability. This kind of extreme values can induce relatively large trends, which is why trend analysis for short time series (less than ten years) should be treated with caution.

3.4 Seasonal variation

[revised manuscript text omitted]

The seasonal variation in $\alpha_{sca}$ and $b$ depends on the seasonal variation in the size distribution of the particles. Both $\alpha_{sca}$ and $b$ were maximal in summer and minimal in winter, suggesting that in summer, the particle population consisted of smaller particles than in winter. Closer investigation on the size distribution, which is presented in Fig. S6 and S7, reveals that in winter, the $VMD_{tot}$ was experiencing it minimum due to a lack of coarse mode particles. This is in contrast with the observation or smaller $\alpha_{sca}$ and $b$. In fact, the seasonal variation of $\alpha_{sca}$ and $b$ was explained by the seasonal variation of accumulation mode and $VMD_{fine}$, which is a good indicator for the shifting accumulation mode. In winter, the accumulation mode was shifted

towards larger sizes and the median of $VMD_{fine}$ was about 350 nm. In summer the situation was the opposite and $VMD_{fine}$ was about 250 nm.

3.5 Variation between the PM10 and PM1 measurements

5   Even though the average values between the optical properties of the PM10 and PM1 particles differed, their seasonal variation was similar for all the various properties. However, there was a seasonal variation in the relationship between the PM10 and PM1 extensive properties, as shown in Fig. 6. The seasonal variation in the PM1/PM10 ratio describes the impact of the coarse and fine particles on the $\sigma_{sca}$ and $\sigma_{abs}$.

10  For the $\sigma_{sca}$ the seasonal variation in the PM1/PM10 ratio was clear, but for the $\sigma_{abs}$ there seemed to be no seasonal variation in the ratio whatsoever. The seasonal medians of the PM1/PM10 ration for the $\sigma_{sca}$ varied from 0.7 to 0.8, and on average submicron particles caused about 75 % of the total scattering of the PM10 particles. This was apparently a lower fraction than in the previous analysis of SMEAR II scattering data. Virkkula et al. (2011) stated that the seasonal average contributions of submicron particles to the total $\sigma_{sca}$ was in the range of 88–92 %, clearly more than in the present work. However, in that study
15  the scattering size distribution and the contributions of the various size ranges were calculated from particle number size distributions with a Mie model and the physical diameters ($D_p$) were used whereas here the PM1 corresponds to particles smaller than the aerodynamic diameter $D_a$ of 1 µm. With particle density of 1.7 g cm$^{-3}$ this corresponds to the physical diameter $D_p = (1/1.7)^{\frac{1}{2}} \cdot 1$ µm ~ 0.77 µm. The contribution of particles smaller than 0.77 µm is approximately 85 % if it is estimated from Fig. 11 of Virkkula et al. (2011), still more than the ~ 75 % contribution of submicron scattering shown here. This may have
20  resulted from the cutoff diameter of the PM1 impactor is not exactly sharp and also that the particles entering the impactor may have still been somewhat moist and thus larger than their dry size and were therefore removed from the sample stream. Further analysis of the difference is omitted here.

The maxima of the submicron particle scattering occurred in winter and summer. The summer peak coincided with the maxima
25  of the PM10 $\alpha_{sca}$, which already indicates that smaller particles play a major role in the size distribution. However, this correlation between the PM1/PM10 ratio and $\alpha_{sca}$ was not observed in winter. In Fig. 2 (and in Fig. S7), it can be seen that the $VMD_{tot}$ always decreased in the wintertime indicating also the lack of coarse particles. However, on average, the accumulation mode is relatively large compared to the coarse mode and it is shifted towards the larger diameters. This is presented in the supplementary material (Figs. S6 and S7). 
[revised manuscript text omitted]

15

20 ~~and RFE$_{H\&S}$ at several North American measurement stations; when the $\sigma_{sca}$ increases, the $\omega_0$ increases and the $b$ decreases. Sherman et al. (2015) suggested that this variability could be caused by deposition of larger particles, which typically absorb less light.  observed that RFE$_{H\&S}$ increases (i.e. becomes less negative) with increasing $\sigma_{sca}$, but Sherman et al. (2015) did not observe this trend.~~

25 **4 Summary and conclusions**

In this study, we presented  11.5-yearlong time series of AOPs measured at SMEAR II, a  station in southern Finland.

30  Compared to regional and rural European sites, the $\sigma_{sca}$ at the boreal SMEAR II station was low. However, the average $\sigma_{sca}$ and $\sigma_{abs}$ were higher than those observed at other Finnish measurement stations that were the arctic station in Pallas

and the semi-urban station in Kuopio, Eastern Finland. Because of the more southern location, the SMEAR II was probably more affected by regional emissions and long-transport pollution from Europe than the other Finnish measurement sites, which would explain the higher concentrations.

5   The highest $\sigma_{sca}$ and $\sigma_{abs}$ were measured in winter when the boundary layer is lower and the pollution is not diluted as efficiently as in summer. Transported pollution from the regional area and from Europe, also increases the concentrations in winter, when the energy consumption is higher. In winter, the $\omega_0$ was low (i.e. absorption was relatively high compared to scattering), which also indicates that there was a higher fraction of particles from anthropogenic combustion sources. The $\sigma_{sca}$ had high values also in summer but the $\sigma_{abs}$ had its minimum and therefore the $\omega_0$
10  reached it maximum in summer. This observation indicates that the particle concentration was high in summer due to active vegetation.

Closer investigation on the size distribution revealed that
15   the seasonal variations of $b$ and $\alpha_{sca}$ were caused by shifting accumulation mode and not by concentration of coarse mode particles. In summer, $b$ and $\alpha_{sca}$ had their maxima (i.e. there was a higher fraction of smaller accumulation mode particles); and in winter, they had their minima (i.e. there was a higher fraction of large accumulation mode particles).

The extensive AOPs, as well as the aerosol number and volume concentration, tended to decrease. Our observation was in line with the other studies conducted in Europe and North America, which  also observed decreasing trends for the extensive AOPs (Collaud Coen et al., 2013; Pandolfi et al., 2018; Sherman et al., 2015), number concentration (Asmi et al., 2013) and aerosol optical depth (Li et al., 2014). This uniform decreasing trend in the amount of aerosol particles suggests that
25  the anthropogenic emissions  have been decreasing in Europe and North America. The observed tendency for $b$ and $\alpha_{sca}$ to increase together with the decreasing extensive properties indicated that the particle size distribution consisted of less larger particles. A more detailed investigation revealed that the number of larger accumulation mode particles (500–800 nm in diameter) decreased relatively the fastest, which  would indicate a decrease in transported anthropogenic pollution

 at SMEAR II.

Since the aerosol particles were scattering light more efficiently to backward hemisphere and because they were less dark than before, their RFE tended to decrease (i.e. became more negative), which means that the ability of aerosols to cool the climate per unit $\delta$ increased. However, since the extensive properties and particle number concentration  were

5  decreasing, which means that the $\delta$  decreased as well, the total aerosol forcing  was probably also decreasing. ~~We determined the RFE to dry aerosol particles by using global average values suggested by . To test the sensitivity of RFE to environmental parameters ($D$, $R_S$, and $A_C$), we calculated the RFE also by using more realistic and seasonally averaged environmental parameters. We also determined the RFE for ambient RH, since it is affected by the hygroscopic growth of aerosols. We observed that at SMEAR II the environmental parameters had a higher impact on the RFE than the ambient RH.~~

[revised manuscript text omitted]
. Subplots a) – d) describe the correlation between the PM10 AOPs and GMD; subplots e) – h) describe the correlation between the PM10 AOPs and VMDtot; and the subplots i) – l) describe the correlation between the PM1 AOPs and VMD_fine. The correlation coefficients of the linear regressions are given in each subfigure. The color-coding represents the number of data points in a grid point. In each subfigure, there are 100 grid points on both axes, making 10 000 grid points in total. The orange and black lines represent the values calculated from the unimodal size distributions, which were generated for different GMDs with geometric standard deviation GSD = 2.0 and 1.5 nm. The scattering was modeled from the generated size distribution at wavelengths 450, 550, and 700 nm with a refractive index $m = 1.517 + 0.19i$.

[Figure]

Figure 4: Median volume and number size distributions for the various VMD and GMD limits. The median $b$ and $\alpha_{sca}$ for the VMD and GMD limits are given in each legend box. The vertical grid lines represent the typical diameter limits for the nucleation, Aitken, accumulation and coarse particle modes (same as in Figs. S4 and S6). a) Volume size distribution for different PM10 VMD$_{tot}$ limits. b) Volume size distribution for different PM1 VMD$_{fine}$ limits. c) Volume and number size distribution for different PM10 and GMD limits. The c figure also represents volume and number size distribution for different PM1 and GMD limits as well, since the GMD is practically the same for PM10 and PM1 particles.

[Figure]

Figure 5: Seasonal variation in the aerosol optical properties for PM10 particles. The boxes represent the 25[th] and 75[th] percentiles and the whiskers the 10[th] and 90[th] percentiles of the data. The orange line is the median and the mean is presented with a black circle.

[Figure]

[Figure]

**Figure 7̶6:** **Variations in the different radiative forcing efficiencies at SMEAR II in 2006 – 2018.** a) Time series of the $RFE_{H\&S}$, $RFE_S$, and $RFE_{S,moist}$. The monthly medians are presented if the month had at least 14 days of valid data. b) Seasonal variation of the $RFE_{H\&S}$, $RFE_S$, and $RFE_{S,moist}$ as overall monthly medians. RFE was calculated for PM10 particles.

[Figure]

Figure 8: Relationships between $\omega_0$, $b$ and RFE$_{H\&S}$. The RFE$_{H\&S}$ is shown as the dashed isolines in the background. The boxes represent the data measured at SMEAR II and they are colored by the median $\sigma_{sca}$. The explanation for the boxplots is the same as in Fig. 5.

---

## Author Response (AR3)

**AUTHOR'S RESPONSE TO EDITOR'S REVIEW**

Manuscript: ACP-2018-981

Here are the answers to the comments. Also, the marked up version is attached at the end of this document.

The editor's comments are marked with bold text and the answers are in normal font.

**GENERAL REMARKS**

**You often use the phrase "The X is …" when you refer to the value of the property X. I suggest rephrasing it in way like "The value of X …" or "The parameter X …".**

I rephrased the parts of the text where we refer to the values.

**Please check also the use of singular and plural throughout the text.**

I did some corrections to this issue.

**You sometimes use the phrase "white" and "dark" aerosol. This is not very physical. Particularly the phrase "dark" should be replaced by "absorbing".**

I rephrased the parts where we referred to "dark" or "light colored" aerosols.

**When you introduce RFE on page 10, line 3 you must mention here that the values you use for the calculation of RFE are global average values which may be completely different from the regional situation. Calculating RFE by these global mean values does not mean anything for the regional effects but may only be used to compare the aerosol properties to values reported for other types of aerosol from other parts of the world. You say that in the next paragraph but a clear statement at the very beginning would be better.**

Added a statement in the beginning of the paragraph that states that the RFE determined by using the global averages does not necessarily give realistic values at SMEAR II.

**SPECIFIC COMMENTS**

**Abstract, page 1, line 16: it should read "… aerosol number and volume concentrations …". Please check the text since you used the singular word "concentration" a couple of times when the plural would be correct.**

**Same paragraph, line 18: I suggest writing: "… absorb less light than at the beginning of the measurements." The same sentence is written in the main text later where I suggest also correcting the phrase.**

**Page 2, line 22: rephrase: "… particles formed by gas-to-particle conversion." Same page, line32: Rephrase "… both scattering and absorption coefficients …".**

**Page 4, line1: I suggest introducing the symbols for the Ångström exponents here, like you do in the next line for the extensive properties.**

**Page 5, It should read"… described by the empirical …".**

**Page 8, line 11: It should read "…linked to the …".**

**Page 9, line 11: Delete "are".**

**Page 10, line 4: Please insert: "the values were chosen according to …".**

**Page 16: There is no description or reference given for Fig. 2b.**

**Page 17: There are no descriptions or references given for Figs. 5b and 5c.**

**Page 21, 2nd paragraph: I assume you refer here to Fig. 6b instead of Fig. 7b.**

We have fixed all the specific issues according to the suggestions.

[revised manuscript text omitted]